# KAGE-Bench: Fast Known-Axis Visual Generalization Evaluation for Reinforcement Learning

Egor Cherepanov [1 2]   Daniil Zelezetsky [2]   Aleksandr I. Panov [1 2]   Alexey K. Kovalev [1 2]

avanturist322.github.io/KAGEBench

## Abstract

Pixel-based reinforcement learning agents often fail under purely visual distribution shift even when latent dynamics and rewards are unchanged, but existing benchmarks entangle multiple sources of shift and hinder systematic analysis. We introduce **KAGE-Env**, a JAX-native 2D platformer that factorizes the observation process into independently controllable visual axes while keeping the underlying control problem fixed. By construction, varying a visual axis affects performance only through the induced state-conditional action distribution of a pixel policy, providing a clean abstraction for visual generalization. Building on this environment, we define **KAGE-Bench**, a benchmark of six known-axis suites comprising 34 train–evaluation configuration pairs that isolate individual visual shifts. Using a standard PPO-CNN baseline, we observe strong axis-dependent failures, with background and photometric shifts often collapsing success, while agent-appearance shifts are comparatively benign. Several shifts preserve forward motion while breaking task completion, showing that return alone can obscure generalization failures. Finally, the fully vectorized JAX implementation enables up to 33M environment steps per second on a single GPU, enabling fast and reproducible sweeps over visual factors. The code is in the supplementary material.

## 1. Introduction

Reinforcement learning (RL) agents trained from high-dimensional pixel observations are brittle to changes in appearance, lighting, and other visual nuisance factors (Cetin et al., 2022; Yuan et al., 2023; Klepach et al., 2025). Policies that perform well in-distribution can degrade sharply under

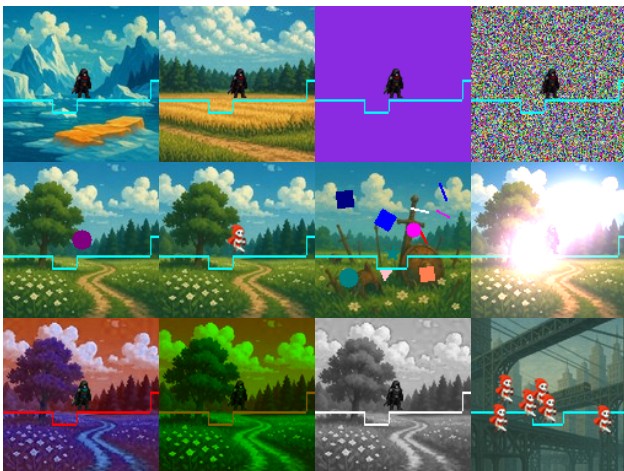

*Figure 1.* **Representative observations from KAGE-Env illustrating controlled, known-axis visual variation.** Each panel differs along one or more explicitly configurable axes, including background imagery and color, agent appearance and animation, moving distractors, photometric filters, and dynamic lighting effects, while task semantics and underlying dynamics are held fixed.

purely visual distribution shifts, even when task semantics, transition dynamics, and rewards are unchanged (Staroverov et al., 2023; Kachaev et al., 2025; Mirjalili et al., 2025). This brittleness poses a fundamental obstacle to real-world deployment, where observations inevitably vary due to viewpoint changes, illumination, surface appearance, and sensor noise while the control-relevant latent state remains fixed (Raileanu et al., 2020; Kostrikov et al., 2020; Kirilenko et al., 2023; Korchemnyi et al., 2024; Yang et al., 2024; Ugadiarov et al., 2026). As a result, pixel-based RL policies that rely on incidental visual correlations can fail abruptly despite convergence, undermining reliability in robotics, autonomous navigation, and interactive environments (Stone et al., 2021; Yuan et al., 2023). More broadly, visual generalization is needed wherever models must robustly extract information from visual structure, even in scientific texts and figures (Sherki et al., 2025).

Despite substantial progress in representation learning (Mazoure et al., 2021; Rahman & Xue, 2022; Ortiz et al., 2024) and data augmentation (Laskin et al., 2020a; Raileanu et al., 2020; Hansen & Wang, 2021), understanding visual generalization failures remains challenging. A central obstacle lies in evaluation benchmarks, which often entangle multiple

[1]AXXX, Moscow, Russia [2]MIRAI, Moscow, Russia. Correspondence to: Egor Cherepanov <cherepanov@axxx.tech>.

*Proceedings of the 43rd International Conference on Machine Learning*, Seoul, South Korea. PMLR 306, 2026. Copyright 2026 by the author(s).

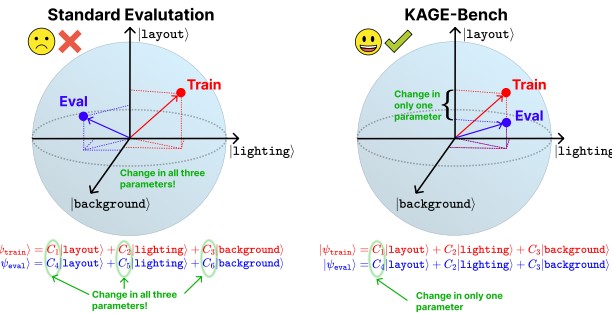

*Figure 2.* **KAGE-Bench: Motivation.** Existing generalization benchmarks entangle multiple sources of visual shift between training and evaluation, making failures difficult to attribute. KAGE-Bench factorizes observations into independently controllable axes and constructs train–evaluation splits that vary one (or a selected set) of axes at a time, enabling precise diagnosis of which visual factors drive generalization gaps. The observation vector notation $|\psi\rangle$ is used for intuition only.

visual and structural changes such as background appearance, geometry, dynamics, and distractors (Cobbe et al., 2020; Stone et al., 2021; Yuan et al., 2023). In these settings, train–evaluation performance gaps cannot be cleanly attributed to specific sources of shift, and failures may reflect visual sensitivity, altered task structure, or interactions between confounded factors. Compounding this issue, many pixel-based RL environments are computationally expensive to simulate, limiting large-scale ablations and slowing hypothesis testing.

We address these limitations with **KAGE-Bench** (*Known-Axis Generalization Evaluation Benchmark*), a visual generalization benchmark in which sources of distribution shift are isolated by construction. KAGE-Bench is built on **KAGE-Env** (Figure 1), a JAX-native (Bradbury et al., 2018) 2D platformer whose observation process is factorized into independently controllable visual axes while latent dynamics and rewards are held fixed (see Figure 2). Under this known-axis design, each axis corresponds to a well-defined component of the observation kernel, and any train–evaluation performance difference arises solely from how a fixed observation-based policy responds to different renderings of the same latent states, enabling unambiguous attribution of visual generalization failures.

Systematic analysis of visual generalization requires evaluating many controlled shifts at scale. KAGE-Env is implemented entirely in JAX with end-to-end `jit` compilation and vectorized execution via `vmap` and `lax.scan`, enabling efficient large-batch simulation on a single accelerator. In practice, this design scales up to $2^{16}$ parallel environments on one GPU and achieves up to 33M environment steps per second (see Figure 3), making exhaustive sweeps over visual parameters and fine-grained diagnosis of generalization behavior feasible.

[1] https://colab.research.google.com/

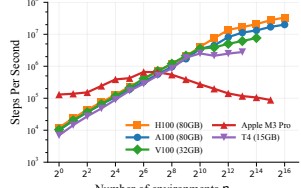
*(a)* Easy configuration.

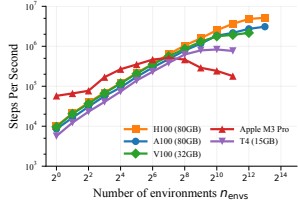
*(b)* Hard configuration.

*Figure 3.* **Environment stepping throughput vs. parallelism.** Environment stepping throughput (steps per second, higher is better) as a function of the number of parallel environments $n_{\text{envs}}$ for KAGE-Env across heterogeneous hardware backends. GPU results are shown for NVIDIA H100 (80 GB), A100 (80 GB), V100 (32 GB), and T4 (15 GB, Google Colab[1]), with CPU-only results on an Apple M3 Pro laptop. **(a) Easy configuration:** lightweight setup with all visual generalization parameters disabled. **(b) Hard configuration:** most demanding setup with all visual generalization parameters enabled at maximum values.

Building on this environment, we construct six visual generalization suites comprising 34 train–evaluation configuration pairs, each targeting a specific visual axis. Using these suites, we demonstrate that visual generalization is strongly axis-dependent and identify classes of visual shifts that reliably induce severe performance degradation, even for a standard PPO-CNN baseline (Schulman et al., 2017). Representative negligible, moderate, and severe train–evaluation gaps are previewed in Figure 4.

> **We summarize our main contributions as follows:**
>
> 1. **KAGE-Env**, a JAX-native RL environment with 93 explicitly controllable parameters, configurable via a single `.yaml` file and vectorized to reach up to 33M environment steps per second with $2^{16}$ parallel environments on a single GPU.
>
> 2. **KAGE-Bench**, a benchmark that isolates visual distribution shifts by construction via six known-axis suites and 34 train–evaluation configuration pairs with fixed dynamics and rewards.
>
> 3. **Empirical diagnosis of visual generalization:** using a PPO-CNN baseline, we quantify how visual generalization behavior differs across axes and identify classes of visual shifts that reliably induce severe performance degradation.

## 2. Related Work

**Visual generalization in RL.** Visual generalization studies whether policies trained from pixel observations retain performance when the observation process changes while latent dynamics and rewards remain fixed. Prior work shows that agents often overfit incidental visual features, leading to substantial train–test gaps across a wide range of envi-

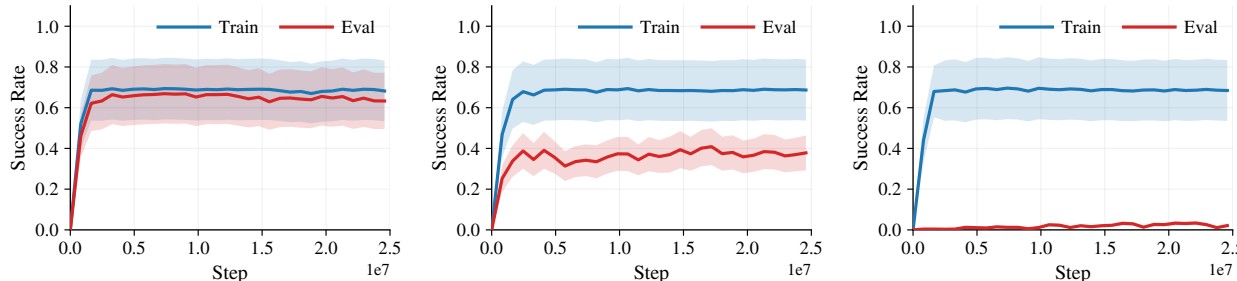

*Figure 4.* **Examples of visual generalization gaps.** Success rate for three train–evaluation pairs showing (left) negligible, (middle) moderate, and (right) severe generalization gaps.

ronments and settings (Cobbe et al., 2019; Beattie et al., 2016; Xia et al., 2018; Ortiz et al., 2024). A common explanation is that standard architectures and objectives exploit spurious visual correlations, such as background textures or color statistics, rather than learning task-relevant invariances (Cobbe et al., 2020; Hansen & Wang, 2021; Stone et al., 2021). Accordingly, many approaches have been proposed to improve robustness, including data augmentation, auxiliary representation learning objectives, and regularization methods (Laskin et al., 2020a; Raileanu et al., 2020; Mazoure et al., 2021; Raileanu & Fergus, 2021; Cobbe et al., 2021; Wang et al., 2020; Bertoin & Rachelson, 2022; Bertoin et al., 2022; Zisselman et al., 2023; Rahman & Xue, 2022; Jesson & Jiang, 2024). KAGE-Env and KAGE-Bench provide diagnostic infrastructure for this literature by enabling fast, controlled, axis-specific evaluation that isolates changes in the observation kernel.

**Benchmarks for visual generalization in RL.** A range of benchmarks study visual generalization in pixel-based RL, differing in task domains and in how explicitly they isolate sources of visual variation. RL-ViGen (Yuan et al., 2023) spans multiple domains, including locomotion, manipulation, navigation, and driving, with shifts in textures, lighting, viewpoints, layouts, and embodiments. Hansen & Wang (2021) evaluates continuous control under controlled appearance changes such as color randomization and dynamic video backgrounds. Obstacle Tower (Juliani et al., 2019) and LevDoom (Tomilin et al., 2022) consider 3D settings where many factors vary jointly, making attribution of failures to specific visual causes difficult. Related benchmarks such as DMC-VB (Ortiz et al., 2024) and Distracting Meta-World (Kim et al., 2024) introduce task-irrelevant visual distractors while keeping task dynamics fixed.

Among widely used benchmarks, Procgen (Cobbe et al., 2020) relies on procedural generation, so train–test gaps typically reflect entangled shifts in appearance and scene composition rather than isolated visual factors. The Distracting Control Suite (DCS) (Stone et al., 2021) introduces explicit distraction axes but is limited to a small set of factors, and broad axis-wise sweeps are costly in its underlying continuous-control simulator. KAGE-Env

and KAGE-Bench complement these benchmarks by explicitly factorizing the observation process into independently controllable visual axes. KAGE-Env uses a simple platformer to reduce optimization and exploration confounds, while KAGE-Bench constructs train–evaluation splits that vary specified axes (e.g., backgrounds, sprites, distractors, filters, and lighting) with fixed dynamics and rewards, enabling systematic, axis-specific attribution of generalization failures. Compared with DMControl Generalization and DCS-style evaluation, KAGE-Bench emphasizes a fast JAX-native pixel environment where axis-isolated visual shifts can be swept systematically rather than sampled from a small number of expensive simulator variants. This also addresses a practical gap in the JAX RL ecosystem. MuJoCo Playground (Zakka et al., 2025) provides MJX-based robot-learning environments with state and pixel inputs, and Octax (Radji et al., 2025) provides high-throughput JAX arcade environments. KAGE-Env is complementary because it targets controllable RGB rendering axes and paired train–evaluation shifts for known-axis attribution. Combining MJX-style simulators with similarly controlled visual-shift layers remains a promising future direction.

**Fast and scalable evaluation in RL.** Evaluating generalization in RL is sample intensive, as reliable conclusions require averaging over random seeds, environment instances, and distribution shifts. In visual generalization benchmarks, this leads to combinatorial scaling $N_{\text{steps}} \times N_{\text{seeds}} \times N_{\text{shifts}}$, often compounded by checkpointing and hyperparameter sweeps, making evaluation costly in CPU-bound simulators.

Recent work addresses this bottleneck through *accelerator-native* RL systems, where environment stepping is implemented as compiled, vectorized computation on GPUs or TPUs. Examples include JAX-based simulators such as Brax (Freeman et al., 2021), Jumanji (Bonnet et al., 2023), XLand-MiniGrid (Nikulin et al., 2024), CAMAR (Pshenitsyn et al., 2025), and Craftax (Matthews et al., 2024), as well as GPU-native platforms such as ManiSkill3 (Tao et al., 2024), MIKASA-Robo (Cherepanov et al., 2026), and Warp-Drive (Lan et al., 2021). By eliminating host-side control flow, these systems achieve orders-of-magnitude throughput. However, high throughput alone does not yield diag-

nostic evaluation of visual robustness. Benchmarks such as Procgen and DCS do not support exhaustive, axis-isolated sweeps over rendering factors, limiting failure attribution. KAGE-Env combines the accelerator-native paradigm with explicit factorization of the observation process into independently controllable axes, enabling large-batch, reproducible evaluation of known-axis visual shifts under fixed latent dynamics and rewards. This distinction is practical: raw KAGE-Env stepping reaches 33M environment steps per second, while the full PPO-CNN training pipeline used for the benchmark runs at about 52k training steps per second per GPU after CNN forward passes, backpropagation, optimizer updates, logging, checkpointing, and video generation on $128 \times 128 \times 3$ RGB inputs.

## 3. Background

**Partially Observable Markov Decision Processes.** We consider episodic control with horizon $T$ in a partially observable Markov decision process (POMDP). Each environment instance is indexed by a visual configuration $\xi \in \Xi$ and defined as $\mathcal{M}_\xi = (\mathcal{S}, \mathcal{A}, P, r, \Omega, O_\xi, \rho_0, \gamma)$, where $\mathcal{S}$ is the latent (control-relevant) state space, $\mathcal{A}$ is the action space, $P(\cdot \mid s, a)$ is the transition kernel, $r(s, a)$ is the reward function, $\Omega$ is the observation space, $O_\xi(\cdot \mid s)$ is the observation (rendering) kernel parameterized by $\xi$, $\rho_0$ is the initial state distribution, and $\gamma \in [0, 1)$ is the discount factor. At each timestep $t$, the environment occupies a latent state $s_t \in \mathcal{S}$. An observation is generated according to $o_t \sim O_\xi(\cdot \mid s_t)$, $o_t \in \Omega \subseteq \{0, \dots, 255\}^{H \times W \times 3}$. Based on this observation, the agent selects an action $a_t \in \mathcal{A}$, receives reward $r(s_t, a_t)$, and transitions to $s_{t+1} \sim P(\cdot \mid s_t, a_t)$.

A key structural property enforced throughout this work is that the transition kernel $P$ and reward function $r$ are *independent* of the visual configuration $\xi$. All dependence on $\xi$ is confined to the observation kernel $O_\xi$. Consequently, the same latent state $s_t$ may give rise to different observations under different values of $\xi$, while inducing identical dynamics and rewards. Visual generalization concerns the behavior of policies under such changes in the observation process, with the underlying control problem held fixed.

**Policies and return.** We focus on reactive pixel-based policies that map observations directly to action distributions: $\pi(a \mid o)$. The expected discounted return of a policy $\pi$ in environment $\mathcal{M}_\xi$ is

$$J(\pi; \mathcal{M}_\xi) = \mathbb{E}_{\substack{s_0 \sim \rho_0, \\ o_t \sim O_\xi(\cdot \mid s_t), \\ a_t \sim \pi(\cdot \mid o_t), \\ P}} \left[ \sum_{t=0}^{T-1} \gamma^t r(s_t, a_t) \right]. \quad (1)$$

**Visual generalization.** We study generalization under shifts in visual parameters that affect observations but not the underlying control problem. Let $\Xi$ denote the space of visual configurations, and let $\mathcal{D}_{\text{train}}$ and $\mathcal{D}_{\text{eval}}$ be probability distributions over $\Xi$. Each $\xi \in \Xi$ induces a visual POMDP $\mathcal{M}_\xi$ through its observation kernel $O_\xi$, while sharing the same latent dynamics $P$ and reward function $r$.

A pixel policy $\pi(a \mid o)$ is trained using environments with $\xi \sim \mathcal{D}_{\text{train}}$ and evaluated under $\xi \sim \mathcal{D}_{\text{eval}}$. For any distribution $\mathcal{D}$ over $\Xi$, we define the expected performance

$$J(\pi; \mathcal{D}) = \mathbb{E}_{\xi \sim \mathcal{D}}[J(\pi; \mathcal{M}_\xi)]. \quad (2)$$

We refer to this setting as *visual generalization* when the shift from $\mathcal{D}_{\text{train}}$ to $\mathcal{D}_{\text{eval}}$ changes only the observation kernels $O_\xi$, while preserving the latent state space, transition dynamics, and reward function.

**Known-axis visual shifts.** KAGE-Bench focuses on *known-axis* visual generalization. Each visual configuration is decomposed as $\xi = (\xi_{\text{axis}}, \xi_{\text{rest}})$, where $\xi_{\text{axis}}$ specifies a designated axis of visual variation (e.g., background appearance, agent sprites, lighting, filters), and $\xi_{\text{rest}}$ contains all remaining parameters. By construction, any performance difference between training and evaluation can therefore be attributed to changes in the observation process along the specified visual axis, rather than to changes in task structure, dynamics, or rewards. This intuition is formalized and justified in Section 4 and Appendix A.

**Evaluation metrics.** Given $\mathcal{D}_{\text{train}}$ and $\mathcal{D}_{\text{eval}}$, we report in-distribution and out-of-distribution performance, $J(\pi; \mathcal{D}_{\text{train}})$ and $J(\pi; \mathcal{D}_{\text{eval}})$, and define the return-based generalization gap

$$\Delta(\pi) = J(\pi; \mathcal{D}_{\text{train}}) - J(\pi; \mathcal{D}_{\text{eval}}). \quad (3)$$

While $\Delta(\pi)$ provides a coarse measure of performance degradation under visual shift, it is insufficient to fully characterize generalization behavior. The discounted return aggregates multiple effects, including reward shaping, exploration inefficiency, and penalty terms, and may obscure whether an agent nearly solves the task or fails catastrophically. In particular, if a policy fails under both training and evaluation configurations, the return gap can be small despite the absence of task competence.

For this reason, we complement return-based evaluation with additional trajectory-level metrics that are measurable functions of the latent state trajectory, including distance traveled, normalized progress toward the goal, and binary task success. These metrics distinguish partial progress from complete failure and provide a more fine-grained view of visual generalization behavior. Their precise definitions and empirical use are described in Section 6.

# 4. Known-axis visual generalization

This section states the formal principle behind KAGE-Bench. In our construction (Section 3), the latent control problem is fixed and only the renderer changes: $\xi$ affects performance *only through* the induced state-conditional action law obtained by composing the observation kernel with the pixel policy. The goal is to make this channel explicit and to justify the benchmark protocol: (i) constructing suites that intervene on a single visual axis, and (ii) evaluating not only return but also trajectory-level metrics such as distance, progress, and success.

**From pixel policies to state-conditional behavior.** A reactive pixel policy $\pi(\cdot \mid o)$ maps observations to actions and does not directly specify an action distribution conditioned on the latent state $s$. However, in a visual POMDP $\mathcal{M}_\xi$, the observation kernel $O_\xi(\cdot \mid s)$ induces a distribution over rendered observations for each latent state. Composing these kernels yields a well-defined state-conditional action distribution by marginalizing the intermediate observation:

$$s \xrightarrow{O_\xi(\cdot|s)} o \xrightarrow{\pi(\cdot|o)} a. \qquad (4)$$

Under our construction (and for reactive policies), this composition is the only mechanism by which the visual configuration $\xi$ can affect control, since $P$ and $r$ are invariant across $\xi$. Figure 5 illustrates this marginalization in a concrete discrete example.

**Definition 4.1** (Induced state policy). Fix $\xi \in \Xi$, observation kernel $O_\xi(\cdot \mid s)$, and reactive pixel policy $\pi(\cdot \mid o)$. The **induced state policy** $\pi_\xi$ is defined by

$$\pi_\xi(a \mid s) \; := \; \int_\Omega \pi(a \mid o)\, O_\xi(do \mid s), \qquad (5)$$
$$\forall s \in \mathcal{S},\ \forall a \in \mathcal{A}.$$

For a fixed pixel policy $\pi$, the map $\xi \mapsto \pi_\xi$ summarizes the effect of visual variation on state-conditional behavior. In particular, changing $\xi$ changes $\pi_\xi$ while leaving the latent control problem $(\mathcal{S}, \mathcal{A}, P, r, \rho_0, \gamma)$ unchanged.

**Visual shift is equivalent to induced policy shift.** The next theorem formalizes the reduction used throughout KAGE-Bench: executing $\pi$ in the visual POMDP $\mathcal{M}_\xi$ induces the same latent state–action law as executing $\pi_\xi$ in the latent MDP $\mathcal{M}$.

**Theorem 4.2** (Visual generalization reduces to induced policy shift). *Fix any $\xi \in \Xi$ and reactive pixel policy $\pi(\cdot \mid o)$, and let $\pi_\xi$ be defined by Definition 4.1. Then:*

1. (***Conditional action law.***) $\forall t \geq 0$, $\forall a \in \mathcal{A}$,

$$\mathbb{P}_{\mathcal{M}_\xi, \pi}(a_t = a \mid s_t) \; = \; \pi_\xi(a \mid s_t) \quad a.s. \qquad (6)$$

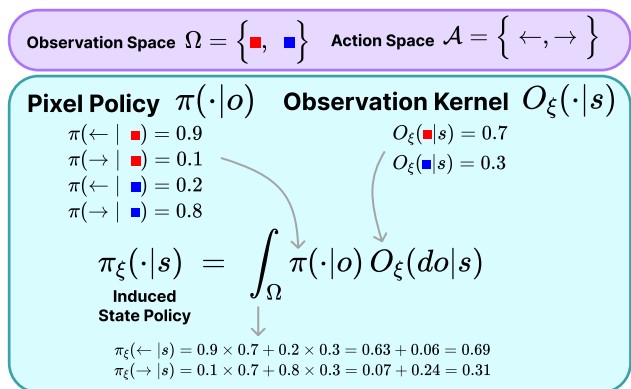

*Figure 5.* **Induced state policy.** The renderer $O_\xi(\cdot \mid s)$ maps a latent state to an observation distribution, and the pixel policy $\pi(\cdot \mid o)$ maps observations to actions. Their composition defines $\pi_\xi(\cdot \mid s)$ by marginalizing $o$.

2. (***Equality in law of state–action processes.***) *The state–action process $(s_t, a_t)_{t \geq 0}$ induced by executing $\pi$ in $\mathcal{M}_\xi$ has the same law as the state–action process induced by executing $\pi_\xi$ in the latent MDP $\mathcal{M}$.*

3. (***Return equivalence.***) *Consequently,*

$$J(\pi; \mathcal{M}_\xi) = J(\pi_\xi; \mathcal{M}). \qquad (7)$$

Theorem 4.2 is purely representational: it does not assume optimality and it does not modify the control problem. A useful consequence is the identity, for any $\xi, \xi' \in \Xi$,

$$J(\pi; \mathcal{M}_\xi) - J(\pi; \mathcal{M}_{\xi'}) \; = \; J(\pi_\xi; \mathcal{M}) - J(\pi_{\xi'}; \mathcal{M}), \quad (8)$$

which states that a visual train–evaluation gap for a fixed pixel policy $\pi$ is exactly a performance difference between induced state policies in the same latent MDP. This is the formal basis for attributing failures to the observation process: since $(P, r)$ are unchanged, any degradation under $\xi \to \xi'$ must be explained by how the renderer changes the induced state-conditional behavior $\pi_\xi$.

**Why known-axis suites enable axis-specific attribution.** KAGE-Bench constructs axis-isolated suites by decomposing $\xi = (\xi_{\text{axis}}, \xi_{\text{rest}})$ and pairing train and evaluation configurations that differ only in the designated axis:

$$\xi^{\text{train}} = (\xi_{\text{axis}}^{\text{train}}, \xi_{\text{rest}}) \text{ and } \xi^{\text{eval}} = (\xi_{\text{axis}}^{\text{eval}}, \xi_{\text{rest}}). \quad (9)$$

Equivalently, for all $s \in S$ the paired renderers satisfy $O_{\xi^{\text{train}}}(\cdot \mid s) = O(\cdot \mid s; \xi_{\text{axis}}^{\text{train}}, \xi_{\text{rest}})$ and $O_{\xi^{\text{eval}}}(\cdot \mid s) = O(\cdot \mid s; \xi_{\text{axis}}^{\text{eval}}, \xi_{\text{rest}})$, so the only change in the observation process is along $\xi_{\text{axis}}$. Under this controlled-intervention design, the induced policies $\pi_{\xi^{\text{train}}}$ and $\pi_{\xi^{\text{eval}}}$ differ only through this axis-dependent change in $O_\xi$. Therefore, by Equation 8, the measured gap isolates how that visual axis perturbs the induced state-conditional behavior of $\pi$.

**Trajectory-level consequences and evaluation metrics.** By Item 2 of Theorem 4.2, the latent state–action trajectory has the same law under $(\mathcal{M}_\xi, \pi)$ and $(\mathcal{M}, \pi_\xi)$, so the

```python
import jax
from kage_bench import (
    KAGE_Env,
    load_config_from_yaml,
)
# Create environment with custom config
env = KAGE_Env(
    load_config_from_yaml("custom_config.yaml")
)
# Vectorize and JIT compile
reset_vec = jax.jit(jax.vmap(env.reset))
step_vec = jax.jit(jax.vmap(env.step))
# Initialize 65,536 parallel environments
N_ENVS = 2**16
keys = jax.random.split(
    jax.random.PRNGKey(42), N_ENVS
)
# Reset all at once
obs, info = reset_vec(keys)
states = info["state"]
# Parallel step: Samples one random discrete
# action per env in [0, 7] (bitmask actions)
actions = jax.random.randint(
    keys[0], (N_ENVS,), 0, 8
)
# obs.shape: (65536, 128, 128, 3)
obs, rewards, terms, truncs, info \
    = step_vec(states, actions)
states = info["state"]
```

*Code 1.* **Python (JAX) usage.** The environment is configured from a `.yaml` file (e.g., `custom_config.yaml`); the code shows JAX-vmap/jit batched `reset/step` over $2^{16}$ parallel envs.

reduction applies to any measurable trajectory functional, not only return. We therefore report distance, progress, and success in addition to episodic return: these are functions of the latent trajectory exposed by KAGE-Env for evaluation, and their gaps under $\xi \to \xi'$ admit the same induced-policy interpretation. Unlike return, which can mask completion failures due to reward shaping, these metrics separate partial progress from task completion.

**Corollary 4.3** (Equivalence of trajectory-level evaluation metrics). *Fix $\xi \in \Xi$ and reactive $\pi(\cdot \mid o)$, and let $\pi_\xi$ be the induced state policy. Let $(s_t, a_t)_{t\geq 0} \sim (\mathcal{M}_\xi, \pi)$ and $(\tilde{s}_t, \tilde{a}_t)_{t\geq 0} \sim (\mathcal{M}, \pi_\xi)$. Then for any measurable functional $F : (S \times A)^{\mathbb{N}} \to \mathbb{R}$,*

$$F\big((s_t, a_t)_{t\geq 0}\big) \ \overset{d}{=} \ F\big((\tilde{s}_t, \tilde{a}_t)_{t\geq 0}\big),$$

*and in particular $\mathbb{E}_{\mathcal{M}_\xi, \pi}[F] = \mathbb{E}_{\mathcal{M}, \pi_\xi}[F]$ whenever the expectation is well-defined.*

**Corollary 4.4** (Specialization to KAGE-Bench metrics). *Assume the latent state contains a one-dimensional position variable $x_t \in \mathbb{R}$ with initial position $x_{\text{init}}$ and task completion threshold $D > 0$. For a fixed horizon $T$ (or terminal time), define $F_{\text{dist}} := x_T - x_{\text{init}}$, $F_{\text{prog}} := \frac{x_T - x_{\text{init}}}{D}$, and $F_{\text{succ}} := \mathbb{I}\{x_T - x_{\text{init}} \geq D\}$. Then each metric has the same distribution under $(\mathcal{M}_\xi, \pi)$ and $(\mathcal{M}, \pi_\xi)$, and in particular $\mathbb{E}_{\mathcal{M}_\xi, \pi}[F] = \mathbb{E}_{\mathcal{M}, \pi_\xi}[F]$, $F \in \{F_{\text{dist}}, F_{\text{prog}}, F_{\text{succ}}\}$.*

All proofs are deferred to Appendix A.

# 5. KAGE-Environment

KAGE-Env (Figure 1, Code 1) is a JAX-native RL environment designed for controlled evaluation of visual generalization. It implements the visual-POMDP interface from Section 3: configurations $\xi \in \Xi$ parameterize the renderer $O_\xi(\cdot \mid s)$ while the latent control problem is held fixed.

**Task and interface.** KAGE-Env is an episodic 2D side-scrolling platformer with horizon $T$ and a push-scrolling camera. At each timestep $t$, the agent observes a single RGB image $o_t \in \{0, \ldots, 255\}^{H \times W \times 3}$, with default resolution $H = W = 128$, and selects an action $a_t$ from a discrete action space $\mathcal{A} = \{0, \ldots, 7\}$. Actions are encoded as a bitmask over three primitives: LEFT = 1, RIGHT = 2, and JUMP = 4. Policies interact with the environment exclusively through pixels; the latent simulator state is not available to the policy and is exposed only via the `info` dictionary for logging and evaluation. The LEFT action is retained to preserve a conventional platformer-style action space, although a competent policy can learn to ignore it when moving left is not useful.

**Reward and termination.** Let $x_t \in \mathbb{R}$ denote horizontal position and $x_t^{\max} := \max_{0 \leq k \leq t} x_k$ the furthest position reached so far. The per-step reward is

$$r_t = \underbrace{\alpha_1 \max\{0, \, x_{t+1} - x_t^{\max}\}}_{\text{first-time forward progress}}$$
$$- \underbrace{\Big(\alpha_2 \, \mathbb{I}[\text{JUMP}(a_t)] + \alpha_3 + \alpha_4 \, \mathbb{I}[\text{idle}(x_t, x_{t+1})]\Big)}_{\text{penalties}}, \quad (10)$$

where $\mathbb{I}[\text{JUMP}(a_t)]$ indicates the jump bit is active in $a_t$, $\alpha_3$ is a per-timestep time cost, and $\text{idle}(x_t, x_{t+1})$ flags lack of horizontal progress. Episodes terminate only by time-limit truncation at $T = \text{episode\_length}$.

```yaml
background:
  mode: "image"
  image_paths:
    - "src/kage/assets/backgrounds/bg-1.jpeg"
    - "src/kage/assets/backgrounds/bg-64.jpeg"
    - "src/kage/assets/backgrounds/bg-128.jpeg"
  parallax_factor: 0.5
  switch_frequency: 0.0
character:
  mode: "sprite"
  sprite_paths:
    - "src/kage/assets/sprites/clown"
    - "src/kage/assets/sprites/skeleton"
  enable_animation: true
  animation_fps: 12.0
npc:
  mode: "sprite"
  sprite_dir: "src/kage/assets/sprites"
filters:
  brightness: 0.0
  hue_shift: 0.0
```

*Code 2.* **YAML configuration.** KAGE-Env is configured via a single `.yaml` file; shown is a small excerpt of `custom_config.yaml`. We show only a small part of all configuration parameters; for details, see Appendix H.

*Table 1.* **Axis-level summary of KAGE-Bench results (mean±SEM).** During training of each run, we record the maximum value attained by each metric. For each configuration, these per-run maxima are averaged across 10 random seeds, and the resulting per-configuration values are then averaged across all configurations within each generalization-axis suite. We report Distance, Progress, Success Rate (SR), and Return for train and eval configurations, along with the corresponding generalization gaps (mean±SEM). Generalization gaps are color-coded: green indicates smaller gaps (better generalization), while red indicates larger gaps (worse generalization). $\Delta\text{Dist.} = \frac{\text{Dist.}^{\text{train}} - \text{Dist.}^{\text{eval}}}{\text{Dist.}^{\text{train}}} \times 100\%$, $\Delta\text{Prog.} = \frac{\text{Progress}^{\text{train}} - \text{Progress}^{\text{eval}}}{\text{Progress}^{\text{train}}} \times 100\%$, $\Delta\text{SR} = \frac{\text{SR}^{\text{train}} - \text{SR}^{\text{eval}}}{\text{SR}^{\text{train}}} \times 100\%$, $\Delta\text{Ret.} = |\text{Ret.}^{\text{train}} - \text{Ret.}^{\text{eval}}|$.

| | Evaluation on train config | | | | Evaluation on eval config | | | | Generalization gap | | | |
|---|---|---|---|---|---|---|---|---|---|---|---|---|
| | Distance | Progress | SR | Return | Distance | Progress | SR | Return | ΔDist., % | ΔProg., % | ΔSR, % | ΔRet., (abs.) |
| **Agent** | 396.5±26.8 | 0.81±0.05 | 0.76±0.06 | -292.73±77.21 | 386.9±26.1 | 0.79±0.05 | 0.60±0.06 | -408.8±88.5 | 2.4 | 2.5 | 21.1 | 116.1 |
| **Background** | 463.4±10.4 | 0.95±0.02 | 0.90±0.02 | -118.3±32.6 | 322.7±47.5 | 0.66±0.10 | 0.42±0.13 | -935.8±249.6 | 30.5 | 30.5 | 53.3 | 691.0 |
| **Distractors** | 413.5±22.9 | 0.84±0.05 | 0.81±0.05 | -178.0±41.5 | 397.0±23.6 | 0.81±0.05 | 0.56±0.11 | -307.0±78.1 | 4.0 | 3.6 | 30.9 | 129.0 |
| **Effects** | 426.3±15.6 | 0.87±0.03 | 0.82±0.03 | -224.3±64.0 | 337.7±10.8 | 0.69±0.02 | 0.16±0.06 | -725.1±65.6 | 20.8 | 20.7 | 80.5 | 500.8 |
| **Filters** | 431.2±19.3 | 0.88±0.04 | 0.83±0.04 | -204.8±59.6 | 380.6±18.1 | 0.78±0.04 | 0.11±0.04 | -652.4±70.5 | 11.7 | 11.4 | 86.8 | 447.6 |
| **Layout** | 452.3±0.0 | 0.92±0.00 | 0.86±0.00 | -118.6±0.0 | 434.1±0.0 | 0.89±0.00 | 0.32±0.00 | -279.5±0.0 | 4.0 | 3.3 | 62.8 | 160.9 |

**Rendering assets and visual parameters.** KAGE-Env provides a library of visual assets and rendering controls for constructing visual variation. Assets include 128 background images (Appendix, Figure 46) and 27 animated sprite skins for the agent and non-player characters (Appendix, Figure 47); when sprites are disabled, entities can be rendered as geometric shapes (9 types) with a palette of 21 colors. The renderer further exposes photometric and spatial transformations (e.g., brightness, contrast, gamma, hue, blur, noise, pixelation, vignetting) and lighting/overlay effects such as dynamic point lights with configurable count, intensity, radius, falloff, and color.

**Configuration interface.** All parameters are specified through a single `.yaml` configuration file (Code 2). A configuration $\xi \in \Xi$ is organized into groups `background`, `character`, `npc`, `distractors`, `filters`, `effects`, `layout`, and `physics`. These groups include rendering parameters (affecting only $O_\xi$) as well as optional control parameters (affecting $P$ or $r$). KAGE-Env exposes both for extensibility; isolation of purely visual shifts is enforced by the KAGE-Bench pairing protocol (Section 6).

## 6. KAGE-Benchmark

KAGE-Bench is a benchmark protocol built on top of KAGE-Env. It specifies how environment configurations are selected and paired to evaluate *known-axis visual generalization*. Concretely, KAGE-Bench defines a set of train–evaluation configuration pairs $(\xi^{\text{train}}, \xi^{\text{eval}})$ such that the underlying control problem is identical ($P^{\text{train}} = P^{\text{eval}}$, $r^{\text{train}} = r^{\text{eval}}$) and the two configurations differ only in a designated subset of rendering parameters.

**Benchmark construction.** We first conduct a pilot sweep over KAGE-Env's rendering parameters using a standard PPO-CNN, adopted from the CleanRL (Huang et al., 2022) library[2], trained from a single RGB frame without history or frame stacking. Hyperparameters are reported in Appendix, Table 5. This sweep measures how individual rendering parameters affect out-of-distribution performance

when the control problem is fixed. Based on these results, we curate **34 train–evaluation configuration pairs** that exhibit a range of generalization behavior, including both severe and mild gaps. The selected pairs are grouped into six suites corresponding to distinct visual axes: *agent appearance*, *background*, *distractors*, *effects*, *filters*, and *layout*. In each pair, exactly one parameter within the target axis is changed between train and evaluation, while all other parameters are held fixed. Easier pairs are intentionally retained as sanity checks, ensuring that the benchmark distinguishes lack of generalization from lack of task competence. The 34 pairs should be interpreted as a clear reference subset, not as a fixed universal difficulty ranking. PPO-CNN is used as a diagnostic probe to expose which controlled visual shifts break a standard pixel policy; different methods may expose different failure modes, and KAGE-Env configurations can be extended as methods improve. For additional robustness validation, we also evaluated baselines inspired by common visual-RL robustness methods: RAD-PPO with cutout-color augmentation, RAD-PPO with random crop-resize augmentation (Laskin et al., 2020a), CURL (Laskin et al., 2020b), DrQ (Kostrikov et al., 2020), and SVEA (Hansen et al., 2021). These baselines used official PyTorch implementations interfaced with KAGE-Env through a JAX-to-PyTorch wrapper and were evaluated on the most challenging configurations identified in the PPO-CNN sweep.

**Evaluation protocol and metrics.** For each train–evaluation configuration pair, we run 10 independent training seeds and periodically evaluate the current policy on both configurations. For each run and metric, we record the *maximum value attained over training*, average these maxima across seeds to obtain per-configuration results, and then average within each suite to produce the axis-level summaries in Table 1. We use the maximum-over-training statistic to assess whether a visual generalization gap is *in principle mitigable* by a given method. Because generalization performance can be non-monotonic and peak at different iterations across runs, this aggregation provides an upper envelope on achievable transfer and avoids confounding results with arbitrary checkpoint selection. Each seed is trained for 25M environment steps and evaluated every 300 PPO iterations on both the training and evaluation

---

[2] https://github.com/vwxyzjn/cleanrl

configurations using 128 evaluation episodes per checkpoint. We use episodic return and success rate as central metrics, while distance and progress provide interpretability for partial progress and incomplete behavior. Progress normalizes distance by `dist_to_success`; success is triggered when the traveled distance reaches `dist_to_success` = 490, which corresponds to near-complete traversal of an approximately 500-distance optimal run and was not tuned to exaggerate gaps. The full benchmark comprises 34 configuration pairs × 10 seeds, requiring about 500 GPU-hours on four A100 GPUs, or roughly 1.5 hours per run including checkpointing and rollout videos savings. Evaluating a new method does not require all 34 pairs: the hard subset used for the RAD/CURL/DrQ/SVEA experiments is sufficient for a cheaper stress test.

**Generalization gap.** We define the visual generalization gap as the performance difference between the training and evaluation configurations of a pair. Figure 4 illustrates three characteristic regimes observed in KAGE-Bench: (i) negligible gap, where train and eval performance coincide; (ii) moderate gap, where partial transfer occurs; and (iii) severe gap, where evaluation performance collapses despite strong training performance. Full learning curves for all 34 configuration pairs and all suites are reported in Appendix D.

## 7. Results

Table 1 reports axis-level results for PPO-CNN under our maximum-over-training protocol: for each seed we take the maximum of each metric over training checkpoints, then average across 10 seeds and finally across configuration pairs within an axis. Figure 6 complements this summary with representative support-expanded evaluations: (left) we train on a black background and evaluate on cumulative background-color supports (black, black+white, black+white+red, black+white+red+green, black+white+red+green+blue); (right) we train with no distractors and evaluate with increasing numbers of same-as-agent distractors (0, 1, 2, 3, 5, 7, 9, 11), where distractors match the agent's shape and color. Across suites, training

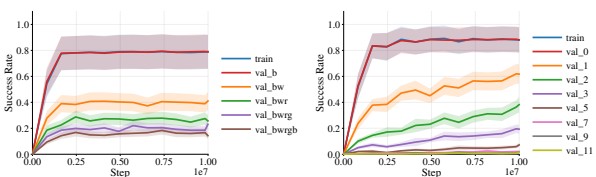

*Figure 6.* **Visual generalization gaps in single-axis shifts.** Each panel shows training success rate (blue) and evaluation on support-expanded visual variants (colored curves). **(Left) Backgrounds:** trained on black background, evaluated with cumulative background-color evaluation supports (black → black+white → black+white+red → etc.). **(Right) Distractors:** trained without distractors, evaluated with increasing numbers of same-as-agent distractors. Full results are presented in the Appendix B, Figure 7.

success rises rapidly, while evaluation success often saturates substantially lower, revealing persistent train–eval gaps under purely visual shifts with fixed dynamics and rewards.

**Generalization is strongly axis-dependent (mean±SEM).** Ranking axes by success-rate degradation, the largest gaps arise from **filters** ($\Delta$SR = 86.8%) and **effects** (80.5%), followed by **layout** (62.8%) and **background** (53.3%); **distractors** (30.9%) and **agent appearance** (21.1%) are comparatively milder (Table 1).

**Background shifts impair both motion and completion.** Averaged across background pairs, distance and progress drop by 30.5% and SR drops from 0.90 to 0.42 ($\Delta$SR = 53.3%), accompanied by a large absolute return gap. In Figure 6 (left), evaluation success decreases as the evaluation support expands from the training background to mixtures that include unseen background colors, while training success on the black background remains high. These curves should be read as support-expansion tests: for example, the black+white+red+green setting evaluates a mixture over those colors, so performance reflects success on the training background and failures on newly included backgrounds. Concretely, support expansion is implemented across evaluation episodes, not by blending backgrounds within an episode: black+white assigns half of evaluation episodes to each color, while black+white+red assigns one third of episodes to each color.

**Photometric and lighting perturbations primarily break completion.** For **filters** and **effects**, distance degradation is moderate ($\Delta$Dist = 11.7% and 20.8%), yet SR collapses ($0.83 \rightarrow 0.11$ and $0.82 \rightarrow 0.16$; $\Delta$SR = 86.8% and 80.5%), indicating that motion and shaped reward can persist while success fails under photometric/lighting shifts.

**Small motion gaps can mask large completion gaps.** **Distractors** and **layout** show small distance/progress gaps ($\sim$3–4%) but sizable SR drops (30.9% and 62.8%). In Figure 6 (right), increasing same-as-agent distractors (0–11) progressively suppresses evaluation success with unchanged training success.

**Per-configuration behavior is heterogeneous.** Table 2 includes both negligible-gap sanity checks and near-failure pairs, e.g., black→noise backgrounds ($\Delta$SR = 98.9%), hue shift 180° (98.8%), light count 4 (95.5%), and 7 same-as-agent distractors (92.0%). Within Background, training with more visual diversity reduces SR gaps. Appendix D provides full learning curves for all 34 pairs; some small return gaps arise because both train and eval fail, motivating joint reporting of distance, progress, and SR. Overall, PPO-CNN is strong in-distribution but brittle under controlled visual shifts, with failures concentrated in task completion rather than basic locomotion. The additional robustness baselines in Appendix C, especially Table 4, support the

*Table 2.* **Per-configuration results for KAGE-Bench (mean±SEM).** Each row corresponds to a train-evaluation configuration pair within a known-axis suite. For each run, we record the maximum value attained by each metric during training; these maxima are then averaged across 10 random seeds. We report Distance, Progress, Success Rate (SR), and Return for both train and eval configurations, together with the resulting generalization gaps. Abbreviations: bg = background, ag = agent, dist = distractor, skelet = skeleton. Generalization gaps are color-coded: green indicates smaller gaps (better generalization), while red indicates larger gaps (worse generalization). The full version of this table with performance across train and eval configurations is presented in the Appendix B, Table 3. The task also admits partial reflexive behavior: a RIGHT+JUMP policy reaches $\text{SR} = 0.59 \pm 0.03$ but has much worse return ($-5000.3 \pm 2.3$) than trained policies, so we interpret success rate together with return rather than treating forward motion alone as task competence.

| | ID | Train config | Eval config | Generalization gap | | | |
| --- | --- | --- | --- | --- | --- | --- | --- |
| | | | | $\Delta$**Dist., %** | $\Delta$**Prog., %** | $\Delta$**SR, %** | $\Delta$**Ret., (abs.)** |
| **Agent** | 1 | teal circle ag | line teal ag | 2.8 | 2.6 | 30.0 | 189.7 |
| | 2 | circle teal ag | circle pink ag | 2.1 | 2.0 | 14.1 | 52.5 |
| | 3 | circle teal ag | line pink ag | 3.1 | 3.6 | 31.3 | 125.3 |
| | 4 | circle teal ag | skelet ag | 3.0 | 2.7 | 21.4 | 84.2 |
| | 5 | skelet ag | clown ag | 1.0 | 1.4 | 8.3 | 128.7 |
| **Background** | 1 | black bg | noise bg | 72.8 | 73.1 | 98.9 | 1867.5 |
| | 2 | black bg | purple bg | 59.6 | 59.8 | 92.2 | 1591.3 |
| | 3 | black bg | purple, lime, indigo bg | 61.2 | 61.3 | 98.9 | 1611.1 |
| | 4 | red, green, blue bg | purple, lime, indigo bg | 2.0 | 2.4 | 18.8 | 50.6 |
| | 5 | black bg | 128 images bg | 50.2 | 50.5 | 93.3 | 1266.6 |
| | 6 | one image bg | another image bg | 1.4 | 2.0 | 9.6 | 170.1 |
| | 7 | 3 images bg | another image bg | 0.0 | 0.0 | -1.3 | 22.7 |
| | 8 | black bg, skelet ag | purple bg, skelet ag | 55.9 | 56.4 | 99.0 | 1463.7 |
| | 9 | one image bg, skelet ag | another image bg, skelet ag | 1.3 | 2.0 | 8.3 | 167.9 |
| | 10 | 3 images bg, skelet ag | another image bg, skelet ag | -0.1 | 0.00 | -1.0 | 9.0 |
| **Distractors** | 1 | no dist., skelet ag | NPC skelets, skelet ag | 0.6 | 0.0 | 1.4 | 14.4 |
| | 2 | no dist., skelet ag | NPC 27 sprites, skelet ag | 0.1 | 0.0 | 0.0 | 0.6 |
| | 3 | no dist., skelet ag | sticky NPC skelets, skelet ag | 5.0 | 5.4 | 31.4 | 176.7 |
| | 4 | no dist., skelet ag | sticky NPC 27 sprites, skelet ag | 1.5 | 1.1 | 14.4 | 48.9 |
| | 5 | no dist., circle teal ag | 7 same-as-ag shapes, circle teal ag | 12.8 | 13.3 | 92.0 | 418.4 |
| | 6 | no dist., circle teal ag | circle indigo dist., circle teal ag | 3.9 | 3.9 | 42.4 | 116.0 |
| **Effects** | 1 | no effects | light intensity 0.5 | 14.5 | 14.1 | 71.4 | 384.6 |
| | 2 | no effects | light fallof 4.0 | 21.6 | 21.7 | 72.5 | 479.2 |
| | 3 | no effects | light count 4 | 25.8 | 25.8 | 95.5 | 638.5 |
| **Filters** | 1 | no filters | brightness 1 | 20.3 | 20.4 | 95.6 | 506.9 |
| | 2 | no filters | contrast 128 | 18.0 | 18.6 | 91.5 | 523.6 |
| | 3 | no filters | saturation 0.0 | 12.6 | 12.8 | 98.0 | 593.3 |
| | 4 | no filters | hue shift 180 | 23.5 | 23.7 | 98.8 | 727.9 |
| | 5 | no filters | color jitter std 2.0 | -3.4 | -3.6 | 91.3 | 283.1 |
| | 6 | no filters | gaussian noise std 100 | 6.4 | 6.5 | 85.6 | 210.3 |
| | 7 | no filters | pixelate factor 3 | 6.6 | 6.6 | 34.3 | 166.8 |
| | 8 | no filters | vinegrette strength 10 | 2.2 | 2.4 | 80.0 | 526.0 |
| | 9 | no filters | radial light strength 1 | 17.1 | 16.7 | 98.3 | 490.6 |
| **Layout** | 1 | cyan layout | red layout | 4.0 | 3.3 | 62.8 | 160.9 |

same conclusion. RAD-PPO with cutout-color improves over PPO-CNN on 7 of the 8 hardest configurations, but substantial success-rate gaps remain. In contrast, RAD-PPO with crop-resize augmentation and off-policy CURL, DrQ, and SVEA did not converge in these platformer runs. Because the non-geometric RAD-PPO cutout-color variant converged as expected, we attribute these failures to spatial augmentations disrupting screen positions needed for cliff distance estimation, obstacle avoidance, and jump timing.

## 8. Limitations

KAGE-Bench is intentionally scoped as a diagnostic benchmark for known-axis visual distribution shifts, not as a comprehensive proxy for real-world RL. The current suite uses a single 2D platformer with fixed dynamics and rewards so that observed performance changes can be cleanly attributed to visual factors. The visual shifts are stylized and factorized rather than photorealistic, and the reported suites were selected using a representative PPO-CNN probe; stronger or differently structured agents may induce different axis rankings. Consequently, the ordering of axes should be interpreted as a property of this controlled benchmark and baseline family, not as a universal ranking of visual robustness difficulty. Future extensions should include geometric and perspective shifts, richer task families, and complementary benchmarks where controlled attribution is traded for broader realism. The environment is simple at the latent-dynamics level by design, but this does not remove all spatial demands: near-complete traversal still requires consistent pixel-to-layout correspondence and accurate jump timing. Adding mechanics such as double jumps, obstacles, or richer objectives is supported by the configurable environment interface, but such additions should be treated as new benchmark variants because they introduce interacting factors beyond isolated visual shift. Continual and curriculum learning are also natural uses of KAGE-Env, but they are outside the scope of the present benchmark.

## 9. Conclusion

We introduced **KAGE-Env**, a JAX-native RL environment for controlled studies of visual generalization that factorizes the observation process into independently configurable visual axes while keeping the underlying control problem fixed, enabling high-throughput evaluation via end-to-end compilation and large-scale parallel simulation. Building on this environment, we presented **KAGE-Bench**, a standardized benchmark comprising six known-axis suites and 34 train–evaluation configuration pairs that isolate specific sources of visual shift and allow precise attribution of performance changes. Empirically, we find that visual generalization difficulty varies substantially across axes: background changes and photometric or lighting perturbations induce the most severe failures, often collapsing task success despite nontrivial progress, whereas agent-appearance shifts are comparatively benign. Overall, KAGE-Bench provides a fast, reproducible, and diagnostic framework for evaluating pixel-based RL under controlled visual variation, and we expect it to support more systematic analysis of visual robustness and future work on richer shifts, broader task families, and alternative learning algorithms.

## Impact Statement

This paper presents work whose goal is to advance the field of machine learning by introducing a fast, reproducible benchmark for studying visual generalization in RL. We do not anticipate immediate negative societal impacts from releasing an evaluation environment and configuration suites; however, as with most progress in robust perception and control, improved generalization methods could enable more capable autonomous systems, which may have downstream applications with safety and misuse considerations. We hope KAGE-Env and KAGE-Bench support more rigorous and transparent evaluation of robustness, helping the community identify failure modes early and develop safer learning systems.

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

## A. Reducing Visual Shifts to State-Policy Shifts

### A.1. Problem setup.

KAGE-Bench is constructed to isolate *purely visual* distribution shift. Formally, each environment instance is indexed by a visual configuration $\xi \in \Xi$ (e.g., the YAML parameters controlling background, filters, lighting, sprites), and $\xi$ determines how a latent simulator state $s \in S$ is rendered into a pixel observation $o \in \Omega$. This rendering mechanism is modeled as an *observation kernel* $O_\xi(\cdot \mid s)$, meaning that, given the same latent state $s$, different $\xi$ may produce different distributions over images. Crucially, KAGE-Bench enforces that $\xi$ does *not* alter the control problem itself: the transition kernel $P(\cdot \mid s, a)$ and reward function $r(s, a)$ are identical for all $\xi$. Hence, when we observe a train–test gap after changing $\xi$, it cannot be caused by different dynamics or rewards; it must be caused by the interaction between the *same* observation-based policy and a different rendering process.

The key point is that a policy trained on pixels, $\pi(a \mid o)$, does not directly specify actions as a function of the latent state $s$, but only as a function of the rendered image $o$. Therefore, the action distribution *conditioned on the latent state* depends on $\xi$ through the distribution of renderings $O_\xi(\cdot \mid s)$. The definition below formalizes this dependence by defining, for each $\xi$, an *induced state policy* (Definition A.1):

**Definition A.1** (Induced State Policy). Given a visual configuration $\xi \in \Xi$, observation kernel $O_\xi(\cdot \mid s)$, and pixel policy $\pi(\cdot \mid o)$, the **induced state policy** $\pi_\xi : S \times \mathcal{A} \to [0, 1]$ is defined as:

$$\pi_\xi(B \mid s) := \int_\Omega \pi(B \mid o) \, O_\xi(do \mid s), \quad \forall B \in \mathcal{A}, \forall s \in S. \tag{11}$$

where $B$ is any measurable subset of the action space (i.e., $B \in \mathcal{A}$, the $\sigma$-algebra of measurable action sets). This represents the conditional distribution over actions given latent state $s$, obtained by marginalizing over the intermediate observation variable $o$.

---

**Takeaway:** Let's fix a latent state $s$. The environment may render multiple different images $o$ due to background choices, filters, effects, etc., depending on the visual configuration $\xi$. The policy $\pi$ maps each $o$ to an action distribution. $\pi_\xi(\cdot|s)$ is the mixture of those action distributions weighted by how likely each $o$ is under $O_\xi(\cdot|s)$. Thus $\pi_\xi(\cdot \mid s)$ is the *effective action distribution* at latent state $s$ induced by the pair (renderer $O_\xi$, pixel policy $\pi$).

---

Induced state policy is the conditional distribution of the action after integrating out (marginalizing) the intermediate observation variable $o$. In this sense, *changing $\xi$ is equivalent to changing the induced state policy*: even if the pixel policy $\pi(a \mid o)$ is fixed, the effective mapping from latent states to action distributions changes because the policy is evaluated on different renderings. This reduction is fundamental for analysis: it converts visual generalization under observation shifts into a standard *policy shift problem in a fixed latent MDP*.

### A.2. Setting and central objects

We work with the following measurable objects.

- **Measurable spaces.** $(S, \mathcal{S})$ is the latent state space equipped with a $\sigma$-algebra $\mathcal{S}$ of measurable subsets; $(A, \mathcal{A})$ is the action space equipped with $\sigma$-algebra $\mathcal{A}$; $(\Omega, \mathcal{O})$ is the observation (pixel). Measurability ensures that probabilities and integrals used below are well-defined.

- **MDP primitives.** $\gamma \in [0, 1)$ is the discount factor and $\rho_0$ is the initial distribution on $(S, \mathcal{S})$.

- **Transition kernel (Markov kernel).** $P(\cdot \mid s, a)$ specifies the environment dynamics. For every state–action pair $(s, a) \in S \times A$, $P(\cdot \mid s, a)$ is a probability distribution over next states in $S$. Operationally, this means that after taking action $a_t$ in state $s_t$, the next state is sampled as $s_{t+1} \sim P(\cdot \mid s_t, a_t)$.

- **Reward function.** $r : S \times A \to \mathbb{R}$ is measurable and bounded: $\|r\|_\infty := \sup_{(s,a) \in S \times A} |r(s, a)| < \infty$. Boundedness guarantees the discounted return $\sum_{t \geq 0} \gamma^t r(s_t, a_t)$ is integrable.

- **Visual configuration space.** $\Xi$ indexes renderers. For each $\xi \in \Xi$, $O_\xi(\cdot \mid s)$ is an observation kernel (a Markov kernel from $(S, \mathcal{S})$ to $(\Omega, \mathcal{O})$). Operationally, given latent state $s$, an image is sampled as $o \sim O_\xi(\cdot \mid s)$.

- **Reactive pixel policy.** $\pi(\cdot \mid o)$ is a Markov kernel from $(\Omega, \mathcal{O})$ to $(A, \mathcal{A})$ (memoryless policy): given observation $o$, an action is sampled as $a \sim \pi(\cdot \mid o)$.

**Latent MDP and visual POMDP.**

**Definition A.2** (Latent MDP). The **latent MDP** is the underlying control problem defined as:

$$\mathcal{M} := (S, A, P, r, \rho_0, \gamma).$$

This represents the true decision process with latent states $S$, actions $A$, transition kernel $P$, reward function $r$, initial distribution $\rho_0$, and discount factor $\gamma$.

**Definition A.3** (Visual POMDP). For each visual configuration $\xi \in \Xi$, define the **visual POMDP** as:

$$\mathcal{M}_\xi := (S, A, P, r, \Omega, O_\xi, \rho_0, \gamma).$$

By construction, $\xi$ affects *only* the observation kernel $O_\xi$; in particular, the transition kernel $P$ and reward function $r$ are invariant across all $\xi \in \Xi$.

### A.3. Main theorem.

**Theorem A.4** (Visual shift reduces to state-policy shift by marginalization). *Fix any $\xi \in \Xi$ and any reactive pixel policy $\pi(\cdot \mid o)$. Let $\pi_\xi$ be defined by Definition A.1. Then:*

1. *(**Conditional action law.**) For every time $t \geq 0$ and every measurable action set $B \in \mathcal{A}$,*

$$\mathbb{P}_{\mathcal{M}_\xi, \pi}(a_t \in B \mid s_t) \;=\; \pi_\xi(B \mid s_t) \quad a.s. \tag{12}$$

   *That is, after conditioning on the latent state, the intermediate observation variable can be integrated out and the resulting action distribution is exactly $\pi_\xi(\cdot \mid s_t)$.*

2. *(**Equality in law of state–action processes.**) The state–action process $(s_t, a_t)_{t \geq 0}$ induced by executing $\pi$ in $\mathcal{M}_\xi$ has the same law as the state–action process induced by executing $\pi_\xi$ in the latent MDP $\mathcal{M}$.*

3. *(**Return equivalence.**) Consequently, the expected discounted return is preserved:*

$$J(\pi; \mathcal{M}_\xi) \;=\; J(\pi_\xi; \mathcal{M}), \qquad J(\pi; \mathcal{M}_\xi) := \mathbb{E}_{\mathcal{M}_\xi, \pi}\Big[\sum_{t=0}^{\infty} \gamma^t r(s_t, a_t)\Big]. \tag{13}$$

### A.4. Proof of Theorem A.4

**Step 0 (Generative dynamics in $\mathcal{M}_\xi$).** By definition of the POMDP $\mathcal{M}_\xi$ (Definition A.3) and the reactive policy $\pi$, the interaction at time $t$ is:

$$\begin{cases} o_t \sim O_\xi(\cdot \mid s_t) \\ a_t \sim \pi(\cdot \mid o_t) \\ s_{t+1} \sim P(\cdot \mid s_t, a_t) \end{cases} \tag{14}$$

Equation (14) (top) represents the rendering step: it formalizes that pixels are generated from the latent state via $O_\xi$. Equation (14) (middle) is the policy step: the agent samples an action using only the pixels. Equation (14) (bottom) is the environment dynamics: the next state depends only on $(s_t, a_t)$ through $P$ and is independent of $o_t$ given $(s_t, a_t)$. Therefore, observations influence the future only through their effect on the chosen action.

**Step 1 (Show conditional action law (12)).** Fix a time $t \geq 0$ and an arbitrary measurable set $B \in \mathcal{A}$. We compute $\mathbb{P}(a_t \in B \mid s_t)$ by conditioning on the intermediate variable $o_t$ (the observation).

**(1a) Law of total probability (tower property).** Recall the tower property of conditional expectation (Weiss et al., 2005): for any integrable random variable $X$ and $\sigma$-algebras $\mathcal{G}_1 \subseteq \mathcal{G}_2$,

$$\mathbb{E}[X \mid \mathcal{G}_1] = \mathbb{E}[\mathbb{E}[X \mid \mathcal{G}_2] \mid \mathcal{G}_1]. \tag{15}$$

This identity states that conditioning can be performed in stages: one may first condition on a finer information set $\mathcal{G}_2$ and then average again while conditioning on the coarser information set $\mathcal{G}_1$.

We apply (15) to the indicator random variable

$$X := \mathbb{I}\{a_t \in B\},$$

which is integrable since it is bounded between $0$ and $1$. Recall that conditional probabilities can be written as conditional expectations of indicator functions:

$$\mathbb{P}(a_t \in B \mid \mathcal{G}) = \mathbb{E}[\mathbb{I}\{a_t \in B\} \mid \mathcal{G}].$$

Next, we specify the two $\sigma$-algebras:

- $\mathcal{G}_1 := \sigma(s_t)$, the $\sigma$-algebra generated by the latent state $s_t$ (i.e., conditioning on knowing $s_t$),

- $\mathcal{G}_2 := \sigma(o_t, s_t)$, the $\sigma$-algebra generated by the pair $(o_t, s_t)$ (i.e., conditioning on knowing both the observation and the latent state).

Clearly, $\mathcal{G}_1 \subseteq \mathcal{G}_2$, since knowing $(o_t, s_t)$ includes knowing $s_t$.

Applying (15) with these choices gives

$$\begin{aligned}
\mathbb{P}_{\mathcal{M}_\xi, \pi}(a_t \in B \mid s_t) &= \mathbb{E}_{\mathcal{M}_\xi, \pi}[\mathbb{I}\{a_t \in B\} \mid s_t] \\
&= \mathbb{E}_{\mathcal{M}_\xi, \pi}\big[\mathbb{E}_{\mathcal{M}_\xi, \pi}[\mathbb{I}\{a_t \in B\} \mid o_t, s_t] \,\big|\, s_t\big].
\end{aligned}$$

Finally, rewriting the inner conditional expectation again as a conditional probability yields

$$\mathbb{P}_{\mathcal{M}_\xi, \pi}(a_t \in B \mid s_t) = \mathbb{E}_{\mathcal{M}_\xi, \pi}\big[\mathbb{P}_{\mathcal{M}_\xi, \pi}(a_t \in B \mid o_t, s_t) \,\big|\, s_t\big]. \tag{16}$$

This equality formalizes the intuitive idea that, to compute the probability of choosing an action in $B$ given the latent state $s_t$, one may first compute this probability given the more detailed information $(o_t, s_t)$ and then average over all possible observations $o_t$ that can occur when the state is $s_t$.

**(1b) Use the policy sampling rule.** Recall from the interaction dynamics that, at time $t$, once the observation $o_t$ is generated, the action is sampled according to the policy:

$$a_t \sim \pi(\cdot \mid o_t).$$

This means that the conditional distribution of $a_t$ given $o_t$ is exactly $\pi(\cdot \mid o_t)$.

Formally, for any measurable action set $B \in \mathcal{A}$,

$$\mathbb{P}_{\mathcal{M}_\xi, \pi}(a_t \in B \mid o_t) = \pi(B \mid o_t).$$

Moreover, because the policy is *reactive* (memoryless), the action depends on the current observation $o_t$ but not directly on the latent state $s_t$ once $o_t$ is known. Therefore, conditioning additionally on $s_t$ does not change the conditional distribution:

$$\mathbb{P}_{\mathcal{M}_\xi, \pi}(a_t \in B \mid o_t, s_t) = \mathbb{P}_{\mathcal{M}_\xi, \pi}(a_t \in B \mid o_t) = \pi(B \mid o_t). \tag{17}$$

This equality expresses the fact that the policy fully mediates the influence of the observation on the action, and no additional information about $s_t$ is used once $o_t$ has been observed.

**(1c) Substitute** (17) **into** (16).

$$\mathbb{P}_{\mathcal{M}_\xi, \pi}(a_t \in B \mid s_t) = \mathbb{E}_{\mathcal{M}_\xi, \pi}[\pi(B \mid o_t) \mid s_t]. \tag{18}$$

At this point, the only remaining randomness inside the conditional expectation comes from $o_t$ given $s_t$.

**(1d) Use the observation sampling rule.** Since $o_t \mid s_t \sim O_\xi(\cdot \mid s_t)$ by (14), the conditional expectation in (18) can be written as an integral with respect to the measure $O_\xi(\cdot \mid s_t)$:

$$\mathbb{E}_{\mathcal{M}_\xi, \pi}[\pi(B \mid o_t) \mid s_t] = \int_\Omega \pi(B \mid o)\, O_\xi(do \mid s_t). \tag{19}$$

This step is precisely what "averaging over renderings" means: we are averaging the policy's probability of selecting an action in $B$ over all images $o$ that can be rendered from $s_t$ under configuration $\xi$.

**(1d) Use the observation sampling rule.** At this point, the random quantity inside the conditional expectation in (18) is $\pi(B \mid o_t)$, and the only remaining source of randomness is the observation $o_t$ given the latent state $s_t$. By the generative dynamics of the visual POMDP (14), the observation at time $t$ is sampled according to the observation kernel: $o_t \mid s_t \sim O_\xi(\cdot \mid s_t)$. Therefore, conditioning on $s_t$, the random variable $\pi(B \mid o_t)$ is distributed according to the pushforward of $O_\xi(\cdot \mid s_t)$ through the function $o \mapsto \pi(B \mid o)$.

By the definition of conditional expectation with respect to a Markov kernel, this conditional expectation can be written as an integral over the observation space:

$$\mathbb{E}_{\mathcal{M}_\xi, \pi}[\pi(B \mid o_t) \mid s_t] = \int_\Omega \pi(B \mid o)\, O_\xi(do \mid s_t). \tag{20}$$

This expression makes explicit what is meant by "averaging over renderings": for a fixed latent state $s_t$, we take all images $o$ that the renderer may produce under configuration $\xi$, weight the policy's action probability $\pi(B \mid o)$ by how likely each image is under $O_\xi(\cdot \mid s_t)$, and sum (integrate) these contributions. The result is the average probability of selecting an action in $B$ after accounting for all possible renderings of the same latent state.

**(1e) Recognize the induced policy definition.** By Definition A.1, the right-hand side of (20) equals $\pi_\xi(B \mid s_t)$. Therefore,

$$\mathbb{P}_{\mathcal{M}_\xi, \pi}(a_t \in B \mid s_t) = \pi_\xi(B \mid s_t),$$

which is exactly (12). This completes Item 1.

**Step 2 (Equality in law of state–action processes.).** We now show that the state–action process in $\mathcal{M}_\xi$ under $\pi$ evolves exactly as in the latent MDP $\mathcal{M}$ under $\pi_\xi$.

**(2a) Effective action selection given $s_t$.** Item 1 implies that, conditional on $s_t$, the action $a_t$ has distribution $\pi_\xi(\cdot \mid s_t)$. Hence, if we are interested only in the joint process $(s_t, a_t)$ (and not in $o_t$), we may replace the two-step procedure

$$o_t \sim O_\xi(\cdot \mid s_t), \qquad a_t \sim \pi(\cdot \mid o_t)$$

by the single step

$$a_t \sim \pi_\xi(\cdot \mid s_t),$$

without changing the conditional distribution of $a_t$ given $s_t$.

**(2b) State transition given $(s_t, a_t)$ is identical.** Under $\mathcal{M}_\xi$, the next state satisfies $s_{t+1} \sim P(\cdot \mid s_t, a_t)$ by (14). This is exactly the same transition rule as in the latent MDP $\mathcal{M}$, and it depends only on $(s_t, a_t)$.

**(2c) Conclude identical recursion.** Combining (2a) and (2b), the pair $(s_t, a_t)$ evolves according to

$$s_0 \sim \rho_0, \qquad a_t \sim \pi_\xi(\cdot \mid s_t), \qquad s_{t+1} \sim P(\cdot \mid s_t, a_t).$$

This is precisely the generative definition of executing the state policy $\pi_\xi$ in the latent MDP $\mathcal{M}$. Therefore, the joint laws of $(s_0, a_0, s_1, a_1, \dots)$ coincide under $(\mathcal{M}_\xi, \pi)$ and $(\mathcal{M}, \pi_\xi)$, proving Item 2.

**Step 3 (Equality of expected discounted return).** Define the discounted return random variable

$$G := \sum_{t=0}^{\infty} \gamma^t r(s_t, a_t).$$

We first verify that $G$ is integrable. By assumption, the reward function $r$ is bounded, meaning that for all $(s, a) \in S \times A$,

$$|r(s, a)| \leq \|r\|_\infty < \infty.$$

Therefore, for every time step $t$,

$$|\gamma^t r(s_t, a_t)| \leq \gamma^t \|r\|_\infty.$$

Summing these bounds over $t$ and using that $\gamma \in [0, 1)$ yields

$$|G| = \left| \sum_{t=0}^{\infty} \gamma^t r(s_t, a_t) \right| \leq \sum_{t=0}^{\infty} \gamma^t |r(s_t, a_t)| \leq \sum_{t=0}^{\infty} \gamma^t \|r\|_\infty.$$

The right-hand side is a convergent geometric series:

$$\sum_{t=0}^{\infty} \gamma^t \|r\|_\infty = \|r\|_\infty \sum_{t=0}^{\infty} \gamma^t = \frac{\|r\|_\infty}{1 - \gamma} < \infty.$$

Hence, $G$ is almost surely finite and integrable.

By Item 2, the state–action processes $(s_t, a_t)_{t \geq 0}$ have the same law under $(\mathcal{M}_\xi, \pi)$ and $(\mathcal{M}, \pi_\xi)$. Since $G$ is a measurable function of the entire state–action trajectory and depends only on $(s_t, a_t)$, it follows that $G$ has the same distribution under both constructions. In particular, their expectations coincide:

$$J(\pi; \mathcal{M}_\xi) = \mathbb{E}_{\mathcal{M}_\xi, \pi}[G] = \mathbb{E}_{\mathcal{M}, \pi_\xi}[G] = J(\pi_\xi; \mathcal{M}).$$

This proves Item 3 and completes the proof. $\qquad\square$

### A.5. Interpretation for KAGE-Bench

Theorem A.4 provides a precise formal justification for how visual generalization should be interpreted in KAGE-Bench. Because the latent dynamics $P$ and reward function $r$ are identical across all visual configurations $\xi$, the theorem shows that changing $\xi$ affects the learning problem only through the observation channel $O_\xi(\cdot \mid s)$. For any fixed pixel policy $\pi(a \mid o)$, this change manifests exclusively as a change in the induced state-conditional action distribution $\pi_\xi(\cdot \mid s)$.

Crucially, the theorem establishes an *exact equivalence in distribution* at the level of latent state–action trajectories: executing the observation-based policy $\pi$ in the visual POMDP $\mathcal{M}_\xi$ produces the same joint law over $(s_t, a_t)$ as executing the induced state policy $\pi_\xi$ in the latent MDP $\mathcal{M}$. This result is purely representational. It does not claim that $\pi_\xi$ is optimal, nor that marginalizing over observations improves performance. Rather, it shows that all effects of visual variation are captured entirely by the induced policy, without altering the underlying control problem.

This equivalence is central to the design and interpretation of KAGE-Bench. It guarantees that any observed train–test performance gap under a visual shift $\xi \to \xi'$ cannot be attributed to changes in dynamics, rewards, or task structure, but must correspond exactly to a performance difference between two state policies $\pi_\xi$ and $\pi_{\xi'}$ acting in the same latent MDP. As a consequence, KAGE-Bench reduces visual generalization to a well-defined policy shift problem in a fixed MDP, enabling principled analysis using standard reinforcement learning tools and ensuring that benchmark results isolate perception-induced failures rather than confounding control effects.

## A.6. Additional consequences: equivalence of trajectory-level metrics

Theorem A.4 implies more than equality of expected return. Because it establishes equality *in distribution* of the latent state–action process $(s_t, a_t)_{t \geq 0}$, any performance metric that is a measurable function of the latent trajectory inherits the same equivalence. We formalize this as a corollary.

**Corollary A.5** (Equivalence of trajectory-level evaluation metrics). *Fix any visual configuration $\xi \in \Xi$ and reactive pixel policy $\pi(\cdot \mid o)$, and let $\pi_\xi$ be the induced state policy. Let*

$$(s_t, a_t)_{t \geq 0} \sim (\mathcal{M}_\xi, \pi) \quad and \quad (\tilde{s}_t, \tilde{a}_t)_{t \geq 0} \sim (\mathcal{M}, \pi_\xi).$$

*Then for any measurable functional*

$$F : (S \times A)^{\mathbb{N}} \to \mathbb{R},$$

*it holds that*

$$F\big((s_t, a_t)_{t \geq 0}\big) \stackrel{d}{=} F\big((\tilde{s}_t, \tilde{a}_t)_{t \geq 0}\big),$$

*and in particular*

$$\mathbb{E}_{\mathcal{M}_\xi, \pi}[F] = \mathbb{E}_{\mathcal{M}, \pi_\xi}[F],$$

*whenever the expectation is well-defined.*

**Proof.** By Item 2 of Theorem A.4, the joint laws of the state–action trajectories coincide:

$$\mathcal{L}_{\mathcal{M}_\xi, \pi}\big((s_t, a_t)_{t \geq 0}\big) = \mathcal{L}_{\mathcal{M}, \pi_\xi}\big((\tilde{s}_t, \tilde{a}_t)_{t \geq 0}\big).$$

Applying any measurable function $F$ to two random elements with the same law yields random variables with the same law. Equality of expectations follows immediately. $\square$

We now specialize Corollary A.5 to the concrete evaluation metrics used in KAGE-Bench.

**Corollary A.6** (Equivalence of distance, progress, and success metrics). *Assume the latent state $s_t$ contains a one-dimensional position variable $x_t \in \mathbb{R}$, with initial position $x_{\mathrm{init}}$, and let $D > 0$ denote the task completion threshold (e.g., $D = 490$ in KAGE-Bench). Define the following trajectory-level metrics:*

- *Passed distance:*

$$F_{\mathrm{dist}} := x_T - x_{\mathrm{init}},$$

  *for a fixed horizon $T$ or terminal time.*

- *Normalized progress:*

$$F_{\mathrm{prog}} := \frac{x_T - x_{\mathrm{init}}}{D}.$$

- *Success indicator:*

$$F_{\mathrm{succ}} := \mathbb{I}\{x_T - x_{\mathrm{init}} \geq D\}.$$

*Then, for each of these metrics,*

$$\mathbb{E}_{\mathcal{M}_\xi, \pi}[F] = \mathbb{E}_{\mathcal{M}, \pi_\xi}[F], \quad F \in \{F_{\mathrm{dist}}, F_{\mathrm{prog}}, F_{\mathrm{succ}}\},$$

*and moreover each metric has the same distribution under $(\mathcal{M}_\xi, \pi)$ and $(\mathcal{M}, \pi_\xi)$.*

**Interpretation.** Corollary A.6 shows that the equivalence established in Theorem A.4 applies not only to discounted return, but also to all trajectory-based evaluation metrics commonly reported in KAGE-Bench, including raw distance traveled, normalized progress, and binary success. These quantities depend only on the latent state trajectory $(s_t)_{t \geq 0}$ and are therefore fully determined by the induced state policy $\pi_\xi$ in the latent MDP.

As a result, differences in success rate, progress, or distance under a visual shift $\xi \to \xi'$ are *exactly* differences between the induced state policies $\pi_\xi$ and $\pi_{\xi'}$ acting in the same latent MDP. This further reinforces that KAGE-Bench isolates perception-induced failures: all reported metrics admit a clean interpretation as properties of state-policy shift rather than changes in the underlying control task.

# B. Extended figures and tables

Figure 7 compares training success rate with evaluation success rate under increasingly broad variants of a single visual axis. Table 3 shows train and eval results across each reported metric across each config.

*Table 3.* **Per-configuration results for KAGE-Bench (mean±SEM).** Each row corresponds to a train-evaluation configuration pair within a known-axis suite. For each run, we record the maximum value attained by each metric during training; these maxima are then averaged across 10 random seeds. We report Distance, Progress, Success Rate (SR), and Return for both train and eval configurations, together with the resulting generalization gaps. Abbreviations: `bg` = background, `ag` = agent, `dist` = distractor, `skelet` = skeleton. Generalization gaps are color-coded: green indicates smaller gaps (better generalization), while red indicates larger gaps (worse generalization).

| | ID | Train config | Eval config | Evaluation on train config | | | | Evaluation on eval config | | | | Generalization gap | | | |
|---|---|---|---|---|---|---|---|---|---|---|---|---|---|---|---|
| | | | | Distance | Progress | SR | Return | Distance | Progress | SR | Return | ΔDist., % | ΔProg., % | ΔSR, % | ΔRet., (abs.) |
| **Agent** | 1 | teal circle ag | line teal ag | 372.7±64.2 | 0.76±0.13 | 0.70±0.15 | -306.1±151.3 | 362.4±61.9 | 0.74±0.13 | 0.49±0.11 | -495.8±158.2 | 2.8 | 2.6 | 30.0 | 189.7 |
| | 2 | circle teal ag | circle pink ag | 495.6±3.2 | 1.01±0.01 | 0.99±0.01 | -35.8±26.6 | 485.0±8.0 | 0.99±0.02 | 0.85±0.07 | -88.3±42.4 | 2.1 | 2.0 | 14.1 | 52.5 |
| | 3 | circle teal ag | line pink ag | 406.9±61.5 | 0.83±0.13 | 0.80±0.13 | -225.3±144.2 | 394.1±60.0 | 0.80±0.12 | 0.55±0.10 | -350.6±147.3 | 3.1 | 3.6 | 31.3 | 125.3 |
| | 4 | circle teal ag | skelet ag | 363.8±69.0 | 0.74±0.14 | 0.70±0.15 | -454.8±243.6 | 353.0±67.5 | 0.72±0.14 | 0.55±0.12 | -539.0±237.8 | 3.0 | 2.7 | 21.4 | 84.2 |
| | 5 | skelet ag | clown ag | 343.6±64.4 | 0.70±0.13 | 0.60±0.16 | -441.8±219.2 | 340.1±63.1 | 0.69±0.13 | 0.55±0.15 | -570.5±230.0 | 1.0 | 1.4 | 8.3 | 128.7 |
| **Background** | 1 | black bg | noise bg | 455.5±43.3 | 0.93±0.09 | 0.90±0.10 | -111.2±102.5 | 123.9±29.7 | 0.25±0.06 | 0.04±0.00 | -1978.7±111.3 | 72.8 | 73.1 | 98.9 | 1867.5 |
| | 2 | black bg | purple bg | 452.7±46.1 | 0.92±0.09 | 0.90±0.10 | -116.8±107.6 | 182.9±52.4 | 0.37±0.11 | 0.07±0.06 | -1708.1±208.8 | 59.6 | 59.8 | 92.2 | 1591.3 |
| | 3 | black bg | purple, lime, indigo bg | 456.6±42.2 | 0.93±0.09 | 0.90±0.10 | -104.2±94.7 | 177.1±40.0 | 0.36±0.08 | 0.01±0.00 | -1715.3±164.1 | 61.2 | 61.3 | 98.9 | 1611.1 |
| | 4 | red, green, blue bg | purple, lime, indigo bg | 415.3±55.9 | 0.85±0.11 | 0.80±0.13 | -290.1±186.6 | 406.8±54.8 | 0.83±0.11 | 0.65±0.12 | -340.7±181.0 | 2.0 | 2.4 | 18.8 | 50.6 |
| | 5 | black bg | 128 images bg | 455.7±43.2 | 0.93±0.09 | 0.90±0.10 | -187.5±178.6 | 226.8±38.9 | 0.46±0.08 | 0.06±0.02 | -1454.1±175.4 | 50.2 | 50.5 | 93.3 | 1266.6 |
| | 6 | one image bg | another image bg | 497.7±1.1 | 1.02±0.00 | 0.94±0.05 | -28.2±16.6 | 490.5±2.2 | 1.00±0.00 | 0.85±0.04 | -198.2±30.8 | 1.4 | 2.0 | 9.6 | 170.1 |
| | 7 | 3 images bg | another image bg | 411.9±57.2 | 0.84±0.12 | 0.77±0.13 | -277.7±170.7 | 411.8±57.6 | 0.84±0.12 | 0.78±0.13 | -255.0±150.0 | 0.0 | 0.0 | -1.3 | 22.7 |
| | 8 | black bg, skelet ag | purple bg, skelet ag | 493.0±5.8 | 1.01±0.01 | 0.95±0.05 | -25.7±16.3 | 217.5±52.5 | 0.44±0.11 | 0.01±0.00 | -1489.4±244.7 | 55.9 | 56.4 | 99.0 | 1463.7 |
| | 9 | one image bg, skelet ag | another image bg, skelet ag | 498.0±0.8 | 1.02±0.00 | 0.97±0.03 | -15.8±7.1 | 491.6±1.9 | 1.00±0.00 | 0.89±0.03 | -183.6±32.6 | 1.3 | 2.0 | 8.3 | 167.9 |
| | 10 | 3 images bg, skelet ag | another image bg, skelet ag | 498.0±0.6 | 1.02±0.00 | 0.97±0.02 | -25.7±14.6 | 498.4±0.4 | 1.02±0.00 | 0.98±0.02 | -34.7±20.7 | -0.1 | 0.00 | -1.0 | 9.0 |
| **Distractors** | 1 | no dist., skelet ag | NPC skelets, skelet ag | 352.6±74.5 | 0.72±0.15 | 0.70±0.15 | -261.9±129.0 | 350.5±75.1 | 0.72±0.15 | 0.69±0.15 | -276.2±130.0 | 0.6 | 0.0 | 1.4 | 14.4 |
| | 2 | no dist., skelet ag | NPC 27 sprites, skelet ag | 407.0±61.4 | 0.83±0.13 | 0.80±0.13 | -219.2±143.6 | 406.7±61.3 | 0.83±0.13 | 0.80±0.13 | -218.6±138.7 | 0.1 | 0.0 | 0.0 | 0.6 |
| | 3 | no dist., skelet ag | sticky NPC skelets, skelet ag | 360.5±70.7 | 0.74±0.14 | 0.70±0.15 | -274.4±135.4 | 342.3±68.0 | 0.70±0.14 | 0.48±0.11 | -451.1±149.4 | 5.0 | 5.4 | 31.4 | 176.7 |
| | 4 | no dist., skelet ag | sticky NPC 27 sprites, skelet ag | 456.6±42.3 | 0.93±0.09 | 0.90±0.10 | -97.6±88.3 | 449.7±41.6 | 0.92±0.08 | 0.77±0.09 | -146.5±82.5 | 1.5 | 1.1 | 14.4 | 48.9 |
| | 5 | no dist., circle teal ag | 7 same-as-ag shapes, circle teal ag | 405.4±60.9 | 0.83±0.12 | 0.75±0.13 | -199.9±117.2 | 353.6±53.6 | 0.72±0.11 | 0.06±0.01 | -618.3±156.8 | 12.8 | 13.3 | 92.0 | 418.4 |
| | 6 | no dist., circle teal ag | circle indigo dist., circle teal ag | 498.5±0.3 | 1.02±0.00 | 0.99±0.01 | -14.9±6.0 | 479.2±2.8 | 0.98±0.01 | 0.57±0.04 | -131.0±17.9 | 3.9 | 3.9 | 42.4 | 116.0 |
| **Effects** | 1 | no effects | light intensity 0.5 | 416.1±54.7 | 0.85±0.11 | 0.77±0.15 | -214.3±127.5 | 355.6±46.2 | 0.73±0.09 | 0.22±0.04 | -598.9±117.5 | 14.5 | 14.1 | 71.4 | 384.6 |
| | 2 | no effects | light fallof 4.0 | 406.1±62.1 | 0.83±0.13 | 0.80±0.13 | -339.9±221.8 | 318.4±46.8 | 0.65±0.10 | 0.22±0.04 | -819.1±155.8 | 21.6 | 21.7 | 72.5 | 479.2 |
| | 3 | no effects | light count 4 | 456.9±41.5 | 0.93±0.08 | 0.89±0.10 | -118.8±101.8 | 339.0±30.4 | 0.69±0.06 | 0.04±0.01 | -757.2±50.8 | 25.8 | 25.8 | 95.5 | 638.5 |
| **Filters** | 1 | no filters | brightness 1 | 457.1±41.7 | 0.93±0.09 | 0.90±0.10 | -125.0±115.2 | 364.3±33.9 | 0.74±0.07 | 0.04±0.01 | -631.8±58.7 | 20.3 | 20.4 | 95.6 | 506.9 |
| | 2 | no filters | contrast 128 | 497.6±1.3 | 1.02±0.00 | 0.94±0.06 | -23.1±13.0 | 408.0±12.5 | 0.83±0.03 | 0.08±0.01 | -546.7±50.7 | 18.0 | 18.6 | 91.5 | 523.6 |
| | 3 | no filters | saturation 0.0 | 498.9±0.1 | 1.02±0.00 | 1.00±0.00 | -9.3±0.5 | 435.8±4.8 | 0.89±0.01 | 0.02±0.01 | -602.6±30.3 | 12.6 | 12.8 | 98.0 | 593.3 |
| | 4 | no filters | hue shift 180 | 453.3±44.0 | 0.93±0.09 | 0.86±0.10 | -128.2±104.3 | 346.9±34.4 | 0.71±0.07 | 0.01±0.00 | -856.1±95.5 | 23.5 | 23.7 | 98.8 | 727.9 |
| | 5 | no filters | color jitter std 2.0 | 407.2±61.5 | 0.83±0.13 | 0.80±0.13 | -216.9±138.9 | 421.1±29.0 | 0.86±0.06 | 0.07±0.01 | -500.0±77.7 | -3.4 | -3.6 | 91.3 | 283.1 |
| | 6 | no filters | gaussian noise std 100 | 455.8±43.1 | 0.93±0.09 | 0.90±0.10 | -158.6±149.3 | 426.8±40.5 | 0.87±0.08 | 0.13±0.03 | -368.9±139.0 | 6.4 | 6.5 | 85.6 | 210.3 |
| | 7 | no filters | pixelate factor 3 | 371.5±64.1 | 0.76±0.13 | 0.67±0.15 | -371.7±178.6 | 346.9±59.4 | 0.71±0.12 | 0.44±0.10 | -538.5±149.5 | 6.6 | 6.6 | 34.3 | 166.8 |
| | 8 | no filters | vinegrette strength 10 | 416.1±55.3 | 0.85±0.11 | 0.80±0.13 | -229.7±146.8 | 407.2±44.9 | 0.83±0.09 | 0.16±0.04 | -755.7±91.1 | 2.2 | 2.4 | 80.0 | 526.0 |
| | 9 | no filters | radial light strength 1 | 323.2±71.9 | 0.66±0.15 | 0.60±0.16 | -580.6±257.6 | 268.0±58.6 | 0.55±0.12 | 0.01±0.00 | -1071.2±216.5 | 17.1 | 16.7 | 98.3 | 490.6 |
| **Layout** | 1 | cyan layout | red layout | 452.3±42.0 | 0.92±0.09 | 0.86±0.10 | -118.6±89.1 | 434.1±39.3 | 0.89±0.08 | 0.32±0.08 | -279.5±78.2 | 4.0 | 3.3 | 62.8 | 160.9 |

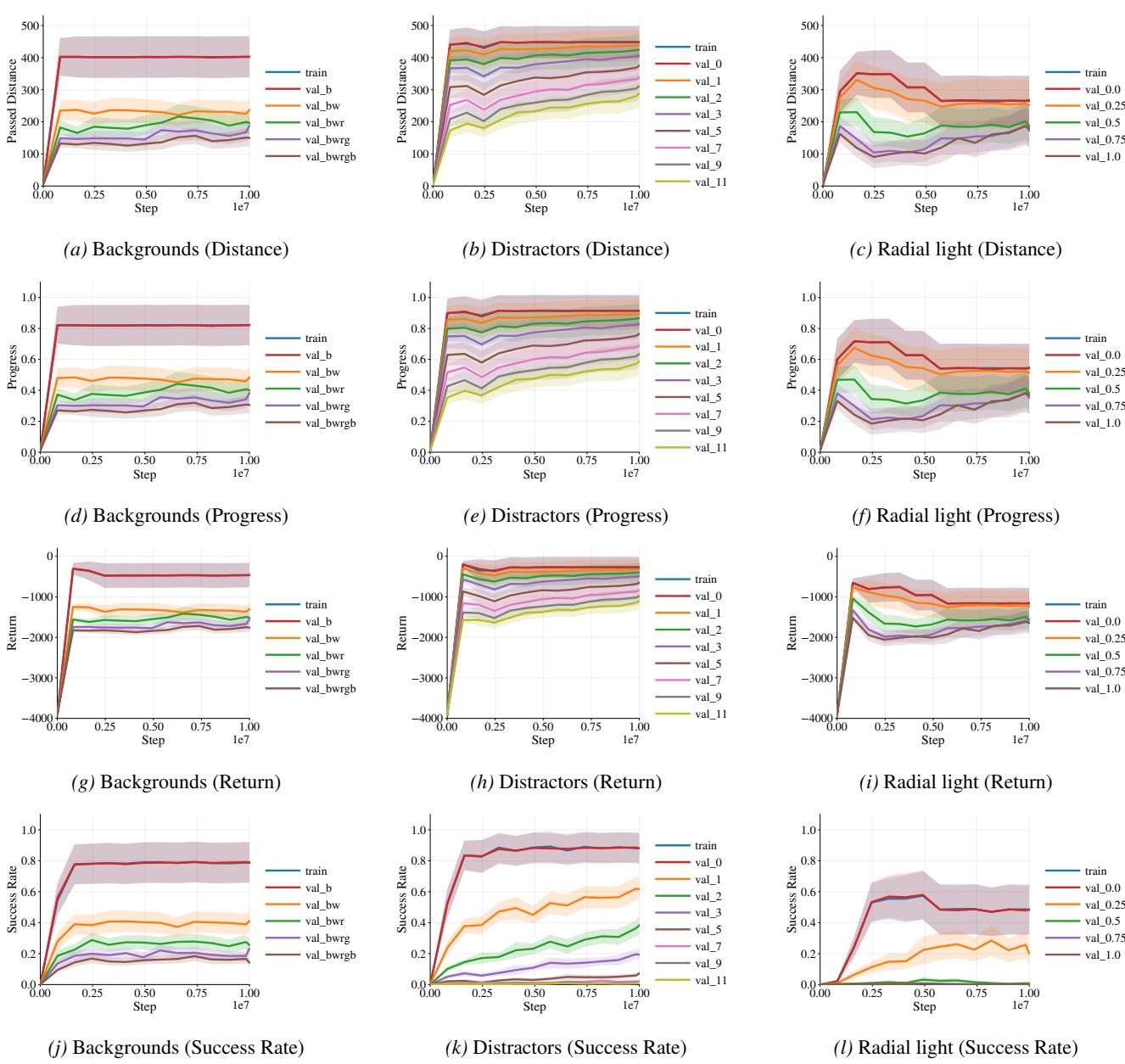

*(a)* Backgrounds (Distance)

*(b)* Distractors (Distance)

*(c)* Radial light (Distance)

*(d)* Backgrounds (Progress)

*(e)* Distractors (Progress)

*(f)* Radial light (Progress)

*(g)* Backgrounds (Return)

*(h)* Distractors (Return)

*(i)* Radial light (Return)

*(j)* Backgrounds (Success Rate)

*(k)* Distractors (Success Rate)

*(l)* Radial light (Success Rate)

*Figure 7.* **Visual generalization gaps in single-axis shifts across all metrics.** Each row shows a different metric (Distance, Progress, Return, Success Rate), and each column shows a different axis (Backgrounds, Distractors, Radial light effect). Blue curves show training performance, while colored curves show evaluation performance as the visual support is expanded along the selected axis. **Backgrounds:** trained on black background, evaluated with cumulative background-color evaluation supports (black → black+white → black+white+red → etc.). **Distractors:** trained without distractors, evaluated with increasing numbers of same-as-agent distractors. **Radial light effect:** trained without radial light effects, evaluated with increasing radial light strength.

*Table 4.* **Per-configuration robustness results for KAGE-Bench (mean±SEM).** Each row corresponds to a hard train-evaluation configuration pair within a known-axis suite. For each run, we record the maximum value attained by each metric during training; these maxima are then averaged across 10 random seeds. We report Success Rate (SR) and Return for both train and eval configurations, together with the resulting generalization gaps. Abbreviations: bg = background, ag = agent, dist = distractor, skelet = skeleton. Generalization gaps are color-coded: green indicates smaller gaps (better generalization), while red indicates larger gaps (worse generalization). We also trained on-policy RAD-PPO with crop-resize augmentation, as well as off-policy CURL, DrQ, and SVEA, but these runs did not converge (training SR $\approx 0.01 \pm 0.01$ across all considered configurations), so they are not included in the table. We hypothesize that runs using geometric augmentations failed to converge because successful platformer play relies on absolute on-screen positions of the agent and obstacles. This spatial information is critical for estimating distances to cliffs and obstacles and for precise jump timing; see Figure 8, Figure 9, and Figure 10. For comparison, a fully random policy achieves SR $= 0.00 \pm 0.00$ and Return $= -3892.2 \pm 10.5$; a reflective policy that always takes the RIGHT+JUMP action achieves SR $= 0.59 \pm 0.03$ and Return $= -5000.3 \pm 2.3$.

| | ID | Train config | Eval config | PPO-CNN | | | | | | RAD-PPO (cutout color) | | | | | |
| | | | | Evaluation on train config | | Evaluation on eval config | | Generalization gap | | Evaluation on train config | | Evaluation on eval config | | Generalization gap | |
| | | | | SR | Return | SR | Return | ΔSR, % | ΔRet., abs. | SR | Return | SR | Return | ΔSR, % | ΔRet., abs. |
| Background | 1 | black bg | noise bg | 0.90±0.10 | -111.2±102.5 | 0.01±0.00 | -1978.7±111.3 | 98.9 | 1867.5 | 1.00±0.00 | -12.3±0.8 | 0.12±0.04 | -304.1±29.0 | 88.0 | 291.8 |
| | 8 | black bg, skelet ag | purple bg, skelet ag | 0.95±0.05 | -25.7±16.3 | 0.01±0.00 | -1489.4±244.7 | 99.0 | 1463.7 | 1.00±0.00 | -12.9±1.1 | 0.31±0.05 | -261.2±20.5 | 69.0 | 248.3 |
| Distractors | 5 | no dist., circle teal ag | 7 same-as-ag shapes, circle teal ag | 0.75±0.13 | -199.9±117.2 | 0.06±0.01 | -618.3±156.8 | 92.0 | 418.4 | 1.00±0.00 | -13.2±1.1 | 0.34±0.02 | -190.9±9.1 | 66.0 | 177.7 |
| Effects | 3 | no effects | light count 4 | 0.89±0.10 | -118.8±101.8 | 0.04±0.01 | -757.2±50.8 | 95.5 | 638.5 | 1.00±0.00 | -12.0±1.2 | 0.23±0.03 | -535.4±27.4 | 77.0 | 523.4 |
| Filters | 1 | no filters | brightness 1 | 0.90±0.10 | -125.0±115.2 | 0.04±0.01 | -631.8±58.7 | 95.6 | 506.9 | 1.00±0.00 | -10.3±1.2 | 0.16±0.03 | -322.9±40.1 | 84.0 | 312.6 |
| | 4 | no filters | hue shift 180 | 0.86±0.10 | -128.2±104.3 | 0.01±0.00 | -856.1±95.5 | 98.8 | 727.9 | 1.00±0.00 | -11.0±1.8 | 0.03±0.02 | -682.7±21.4 | 97.0 | 671.7 |
| | 9 | no filters | radial light strength 1 | 0.60±0.16 | -580.6±257.6 | 0.01±0.00 | -1071.2±216.5 | 98.3 | 490.6 | 1.00±0.00 | -12.1±1.0 | 0.11±0.04 | -577.2±33.7 | 89.0 | 565.1 |
| Layout | 1 | cyan layout | red layout | 0.86±0.10 | -118.6±89.1 | 0.32±0.08 | -279.5±78.2 | 62.8 | 160.9 | 1.00±0.00 | -11.3±1.7 | 0.04±0.02 | -605.0±23.3 | 96.0 | 593.7 |

# C. Additional robustness baseline results

This appendix preserves the additional experiments introduced during rebuttal. The table compares PPO-CNN with RAD-PPO using cutout-color augmentation on the hardest KAGE-Bench configurations, and the figures visualize why spatial augmentations such as crop-resize and random shift can damage platformer observations.

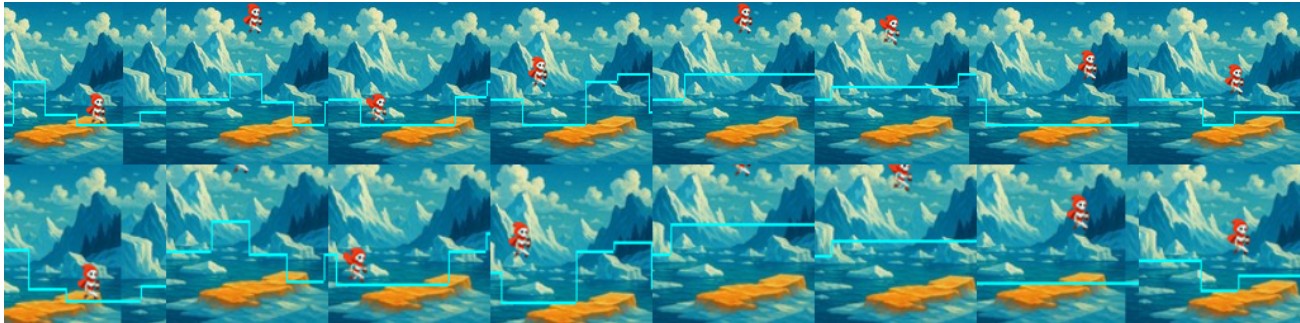

*Figure 8.* **Random crop resize augmentation preview.** The top row shows original observations, and the bottom row shows the corresponding augmented observations. Sometimes the agent is lost from the frame during augmentation and the spatial structure is disrupted, making it difficult for the agent to estimate distances to cliffs and obstacles and to time jumps precisely.

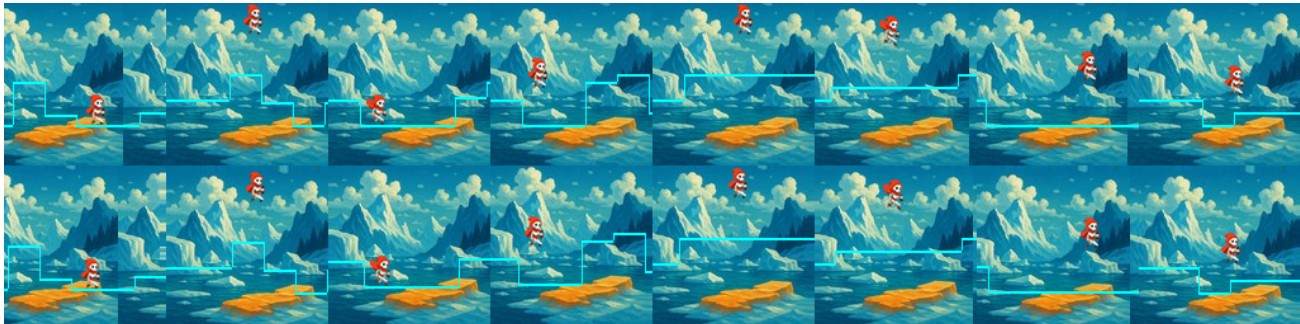

*Figure 9.* **Random shift augmentation preview.** The top row shows original observations, and the bottom row shows the corresponding augmented observations. Spatial structure is disrupted at every step, making it difficult for the agent to estimate distances to cliffs and obstacles and to time jumps precisely.

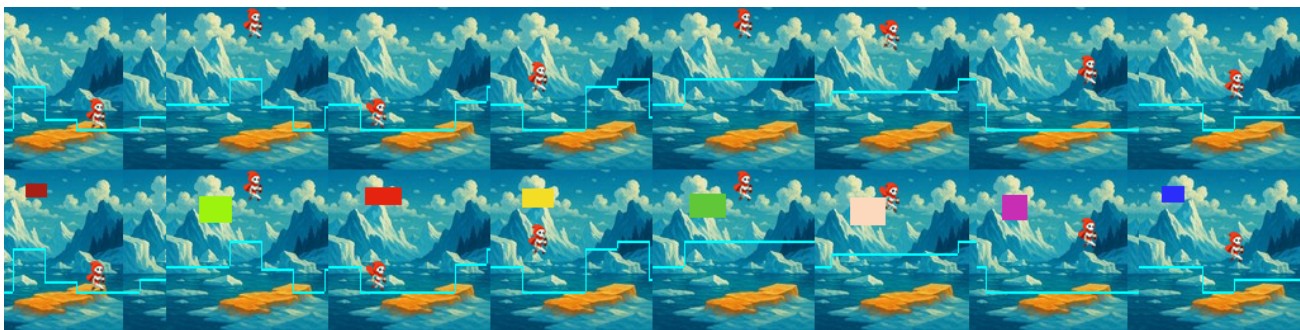

*Figure 10.* **Cutout color augmentation preview.** The top row shows original observations, and the bottom row shows the corresponding augmented observations.

# D. Benchmark Training Details

This appendix reports the full learning curves for the PPO-CNN baseline on all 34 KAGE-Bench train-evaluation configuration pairs. At each logging checkpoint, we evaluate the current policy on both the corresponding training configuration (in-distribution) and its paired evaluation configuration (out-of-distribution), and plot the resulting metrics over environment steps. Figures are grouped by generalization axis: **Agent Appearance** (Figure 11); **Background** (Figure 12, Figure 13); **Distractors** (Figure 15); **Effects** (Figure 16); **Filters** (Figure 17, Figure 18); and **Layout** (Figure 14).

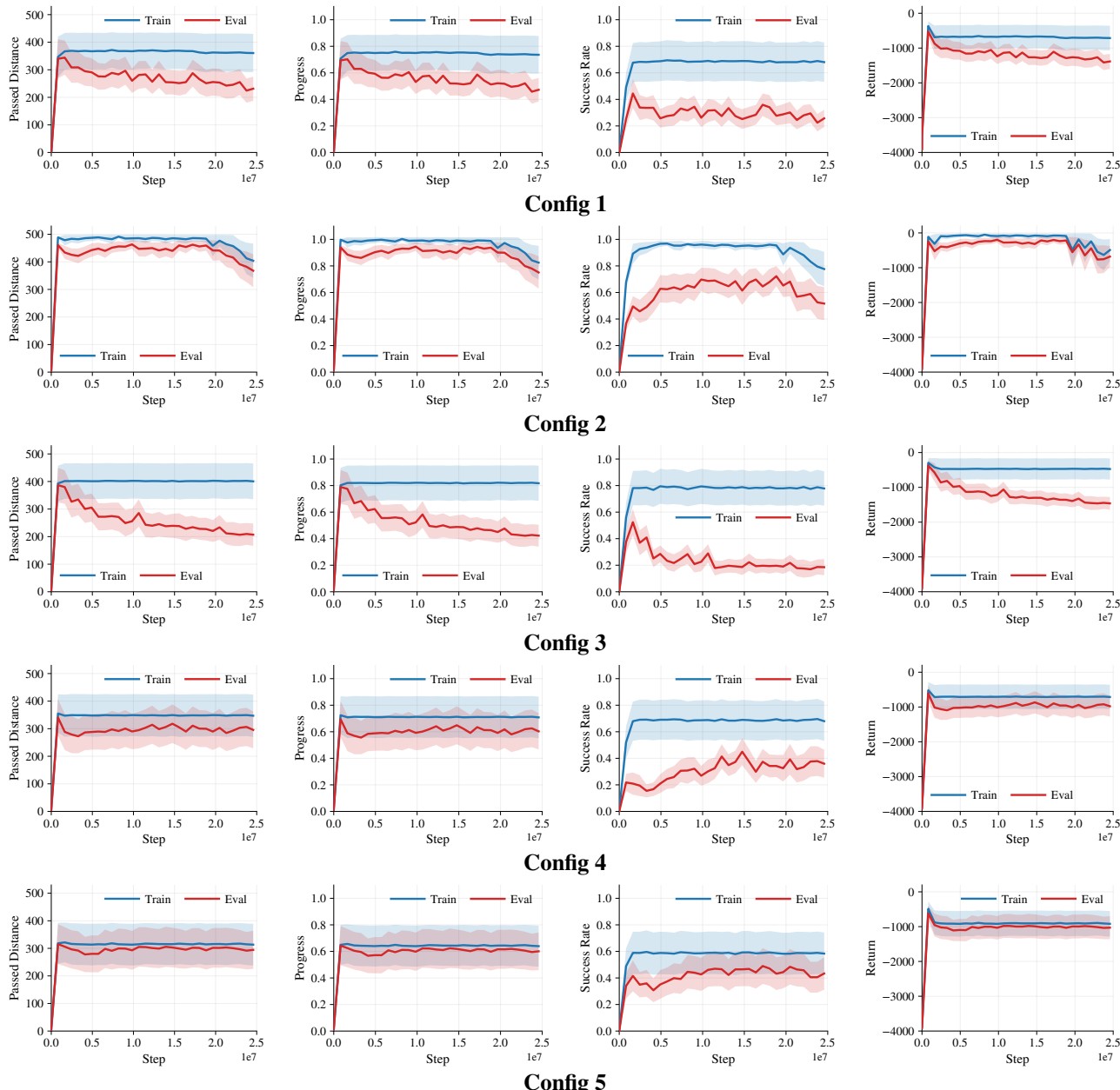

*Figure 11.* **Agent appearance training metrics for Configs 1–5:** covering passed distance, progress, success rate, and episodic return; curves are mean±sem across 10 independent runs.

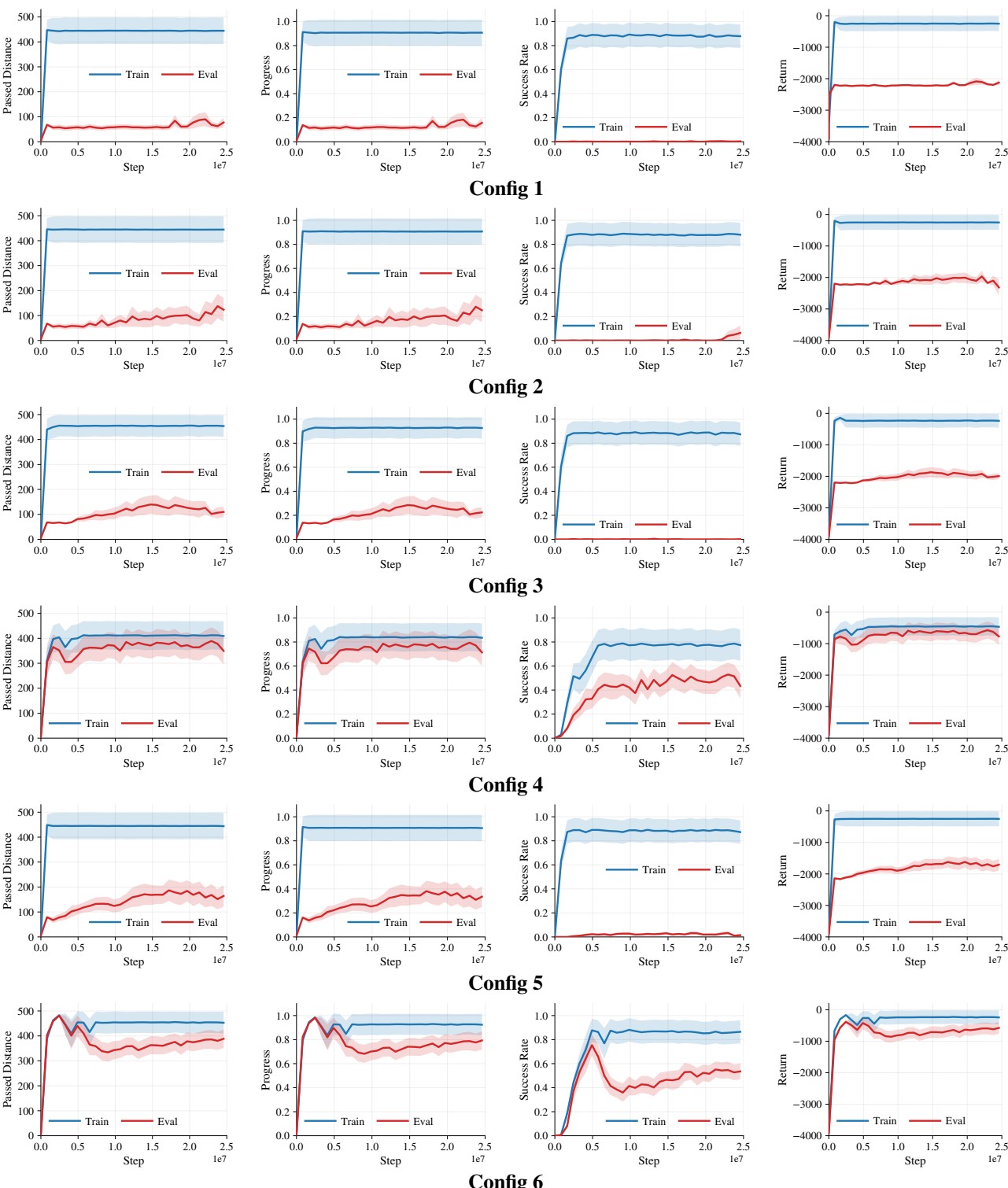

*Figure 12.* **Background-only training metrics for Configs 1–6:** showing passed distance, progress, success-once, and episodic return curves that represent mean±sem across 10 independent runs.

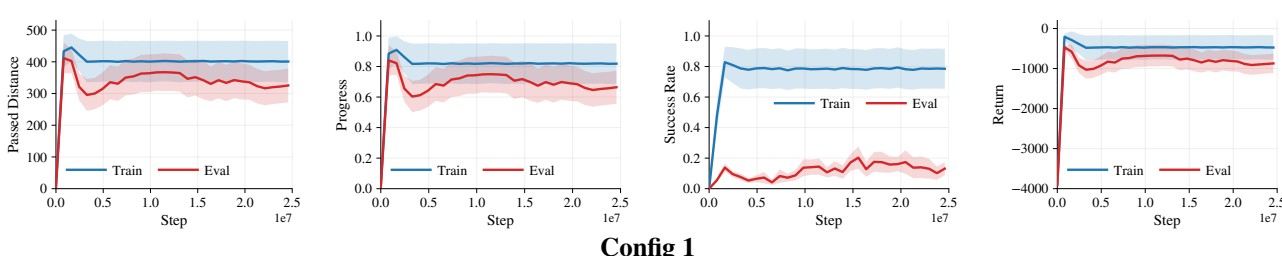

*Figure 13.* **Background-only training metrics for Configs 7–10:** showing passed distance, progress, success-once, and episodic return curves that represent mean±sem across 10 independent runs.

*Figure 14.* **Layout training metrics for Config 1:** plotting passed distance, progress, success rate, and episodic return; traces are mean±sem across 10 independent runs.

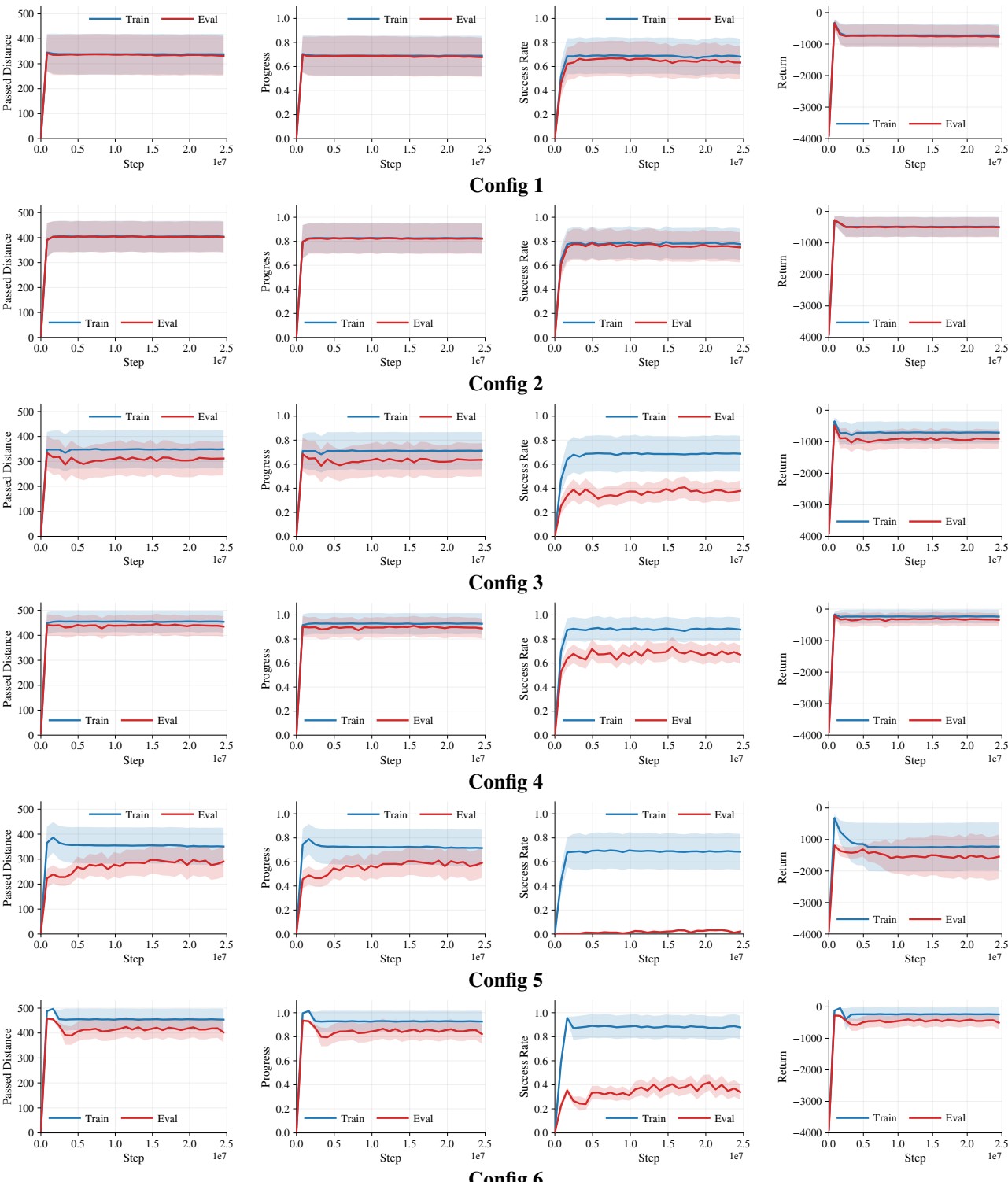

*Figure 15.* **Distractors training metrics for Configs 1–6:** with passed distance, progress, success-once, and episodic return curves; each trace is mean±sem across 10 independent runs.

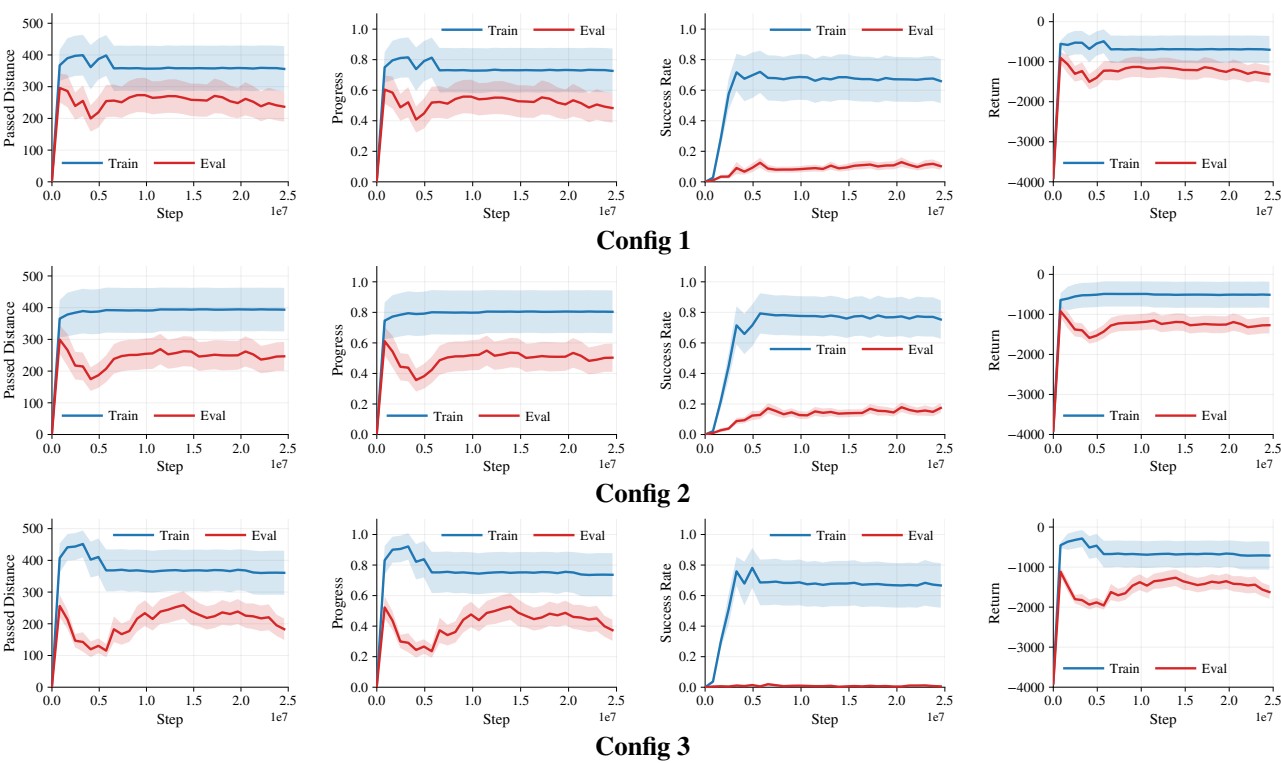

*Figure 16.* **Effects training metrics for Configs 1–3:** showing passed distance, progress, success rate, and episodic return; curves depict mean±sem across 10 independent runs.

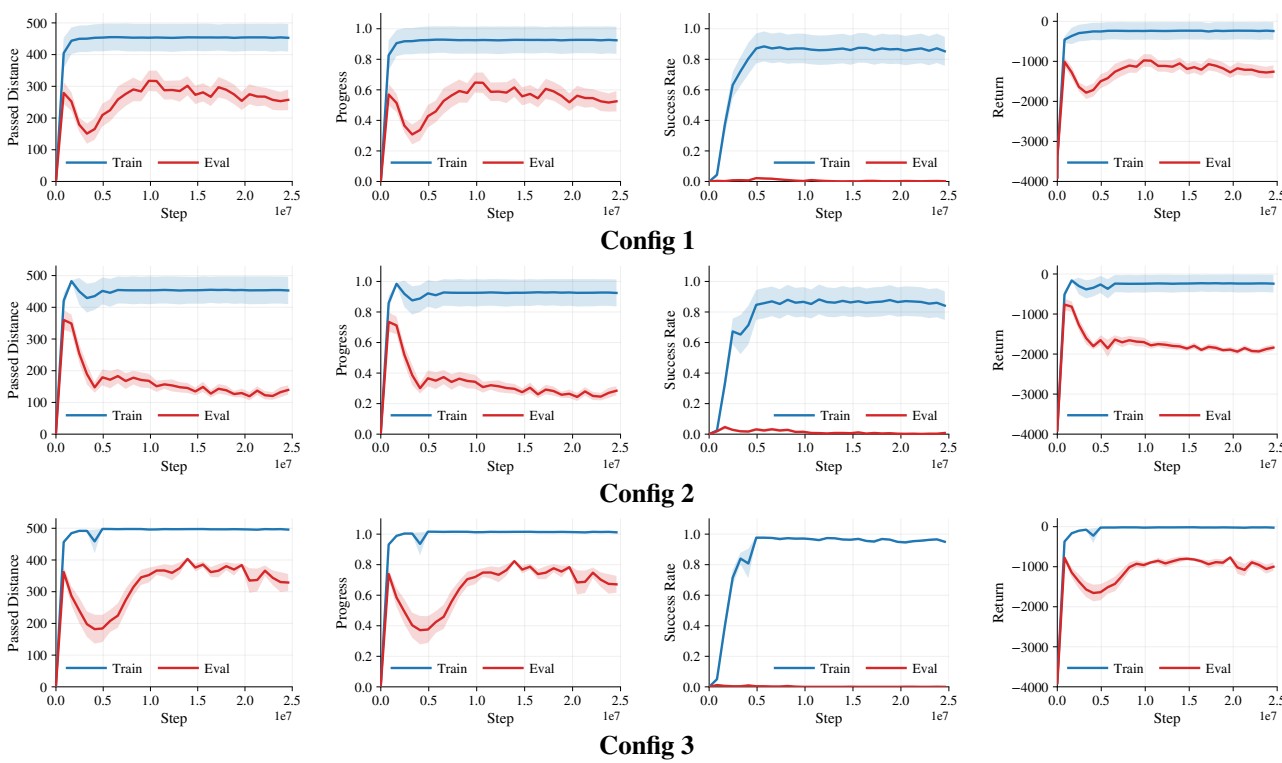

*Figure 17.* **Filters training metrics for Configs 1–3:** displaying passed distance, progress, success-once, and episodic return; curves show mean±sem across 10 independent runs.

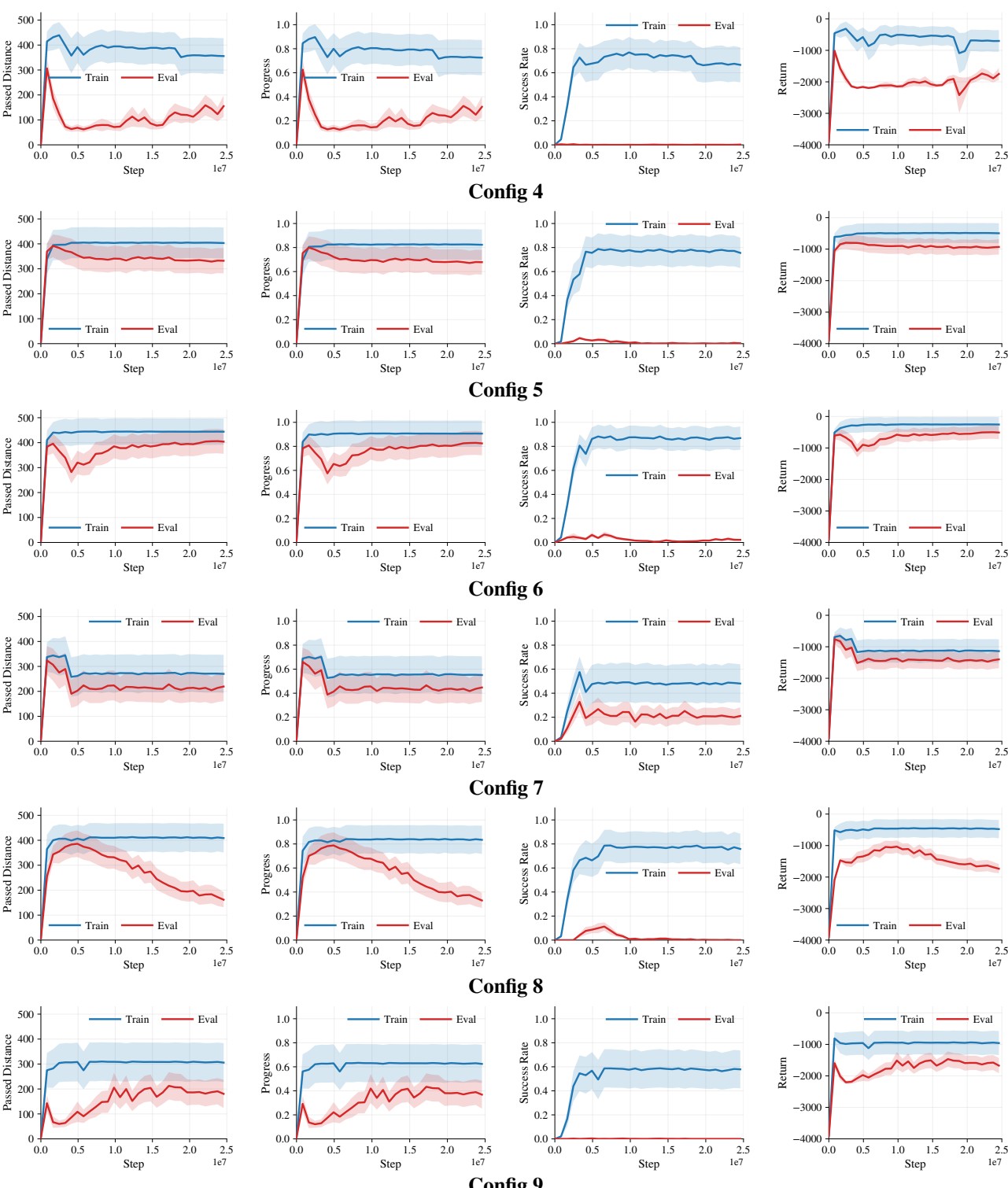

*Figure 18.* **Filters training metrics for Configs 4–9:** displaying passed distance, progress, success-once, and episodic return; curves show mean±sem across 10 independent runs.

# E. Generalization axes review

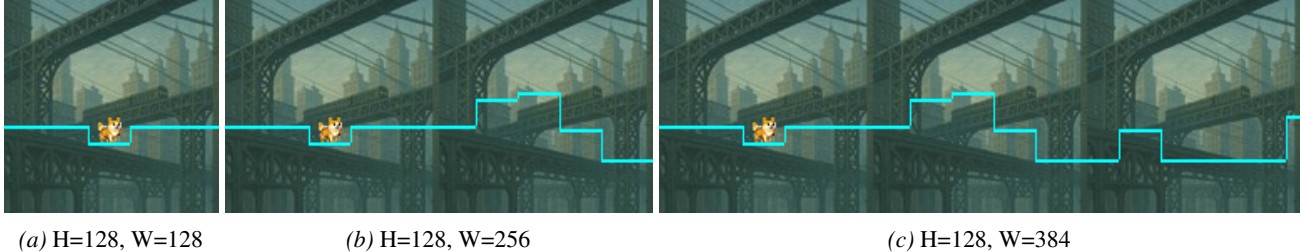

*(a)* H=128, W=128       *(b)* H=128, W=256       *(c)* H=128, W=384

*Figure 19.* **Global Screen Settings.** Representative renders under different screen configurations. YAML parameter: `H: 128, W: 128`.

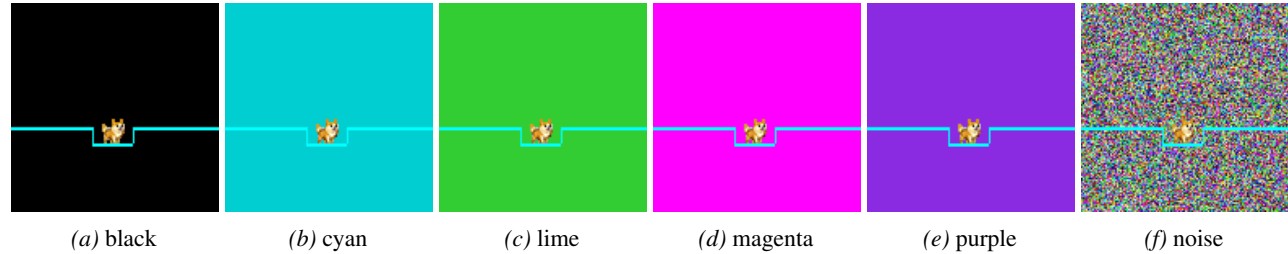

*(a)* black    *(b)* cyan    *(c)* lime    *(d)* magenta    *(e)* purple    *(f)* noise

*Figure 20.* **Background Color Modes.** Representative renders under different background color configurations. YAML parameter(s): `background.mode` (color/noise) and `background.color_names` controlling the palette for the color mode.

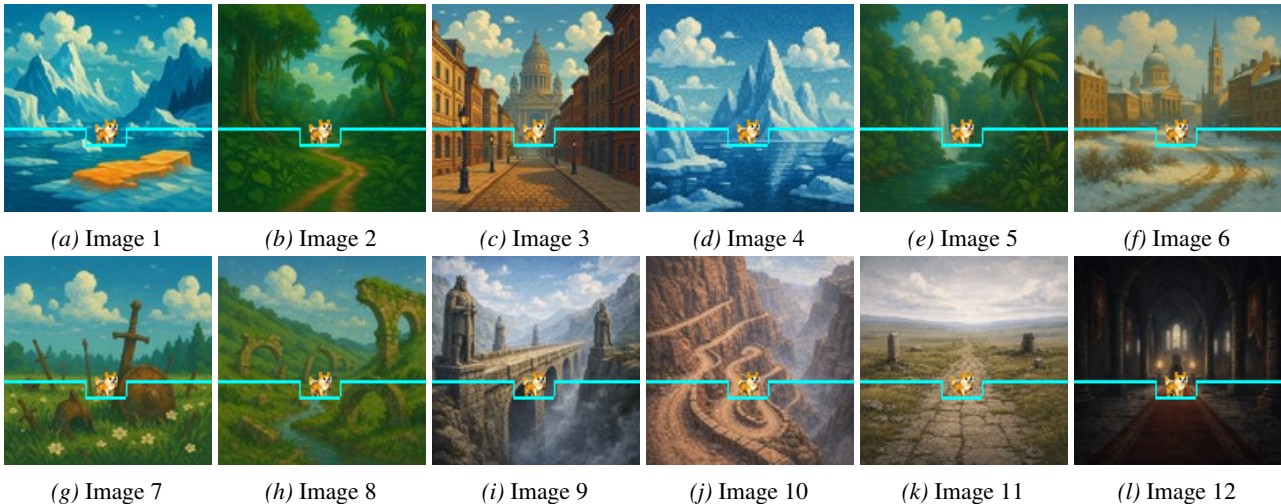

*(a)* Image 1    *(b)* Image 2    *(c)* Image 3    *(d)* Image 4    *(e)* Image 5    *(f)* Image 6

*(g)* Image 7    *(h)* Image 8    *(i)* Image 9    *(j)* Image 10    *(k)* Image 11    *(l)* Image 12

*Figure 21.* **Background Image Modes.** Representative renders under different background image configurations. YAML parameter(s): `background.mode: "image", background.image_paths`.

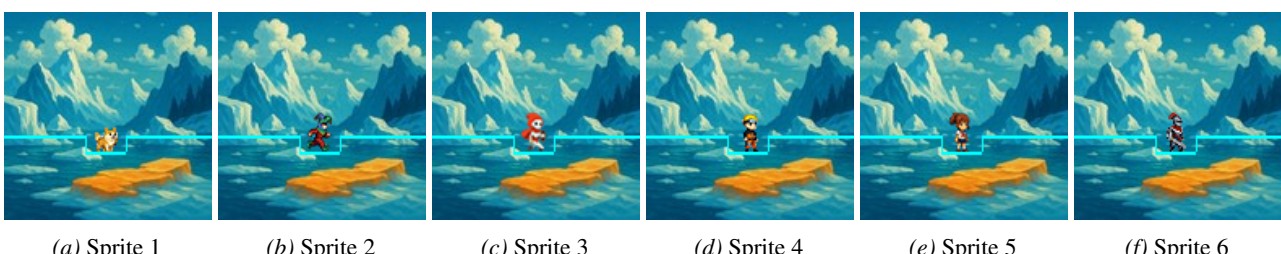

*(a)* Sprite 1    *(b)* Sprite 2    *(c)* Sprite 3    *(d)* Sprite 4    *(e)* Sprite 5    *(f)* Sprite 6

*Figure 22.* **Agent Sprites.** Representative renders showing different agent sprite configurations. YAML parameter(s): `character.use_sprites: true, character.sprite_paths`.

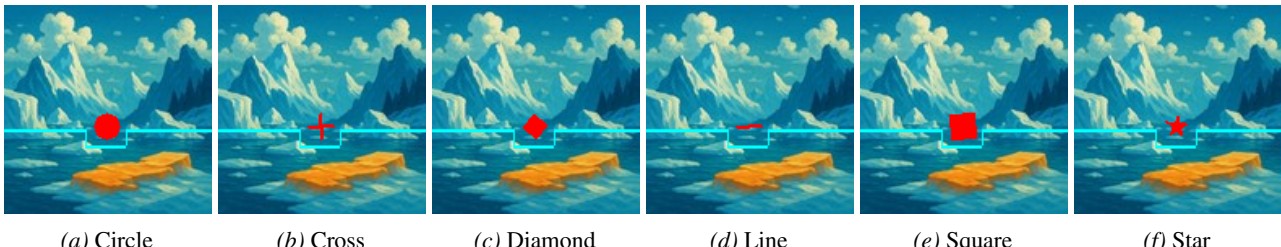

*(a)* Circle     *(b)* Cross     *(c)* Diamond     *(d)* Line     *(e)* Square     *(f)* Star

*Figure 23.* **Agent Shapes.** Representative renders showing different agent shape configurations. YAML parameter(s): `character.use_shape:  true, character.shape_types.`

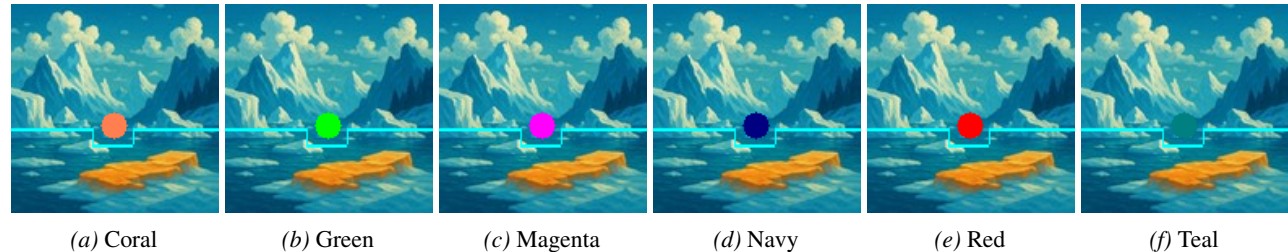

*(a)* Coral     *(b)* Green     *(c)* Magenta     *(d)* Navy     *(e)* Red     *(f)* Teal

*Figure 24.* **Agent Colors.** Representative renders showing different agent color configurations. YAML parameter(s): `character.use_shape:  true, character.shape_colors.`

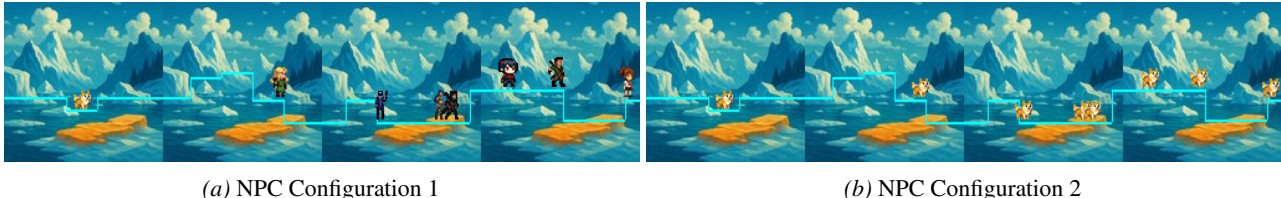

*(a)* NPC Configuration 1          *(b)* NPC Configuration 2

*Figure 25.* **NPCs.** Representative renders showing different NPC configurations. YAML parameter(s): `npc.enabled:  true, npc.sprite_dir, npc.min_npc_count, npc.max_npc_count.`

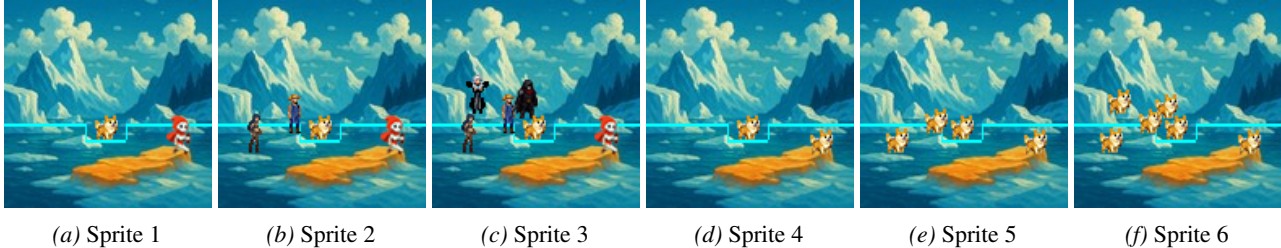

*(a)* Sprite 1     *(b)* Sprite 2     *(c)* Sprite 3     *(d)* Sprite 4     *(e)* Sprite 5     *(f)* Sprite 6

*Figure 26.* **Sticky NPCs.** Representative renders showing different sticky NPCs configurations. YAML parameter(s): `npc.sticky_enabled:  true, npc.min_sticky_count, npc.max_sticky_count, npc.sticky_sprite_dirs.`

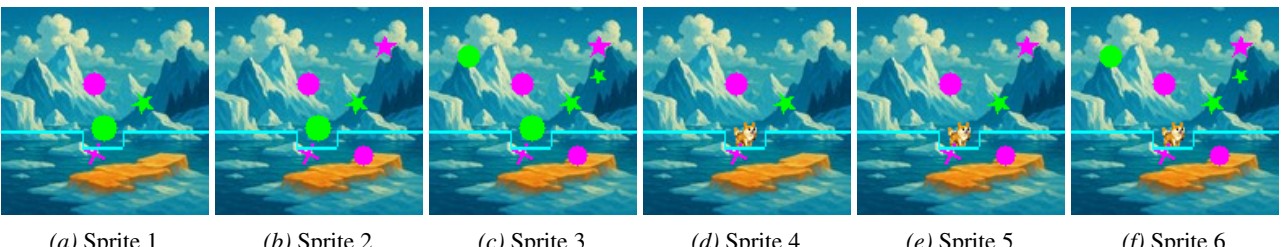

*(a)* Sprite 1     *(b)* Sprite 2     *(c)* Sprite 3     *(d)* Sprite 4     *(e)* Sprite 5     *(f)* Sprite 6

*Figure 27.* **Shape Distractors.** Representative renders showing different shape distractors configurations. YAML parameter(s): `distractors.enabled: true, distractors.count, distractors.shape_types, distractors.shape_colors.`

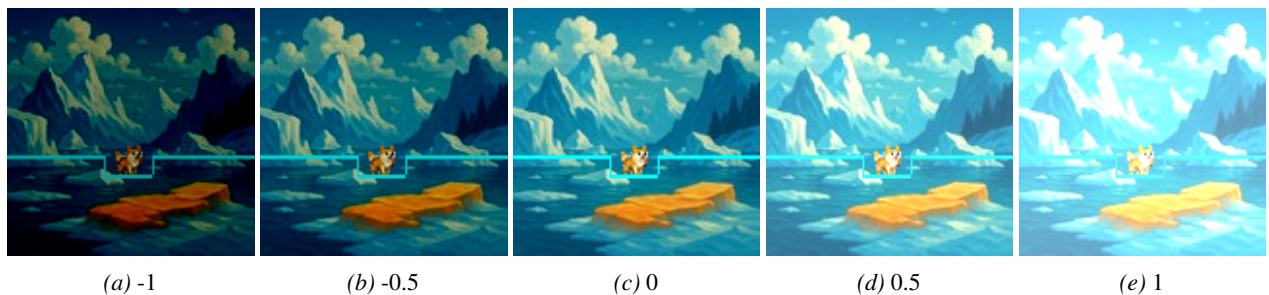

*(a)* -1     *(b)* -0.5     *(c)* 0     *(d)* 0.5     *(e)* 1

*Figure 28.* **Brightness Levels.** Representative renders showing different brightness configurations. YAML parameter(s): `filters.brightness` varied from -1 to 1, with other `filters.*` held at their base values.

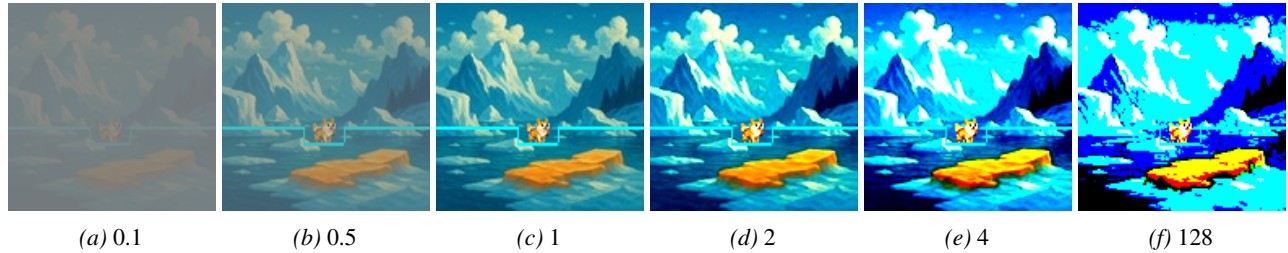

*(a)* 0.1     *(b)* 0.5     *(c)* 1     *(d)* 2     *(e)* 4     *(f)* 128

*Figure 29.* **Contrast Levels.** Representative renders showing different contrast configurations. YAML parameter(s): `filters.contrast` varies from 0.1 to 128 while other filters stay at defaults.

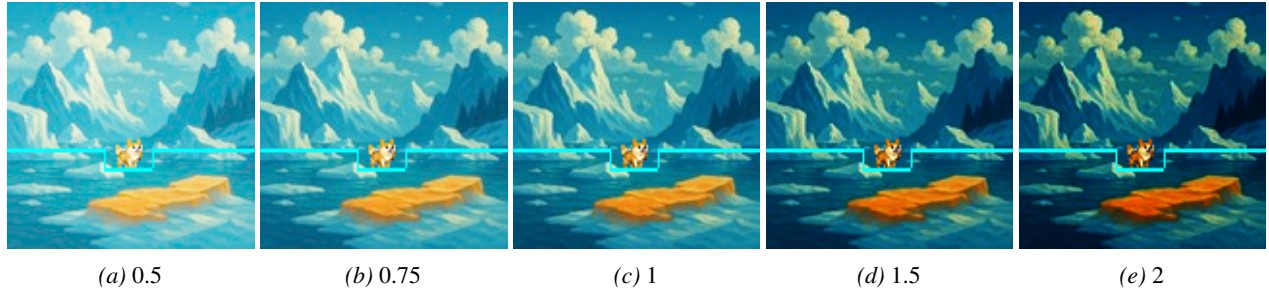

*(a)* 0.5     *(b)* 0.75     *(c)* 1     *(d)* 1.5     *(e)* 2

*Figure 30.* **Gamma Levels.** Representative renders showing different gamma configurations. YAML parameter(s): `filters.gamma` is swept from 0.5 to 2.0 (others default).

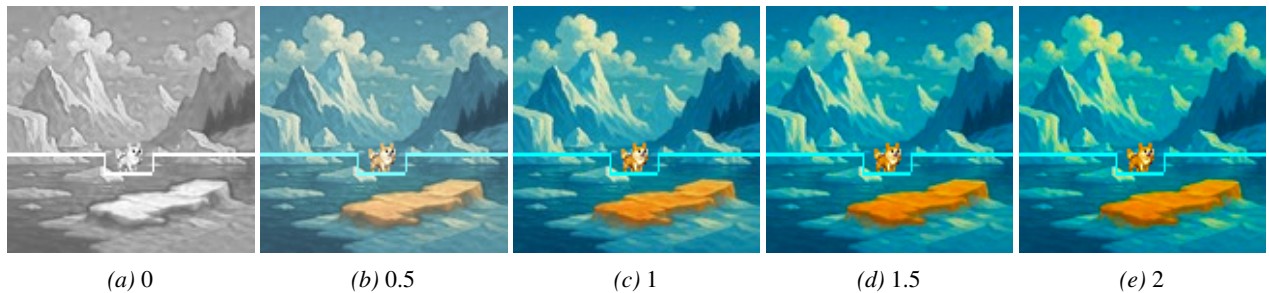

*(a)* 0     *(b)* 0.5     *(c)* 1     *(d)* 1.5     *(e)* 2

*Figure 31.* **Saturation Levels.** Representative renders showing different saturation configurations. YAML parameter(s): `filters.saturation` ranges from 0 to 2 with other filters unchanged.

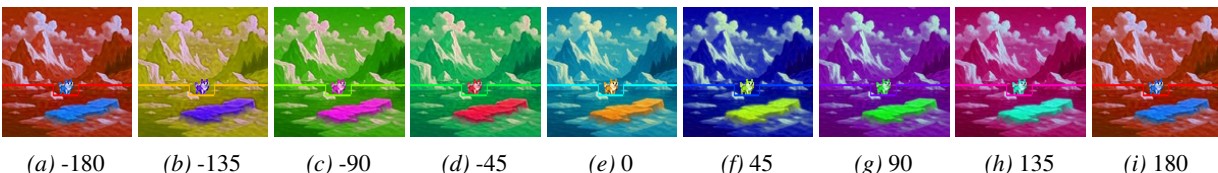

*(a)* -180    *(b)* -135    *(c)* -90    *(d)* -45    *(e)* 0    *(f)* 45    *(g)* 90    *(h)* 135    *(i)* 180

*Figure 32.* **Hue Shift Levels.** Representative renders showing different hue shift configurations. YAML parameter(s): `filters.hue_shift` sweeps through [-180, 180].

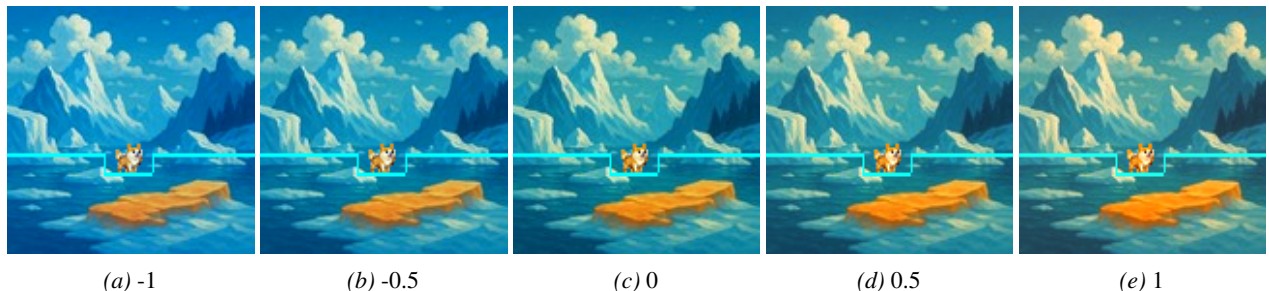

*(a)* -1      *(b)* -0.5      *(c)* 0      *(d)* 0.5      *(e)* 1

*Figure 33.* **Color Temperature Levels.** Representative renders showing different color temperature configurations. YAML parameter(s): `filters.color_temp` is varied between -1 and 1.

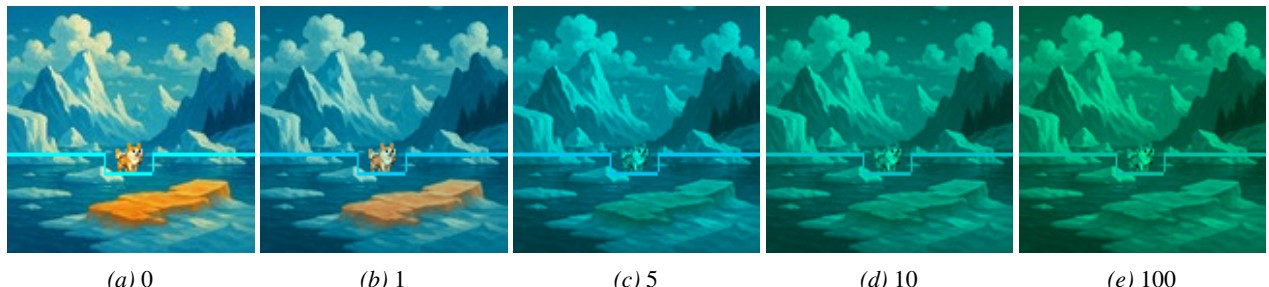

*(a)* 0      *(b)* 1      *(c)* 5      *(d)* 10      *(e)* 100

*Figure 34.* **Color Jitter Standard Deviation Levels.** Representative renders showing different color jitter configurations. This is a stochastic effect, and the jittering changes for each timestep. YAML parameter(s): `filters.color_jitter_std`.

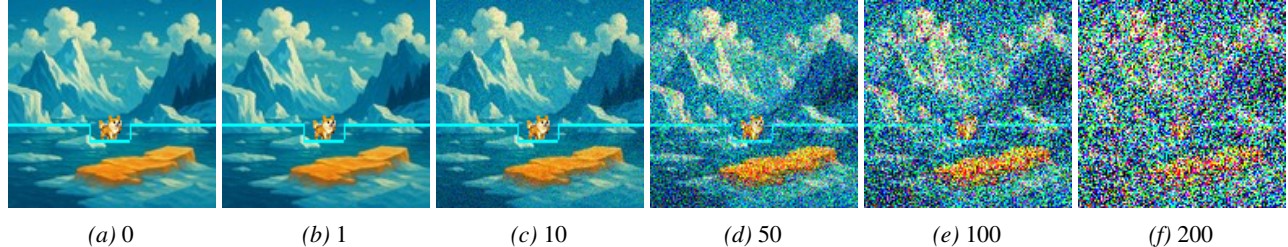

*(a)* 0      *(b)* 1      *(c)* 10      *(d)* 50      *(e)* 100      *(f)* 200

*Figure 35.* **Gaussian Noise Standard Deviation Levels.** Representative renders showing different gaussian noise configurations. This is a stochastic effect, and the noise changes for each timestep. YAML parameter(s): `filters.gaussian_noise_std` ranges from 0 to 200, with other filter noise terms disabled.

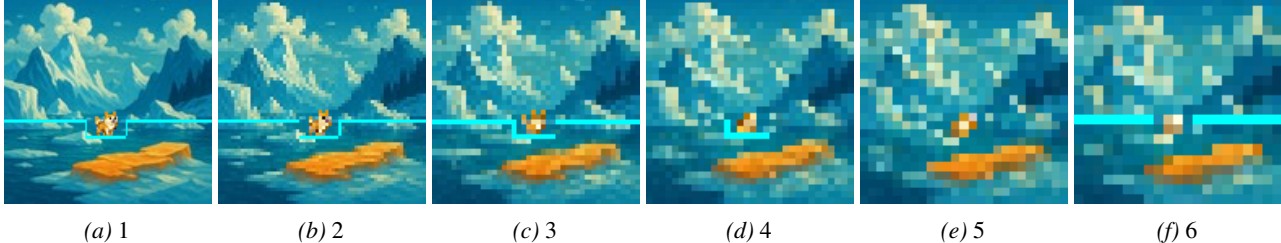

*(a)* 1      *(b)* 2      *(c)* 3      *(d)* 4      *(e)* 5      *(f)* 6

*Figure 36.* **Pixelate Factor Levels.** Representative renders showing different pixelate factor configurations. YAML parameter(s): `filters.pixelate_factor` steps from 1 to 6 while other filters stay default.

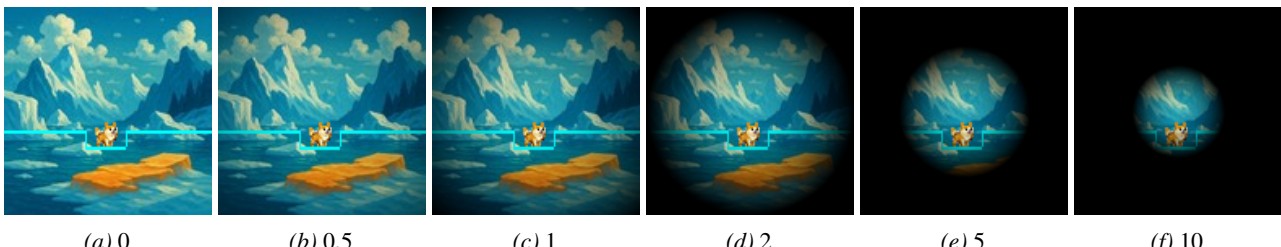

*(a)* 0      *(b)* 0.5      *(c)* 1      *(d)* 2      *(e)* 5      *(f)* 10

*Figure 37.* **Vignette Strength Levels.** Representative renders showing different vignette strength configurations. YAML parameter(s): `filters.vignette_strength` is increased from 0 to 10 (others default).

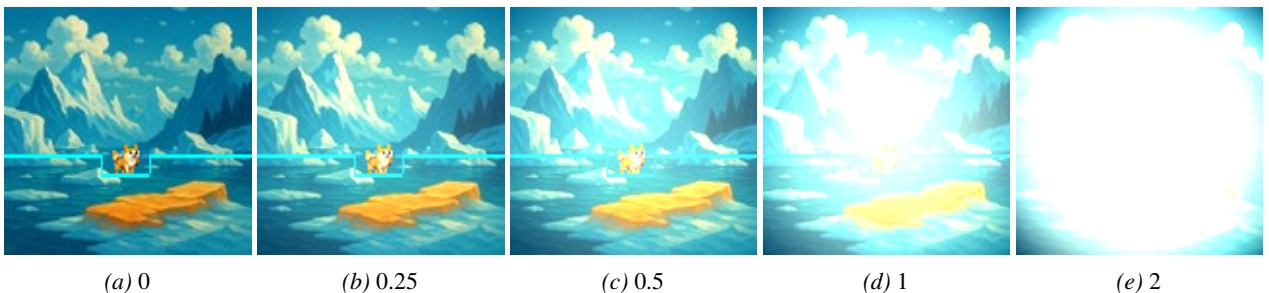

*Figure 38.* **Radial Light Strength Levels.** Representative renders showing different radial light strength configurations. YAML parameter(s): `filters.radial_light_strength` spans 0 to 2.

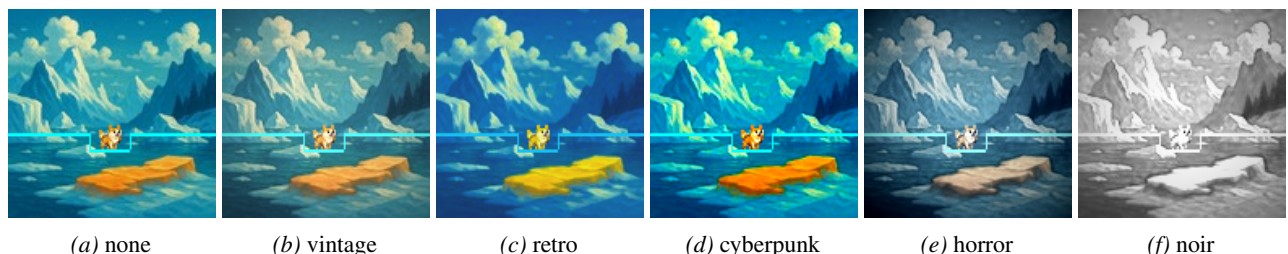

*Figure 39.* **Pop Filter List Presets.** Representative renders showing different pop filter preset configurations. YAML parameter(s): `filters.pop_filter_list`.

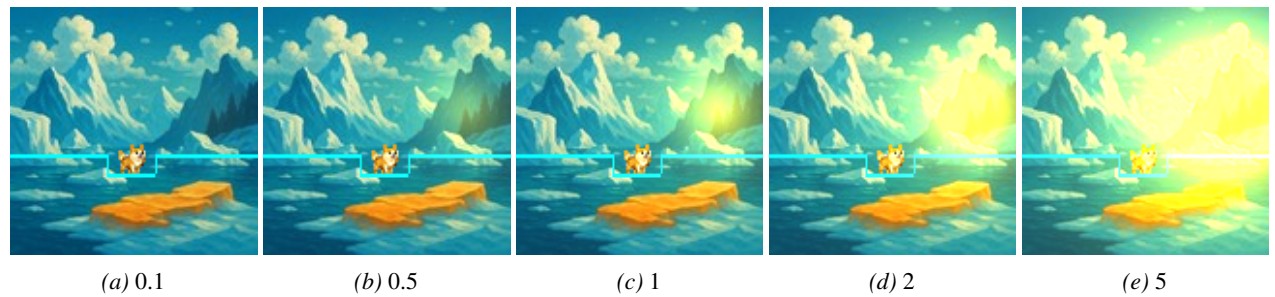

*Figure 40.* **Point Light Intensity Levels.** Representative renders showing different point light intensity configurations. YAML parameter(s): `effects.point_light_enabled: true`, `effects.point_light_intensity` varies from 0.1 to 5.

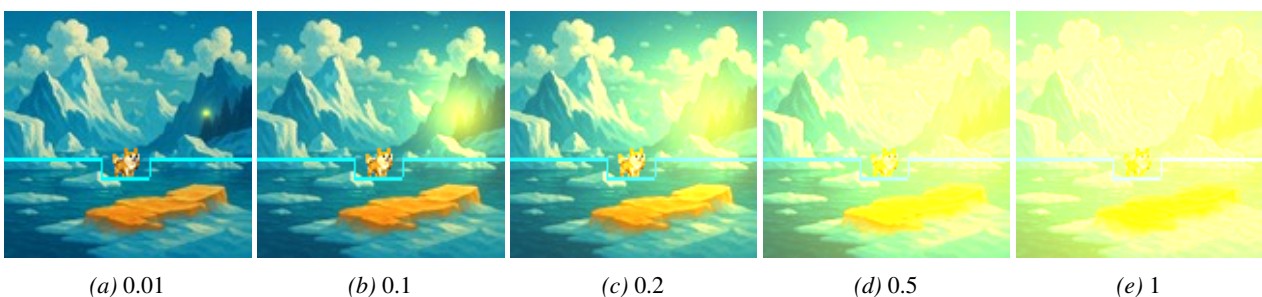

*Figure 41.* **Point Light Radius Levels.** Representative renders showing different point light radius configurations. YAML parameter(s): `effects.point_light_radius` sweeps from 0.01 to 1 (others fixed).

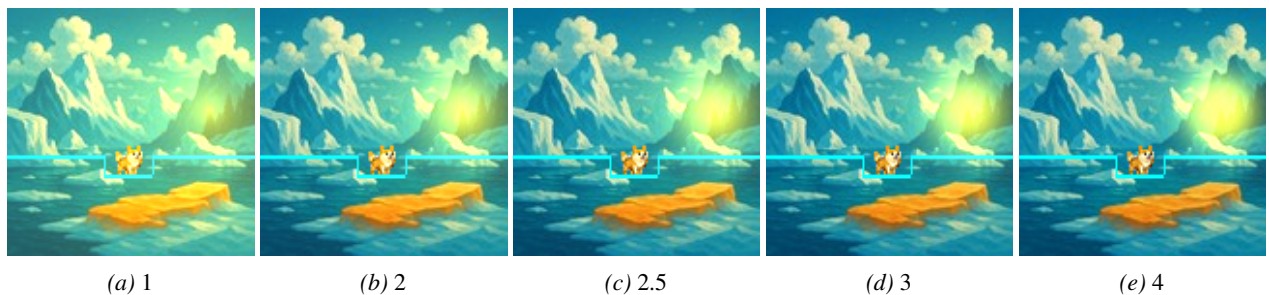

*(a)* 1     *(b)* 2     *(c)* 2.5     *(d)* 3     *(e)* 4

*Figure 42.* **Point Light Falloff Levels.** Representative renders showing different point light falloff configurations. YAML parameter(s): `effects.point_light_falloff` varies between 1 and 4.

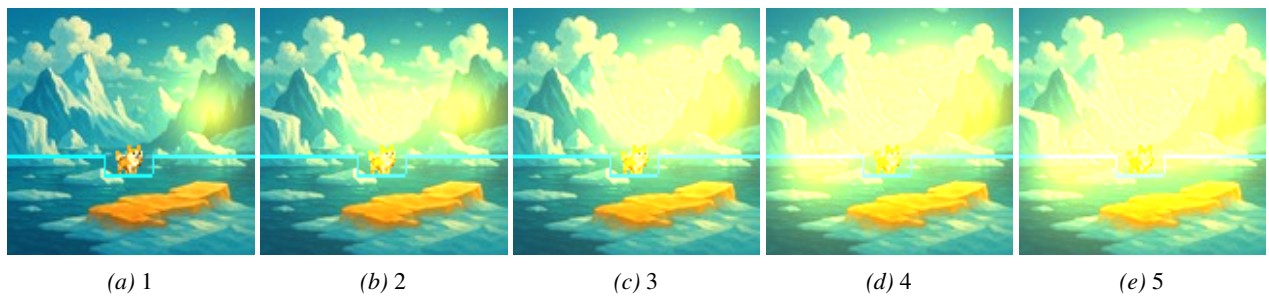

*(a)* 1     *(b)* 2     *(c)* 3     *(d)* 4     *(e)* 5

*Figure 43.* **Point Light Count Levels.** Representative renders showing different point light count configurations. YAML parameter(s): `effects.point_light_count` increases from 1 to 5.

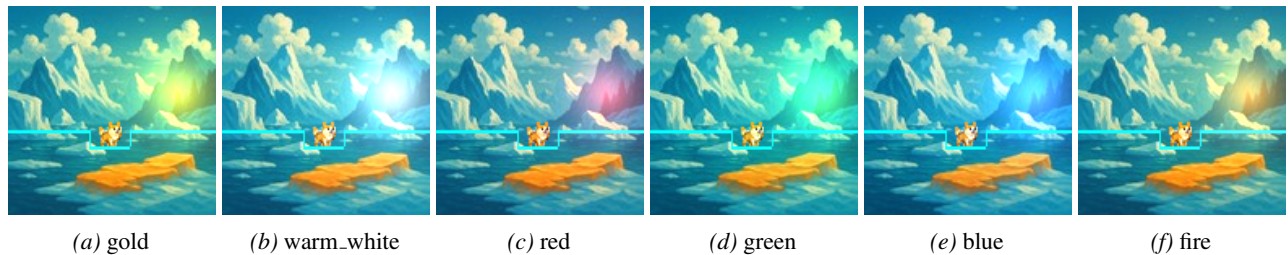

*(a)* gold     *(b)* warm_white     *(c)* red     *(d)* green     *(e)* blue     *(f)* fire

*Figure 44.* **Point Light Color Names.** Representative renders showing different point light color configurations. YAML parameter(s): `effects.point_light_color_names` lists the named lights (gold, warm_white, red, green, blue, fire).

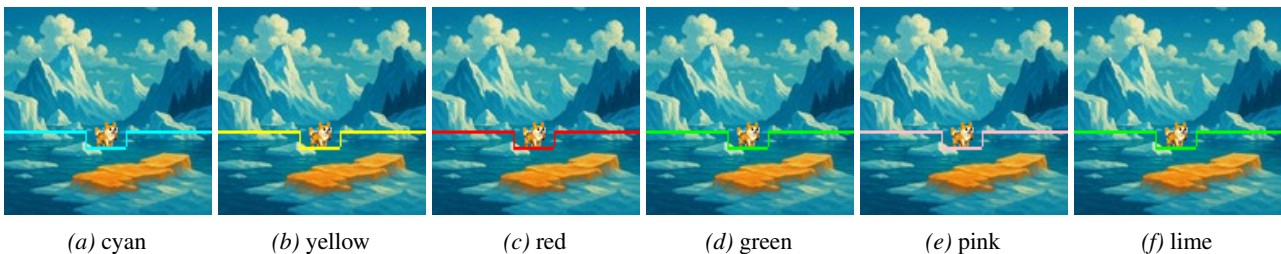

*(a)* cyan     *(b)* yellow     *(c)* red     *(d)* green     *(e)* pink     *(f)* lime

*Figure 45.* **Layout Colors.** Representative renders showing different layout color configurations. YAML parameter(s): `layout.layout_colors` selects the per-level palette.

# F. Backgrounds

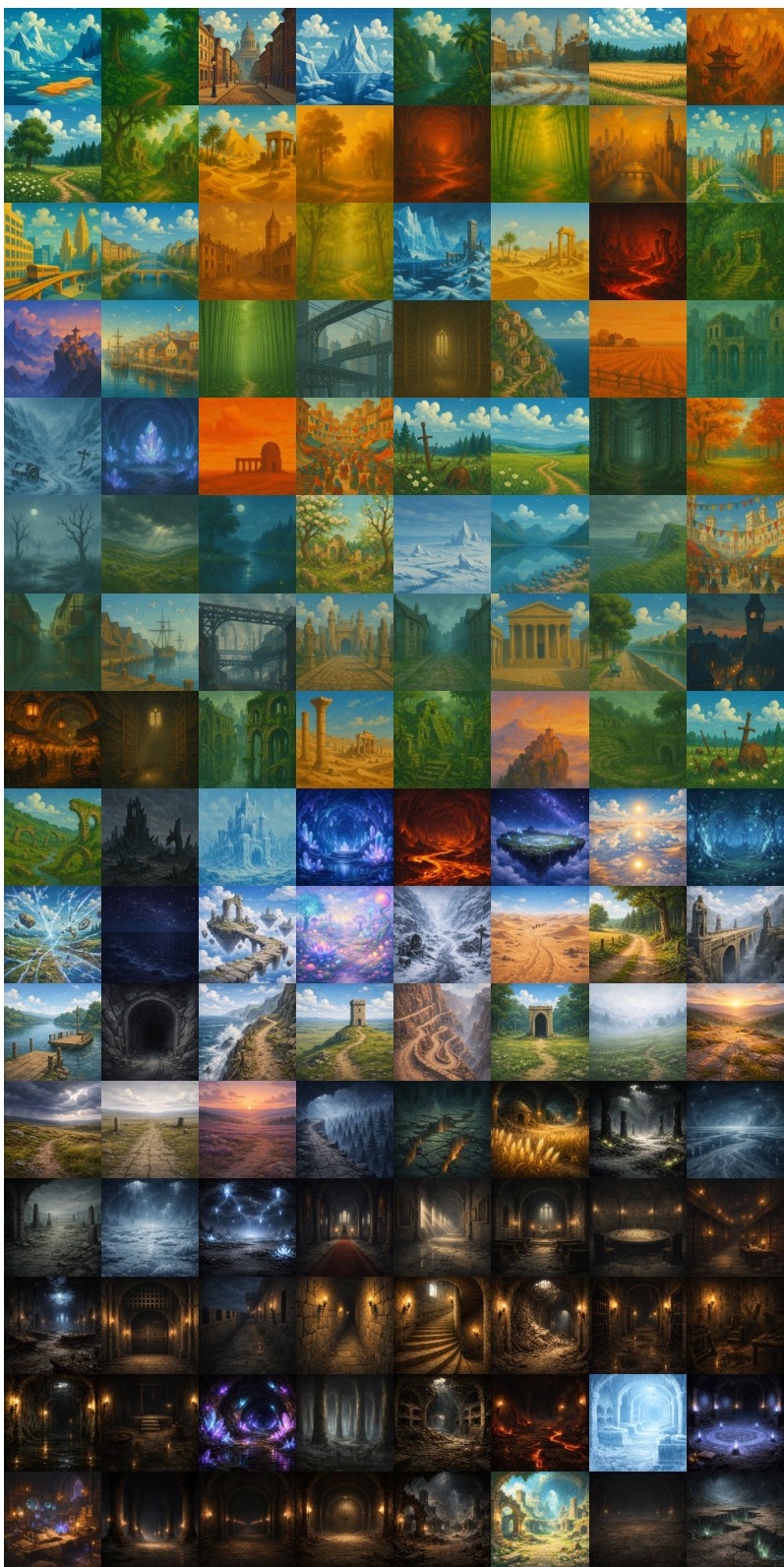

*Figure 46.* **Background palette** used in KAGE-Bench experiments (128 unique scenes). Images are 128×128 pixels and located in `src/kage_bench/assets/backgrounds`.

## G. Agents Sprites

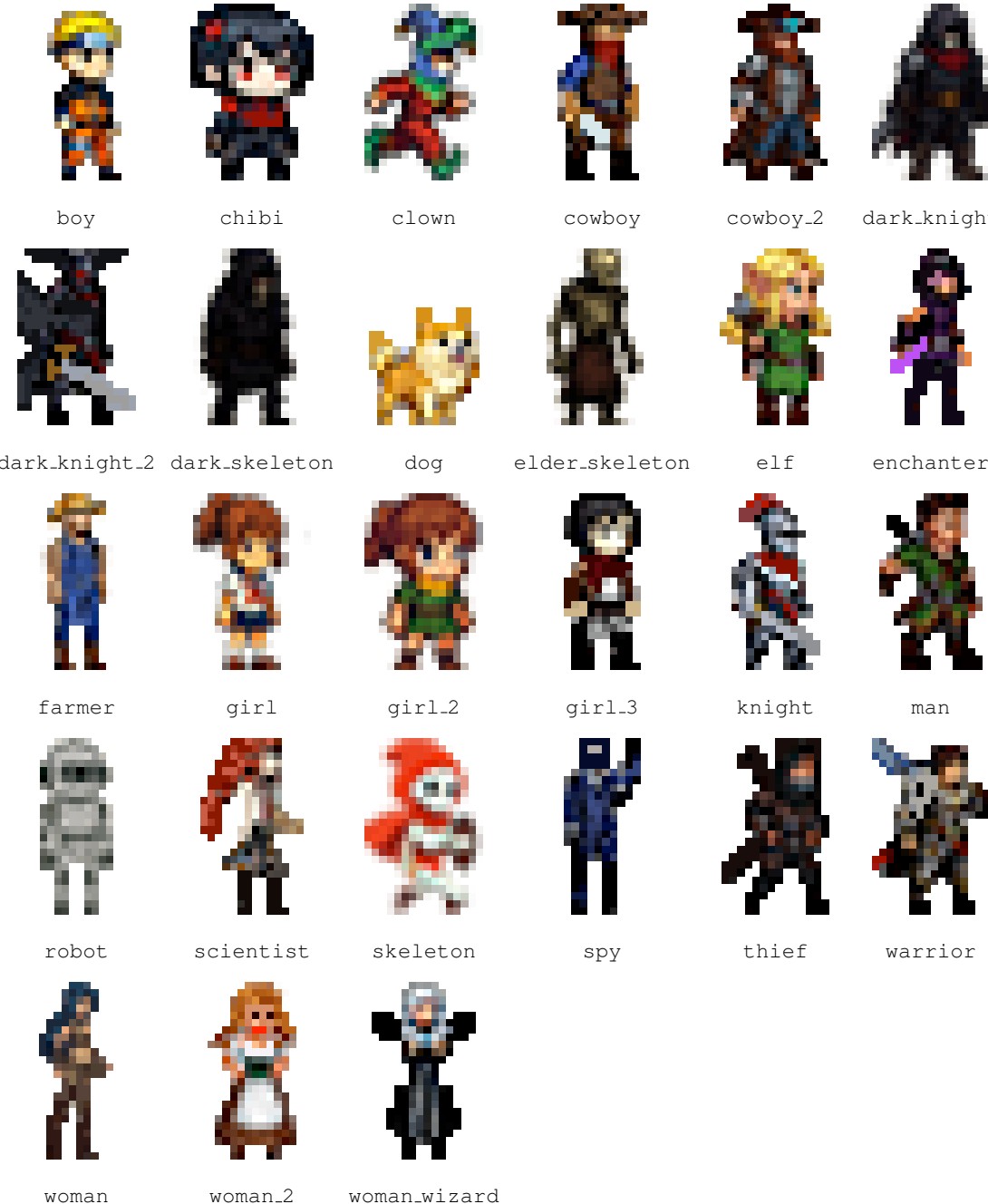

*Figure 47.* **Representative idle states** for each of the 27 animated agent sprites used in **KAGE-Env**. By default, sprite bounding box is 24×16 pixels. Sprites are located in `src/kage_bench/assets/sprites`.

# H. YAML Configuration Details

```yaml
# ==============================================================================
# JAX Platformer Environment Configuration
# ==============================================================================
# This configuration file allows precise control over all environment aspects.
# Edit this file to customize backgrounds, characters, NPCs, physics, and usage.
#
# Load this config using:
#   from kage_bench import EnvConfig, load_config_from_yaml
#   config = load_config_from_yaml("custom_config.yaml")
#   env = PlatformerEnv(config)
# ==============================================================================

# ------------------------------------------------------------------------------
# 0. Episode Settings
# ------------------------------------------------------------------------------
episode_length: 500        # Default: 500 | Max steps per episode
forward_reward_scale: 0.2  # Default: 0.2 | Reward for moving forward
jump_penalty: 10.0         # Default: 10.0 | Penalty for jumping
timestep_penalty: 0.1      # Default: 0.1 | Per-timestep reward penalty
idle_penalty: 5.0          # Default: 5.0 | Penalty when x does not change
dist_to_success: 490.0     # Default: 490.0 | Passed distance needed for success

# ------------------------------------------------------------------------------
# 1. Global Screen Settings
# ------------------------------------------------------------------------------
H: 128  # Default: 128 | Screen height in pixels
W: 128  # Default: 128 | Screen width in pixels

# 2. Background Settings
# ------------------------------------------------------------------------------
background:
  # * Mode (Default: "black"):
    # "black" - just black background
    # "image" - image backgrounds
    # "noise" - white noise background (unique for each episode)
    # "color" - color backgrounds
  mode: "image"

  # * Image mode settings (ignired if mode != "image"):
  # * Please, choose only one of the following options image_dir, image_paths, image_path:
  # [Option 1] Specify a directory to load all images from (randomly selected per episode)
  image_dir: "src/kage_bench/assets/backgrounds" # * Will use all the 128 images

  # [Option 2] Or specify list of explicit paths:
  # image_paths: # * Will use only listed images
  #   - "src/kage_bench/assets/backgrounds/bg-1.jpeg"
  #   - "src/kage_bench/assets/backgrounds/bg-64.jpeg"
  #   - "src/kage_bench/assets/backgrounds/bg-128.jpeg"

  # [Option 3] Or force a single image:
  # image_path: "path/to/your/image.jpeg"

  # Parallax/Tiling (ignored if mode != "image")
  parallax_factor: 0.5   # Default: 0.5 | 0.0 = static, 0.5 = slow scroll, 1.0 = locked to camera, <0 = moves
      ↪ left, >1 = moves fast
  tile_horizontal: true  # Default: true | Repeat image horizontally for infinite worlds

  # Dynamic switching (change background during the episode \ gif effect)
  switch_frequency: 0.0  # Default: 0.0 | Probability per step (0.0 = never switch, 1.0 = every step)

  # * Color mode settings (ignored if mode != "color"):
  # List of colors to randomly select from per episode
  # Available colors:
    # "black", "white", "red", "orange", "yellow", "green", "cyan", "blue",
    # "purple", "pink", "brown", "gray", "lime", "teal", "indigo", "magenta"
  color_names: ["purple", "teal", "indigo"]
```

*Code 3.* **YAML excerpt showing reward structure configuration and background settings for the KAGE-Env.**

```yaml
# ------------------------------------------------------------------------------
# 3. Character Settings (Player)
# ------------------------------------------------------------------------------
character:
  width: 16  # Default: 16 | Character width in pixels  | Recommended to not change
  height: 24 # Default: 24 | Character height in pixels | Recommented to not change

  # * Sprite Settings (ignored if use_shape: true):
  use_sprites: true # Default: true | Use sprites for character
  # * Please, choose only one of the following options sprite_dir, sprite_paths, sprite_path:
  # [Option 1] Directory containing sprite subdirectories (auto-discovers all subdirs as skins):
  sprite_dir: "src/kage_bench/assets/sprites"  # Will auto-discover and use all subdirectories like clown/,
      ↪ robot/, skelet/, etc.

  # [Option 2] List of directories for multiple skins (randomly selected per episode):
  # NOTE: Each directory should contain .png animation frames (e.g., walk-1.png, walk-2.png)
  # sprite_paths:
    # - "src/kage_bench/assets/sprites/clown"        # Skin 1
    # - "src/kage_bench/assets/sprites/dark_knight"  # Skin 2
    # - "src/kage_bench/assets/sprites/skeleton"       # Skin 3

  # [Option 3] Single sprite directory:
  # sprite_path: "path/to/your/folder/with/sprite"

  # Animation settings (ignored if use_sprites: false):
  enable_animation: true # Default: true | Enable sprite animation (if false, use only first frame)
  animation_fps: 12.0     # Default: 12.0 | Frames per second for sprite animation
  idle_sprite_idx: 0      # Default: 0 | Which sprite to use when idle (typically first)

  # * Shape Settings (ignored if use_sprites: true):
  use_shape: false # Default: false | Use shapes for character

  # Available shapes:
    # "circle", "cross", "diamond", "ellipse", "line",
    # "polygon", "square", "star", "triangle"
  shape_types: ["circle", "star"]

  # Available colors:
    # "red", "green", "blue", "orange", "yellow", "violet", "magenta",
    #"cyan", "pink", "brown", "purple", "lime", "navy", "maroon",
    # "olive", "teal", "indigo", "coral", "gold", "silver", "white"
  shape_colors: ["teal", "indigo"]
  shape_rotate: true       # Default: true | Enable shape rotation
  shape_rotation_speed: 5.0 # Default: 5.0 | Rotation speed in degrees per second
```

*Code 4.* **Character configuration showing sprite directories, animation controls, and optional shape fallback settings.**

```
# ------------------------------------------------------------------------------
# 4. NPC Settings (Non-Player Characters)
# ------------------------------------------------------------------------------
npc:
  # Default: true | Enable NPC system:
  # * World-Fixed NPCs (stand on platforms)
  enabled: true        # Default: true | Enable World-Fixed NPC system:
  min_npc_count: 5     # Default: 5 | Minimum number of NPCs per level
  max_npc_count: 20    # Default: 20 | Maximum number of NPCs per level
  spawn_y_offset: 0    # Default: 0 | Offset from ground in pixels (positive = up, negative = down)
  animation_fps: 12.0  # Default: 12.0 | Animation speed for NPCs

  # * Please, choose only one of the following options sprite_dir, sprite_paths, sprite_path:
  # [Option 1] Specify a directory to auto-discover all sprite subdirectories:
  sprite_dir: "src/kage_bench/assets/sprites"  # Will auto-discover all subdirs like robot/, girl/, skelet/, etc
        ↪ .

  # [Option 2] Or specify list of explicit paths:
  # sprite_paths:
  #   - "src/kage_bench/assets/sprites/clown"        # Skin 1
  #   - "src/kage_bench/assets/sprites/dark_knight"  # Skin 2
  #   - "src/kage_bench/assets/sprites/skeleton"        # Skin 3

  # [Option 3] Or force a single sprite directory:
  # sprite_path: "path/to/your/folder/with/sprite"

  # * Sticky NPCs (follow camera and always visible in observation):
  sticky_enabled: false  # Default: false | Enable sticky NPC system:
  min_sticky_count: 1    # Default: 1 | Minimum number of sticky NPCs per level
  max_sticky_count: 5    # Default: 5 | Maximum number of sticky NPCs per level
  # * Please, choose only one of the following options sticky_sprite_dir, sticky_sprite_dirs, sticky_sprite_path
        ↪ :
  # [Option 1] Specify a directory to auto-discover all sticky sprite subdirectories:
  sticky_sprite_dir: "src/kage_bench/assets/sprites"  # Will auto-discover all subdirs

  # [Option 2] Or specify list of explicit paths (backward compatibility):
  # sticky_sprite_dirs: # * Will use only listed sprite directories
  #   - "src/kage_bench/assets/sprites/clown"        # Skin 1
  #   - "src/kage_bench/assets/sprites/dark_knight"  # Skin 2
  #   - "src/kage_bench/assets/sprites/skeleton"        # Skin 3

  # [Option 3] Or force a single sticky sprite directory:
  # sticky_sprite_path: "path/to/your/folder/with/sprite"

  # Sticky NPCs settings:
  sticky_can_jump: true          # Default: true | Enable jumping for sticky NPCs
  sticky_jump_probability: 0.01  # Default: 0.01 | Probability of jumping per step
  sticky_y_min_offset: -40       # Default: -40  | Minimum Y offset from ground (negative = higher)
  sticky_y_max_offset: -10       # Default: -10  | Maximum Y offset from ground (0 = on ground)
  sticky_x_offsets: []           # Default: []   | Camera-relative X offsets for sticky NPCs (e.g., [-40, 0,
        ↪ 40]). If not provided, NPCs will be spread around agent
  sticky_x_min: -60              # Default: -60  | Minimum X offset from center
  sticky_x_max: 60               # Default: 60   | Maximum X offset from center

# ------------------------------------------------------------------------------
# 5. Distractors Settings
# ------------------------------------------------------------------------------
distractors:
  # Whether to enable distractors, i.e. moving geometric shapes on background (always visible in observation)
  enabled: true # Default: false | Enable distractors system:
  count: 5       # Default: 5 | Number of distractors per level

  # Available shapes:
    # "circle", "cross", "diamond", "ellipse",
    # "line", "polygon", "square", "star", "triangle"
  shape_types: ["circle", "star", "cross"]

  # Available colors:
    # "red", "green", "blue", "orange", "yellow", "violet", "magenta",
    # "cyan", "pink", "brown",  "purple", "lime", "navy", "maroon",
    # "olive", "teal", "indigo", "coral", "gold", "silver", "white"
  shape_colors: ["red", "green", "blue"]

  # Dynamics:
  can_move: true            # Default: true | Whether distractors move around
  min_speed: 0.0            # Default: 0.0 | Minimum movement speed (pixels per step)
  max_speed: 1.0            # Default: 2.0 | Maximum movement speed (pixels per step)
  can_rotate: true          # Default: true | Whether distractors rotate
  min_rotation_speed: -0.3  # Default: -3.0 | Minimum rotation speed (degrees per step)
  max_rotation_speed: 0.3   # Default: 3.0 | Maximum rotation speed (degrees per step)
  min_size: 4               # Default: 4 | Minimum size
  max_size: 12              # Default: 12 | Maximum size
```

Code 5. **NPC, sticky NPC, and distractor parameters that govern entity counts, visuals, and motion cues.**

```yaml
# ------------------------------------------------------------------
# 6. Filter & Effect Settings
# ------------------------------------------------------------------
filters:
  brightness: 0.0          # Default: 0.0 | [-1, 1] additive exposure, scaled in code
  contrast: 1.0            # Default: 1.0 | >0, scales around mid-gray (128)
  gamma: 1.0               # Default: 1.0 | [0.5, 2.0] power-law on [0,1]
  saturation: 1.0          # Default: 1.0 | [0, 2] HSV S multiplier
  hue_shift: 0.0           # Default: 0.0 | [-180, 180] degrees, HSV hue offset
  color_temp: 0.0          # Default: 0.0 | [-1, 1] warm(+R,-B) vs cool(+B,-R)

  # Stochastic effects (require PRNG key):
  color_jitter_std: 0.0    # Default: 0.0 | >=0, std of 3x3 RGB mixing perturbation
  gaussian_noise_std: 0.0  # Default: 0.0 | >=0, pixelwise N(0, std^2) in [0,255]
  poisson_noise_scale: 0.0 # Default: 0.0 | [0,1], shot noise with lambda = img*scale

  # Spatial / detail transforms:
  blur_sigma: 0.0          # Default: 0.0 | >=0, box-blur approximation strength
  sharpen_amount: 0.0      # Default: 0.0 | >=0, unsharp mask gain
  pixelate_factor: 1       # Default: 1 | int>=1, down/up nearest (1 disables)

  # Global shading / lighting overlays:
  vignette_strength: 0.0      # Default: 0.0 | >=0, edge darkening factor (code expects ~[0,1])
  radial_light_strength: 0.0 # Default: 0.0 | >=0, additive center light (code expects ~[0,1])

  # Optional preset stack:
  pop_filter_list: []      # Default: [] | ["vintage","retro","cyberpunk","horror","noir"]

# ------------------------------------------------------------------
# 7. Effects Settings
# ------------------------------------------------------------------
effects:
  # Point light effect
  point_light_enabled: false  # Default: false | Enable/disable point light effects
  point_light_intensity: 1.0  # Default: 1.0 | Light intensity in [0.1, 5.0]
  point_light_radius: 0.1     # Default: 0.1 | Light radius as fraction of image size [0.01, 1.0]
  point_light_falloff: 2      # Default: 2.0 | Falloff exponent in [1.0, 4.0], higher = sharper

  # Multiple random lights
  point_light_count: 4        # Default: 1 | Number of lights in [1, 5]

  # Available colors:
    # 'warm_white', 'cool_white', 'yellow', 'orange', 'red',
    # 'green', 'cyan', 'blue', 'purple', 'pink', 'gold', 'fire'
  point_light_color_names: ["blue", "pink", "gold"]   # Default: ["warm_white"] | List of color names from
      ↪ LIGHT_COLORS

# ------------------------------------------------------------------
# 8. Level Layout
# ------------------------------------------------------------------
layout:
  length: 2048             # Default: 2048 | Length of the level in pixels
  height_px: 128           # Default: 128 | Height of the level in pixels
  base_ground_y: 96        # Default: 96 | [70, 127] Hight of the ground level. Higher - lower ground level
  pix_per_unit: 2          # Default: 2 | [0, 3] Pixels per height unit. 0 - flat, 3 - larger steps
  ground_thickness: 2      # Default: 2 | [1, 10] Thickness of the ground band in pixels
  run_width: 25            # Default: 20 | [1, 60] Widths of the stair in pixels
  p_change: 0.7            # Default: 0.7 | [0, 1] Probability of height change. 0 -never change, 1 -always
      ↪ change
  p_up_given_change: 0.5   # Default: 0.5 | [0, 1] Probability of height increase given height change. 0 -always
      ↪ decrease, 1 -always increase
  min_step_height: 5       # Default: 5 | [1, 17] Minimum height of the step in pixels
  max_step_height: 17      # Default: 10 | [1, 17] Maximum height of the step in pixels

  # Colors for platforms (randomly selected per episode). Available colors:
    # "black", "white", "red", "orange", "yellow", "green", "cyan", "blue",
    # "purple", "pink", "brown", "gray", "lime", "teal", "indigo", "magenta"
  layout_colors: ["cyan"] # Default: ["cyan"] | List of color names from COLOR_PALETTE

# ------------------------------------------------------------------
# 9. Physics Settings
# ------------------------------------------------------------------
physics:
  gravity: 0.75            # Default: 0.75 | [0.1, 1.0] Gravity force
  move_speed: 1            # Default: 1 | int and > 1 | Move speed
  jump_force: -7.5         # Default: -7.5 | [-10.0, 0.0] Jump force. Negative is upwards
  ground_friction: 0.8     # Default: 0.8 | [0.1, 1.0] Friction force
  air_resistance: 0.95     # Default: 0.95 | [0.1, 1.0] Air resistance force
  max_fall_speed: 8.0      # Default: 8.0 | [0.1, 10.0] Maximum fall speed
```

*Code 6.* **Filter, lighting, layout, and physics defaults that define the environment appearance and agent dynamics.**

*Table 5.* **PPO-CNN training and model hyperparameters.** We report the exact settings used in our JAX/Flax PPO baseline (CNN encoder with separate actor/critic heads) for KAGE-Env.

| Category | Setting |
|---|---:|
| **Environment & rollout** | |
| Parallel envs | $n_{\text{envs}} = 128$ |
| Rollout length | $n_{\text{steps}} = 128$ |
| Total timesteps | $T = 25{,}000{,}000$ |
| Discount / GAE | $\gamma = 0.999, \lambda = 0.95$ |
| Observation | RGB frame, shape $(H, W, 3)$, uint8 |
| Frame stacking | disabled (optional 4-stack) |
| Auto-reset | enabled |
| Reward normalization | disabled |
| Reward clipping | none $([-\infty, \infty])$ |
| **Optimization & PPO** | |
| Optimizer | Adam $(\varepsilon = 10^{-5})$ |
| Learning rate | $5 \cdot 10^{-4}$ (linear anneal: off) |
| Batch size | $B = n_{\text{envs}} \cdot n_{\text{steps}} = 16{,}384$ |
| Minibatches | 8 (minibatch size $B/8 = 2{,}048$) |
| Update epochs | $K = 3$ |
| Advantage normalization | on |
| Policy clip | $\epsilon_{\text{clip}} = 0.2$ |
| Value loss | clipped (same $\epsilon_{\text{clip}}$) |
| Value coefficient | $c_v = 0.5$ |
| Entropy coefficient | $c_H = 0.01$ (anneal: off) |
| Grad clip | global norm $\leq 0.5$ |
| Target KL | none |
| **Network (CNN encoder + heads)** | |
| Input scaling | $x \leftarrow x/255$ |
| Conv1 | 32 channels, $8{\times}8$, stride 4, valid, ReLU |
| Conv2 | 64 channels, $4{\times}4$, stride 2, valid, ReLU |
| Conv3 | 64 channels, $3{\times}3$, stride 1, valid, ReLU |
| MLP trunk | FC 512 + ReLU |
| Actor head | linear $\rightarrow$ 8 logits (orthogonal init, gain 0.01) |
| Critic head | linear $\rightarrow$ 1 (orthogonal init, gain 1) |
| Initialization | orthogonal (trunk gain $\sqrt{2}$), bias 0 |
| Action distribution | categorical; sampled via Gumbel-max |
| **Evaluation** | |
| Eval frequency | every 300 iterations |
| Eval episodes | 128 episodes |
| Eval parallelism | 32 envs |

