# OpenReview forum: "KAGE-Bench: Fast Known-Axis Visual Generalization Evaluation for Reinforcement Learning"
_ICML.cc/2026/Conference — ICML 2026 regular_

### Official Review · Reviewer_DH9i · 2026-02-26

**Soundness:** 2
**Presentation:** 4
**Significance:** 2
**Originality:** 2
**Overall Recommendation:** 2
**Confidence:** 4

**Summary:**

The paper introduces KAGE-Bench, a benchmark for evaluating visual generalization in reinforcement learning. It is built on KAGE-Env, a fast JAX-native 2D platformer in which visual components such as background, agent appearance, distractors, filters, lighting effects, and layout can be independently controlled. The benchmark consists of train-evaluation pairs that vary along a single visual axis at a time, with other axes held constant. Using a PPO-CNN baseline, the authors evaluate generalization across 34 configuration pairs. The results rank the visual axes by their impact on generalization, with background and filter shifts causing the largest drops.

**Compliance With Llm Reviewing Policy:**

Affirmed.

**Final Justification:**

After the final reply by the authors, my remaining concerns are:

1. **Relevance**. The strongest working baseline is RAD-PPO (cutout-color), a straightforward augmentation method from 2020. It does show generalization gaps, but we don’t know if more recent and specialized methods (listed in my initial review) would not already close these gaps. Since the work is positioned as a *benchmark*, it is important to demonstrate that it is not already solved by existing methods. The authors do state that “*configurations can be extended as methods improve*”, yet there is no evidence as to whether these extended configurations would be more difficult or how this extension should be done or by whom.
2. **Simplicity.** The authors argue that the simple dynamics of KAGE-env aim to isolate visual generalization from other confounds. I understand and agree with this motivation, but it reduces the benchmark to what is essentially a supervised perception problem embedded in an RL loop. The agent needs to detect its position relative to platforms and select from a small action set to perform a near-fixed motor routine. I do see general value in solving toy problems in the context of RL, and the community has done this extensively. However, in this case, it remains unclear to me whether insights from this controlled but minimal setting transfer elsewhere.
3. **Throughput reporting.** The authors separate KAGE-Env (the environment) from KAGE-Bench (the benchmark), and in that light, the 33M raw SPS figure is a fair claim in the context of the environment contribution. However, this is short of telling the full picture. It is more relevant to the users of the benchmark to know what they ought to expect in terms of simulation and time during training. A benchmark paper should lead with "how long does it take to train and evaluate on this benchmark," not "how fast can the environment step in isolation.” The latter has little value if most of this 33M is eaten up by rendering pixel observations. The ~600x performance gap between raw environment simulation (33M SPS) and training throughput (52K SPS) ought to be better highlighted.
4. **Positioning**. Throughout the discussion, the authors have shifted how they characterize KAGE-Bench depending on which criticism is being addressed. In response to concerns about task simplicity, it is a "***controlled diagnostic benchmark***"; in response to concerns about saturation, it is "***not a fixed benchmark but a configurable diagnostic framework***". Yet the paper's title, framing, and structure present it as a benchmark in the conventional sense. It ought to be the paper’s job to manage the expectations of the reader, and this could use improvement. A benchmark needs to demonstrate that it is not trivially solved, and a diagnostic framework needs to demonstrate that it yields interesting insights that are generalizable or potentially transfer elsewhere, such as more complex settings. The paper does not convincingly deliver on either framing.
5. **Metrics.** The authors persist in keeping multiple metrics along with the success and return metrics. Admittedly, additional metrics do not harm the evaluation. However, they muddy the waters. I would expect a **benchmark** to report a **central metric** based on which we compare methods. This could have been handled better in this work. It still appears to me that the chosen threshold D=490 exaggerates the generalization gap, and the chosen metrics heavily overlap without reporting anything additionally novel.

**Key Questions For Authors:**

1. How long does it take to run the full evaluation set?
2. How is the cumulative background setting implemented?
3. How would modern visual RL generalization methods perform?
4. What is the purpose of the ⟵ action?

**Limitations:**

There is no discussion of limitations whatsoever. The authors could be more open about the shortcomings of their work. Properly outlining limitations is important for setting readers' expectations and positioning the paper within the field.

**Strengths And Weaknesses:**

# Strengths

1. The writing is clear and coherent. It is easy to understand how the environment works and how the experiments are designed.
2. The distance and return metrics are suitable for measuring the generalization problem.
3. The inclusion of code snippets and plentiful imagery is useful for the readers to understand the benchmark.
4. The benchmark design is straightforward. The factorization into known visual axes is conceptually clean and makes the experimental setup intuitive.
5. The environment is extremely fast. The reported throughput and JAX-native implementation make large-scale sweeps feasible, which is a strong benefit compared to existing pixel-based RL benchmarks for zero-shot generalization.
6. There is a lot of visual variation across axes. The range of backgrounds, distractors, filters, and effects provides a wide variety of evaluation settings.

# Weaknesses

1. KAGE-Env is a single 2D side-scrolling domain with very simple dynamics and a monotonic objective of moving right. Although the focus is on the visual aspect, every environment instance could nevertheless be solved by just holding ⟶ and spamming JUMP (albeit with reduced returns due to the penalty for spamming JUMP). This allows agent with a reflexive policy to succeed. A 2D sidescroller domain can be reasonable for isolating visual factors. However, as it stands, the environment could use some additional factors of complexity.  For example:  1) introduce a *double-jump* mechanic, and make some surfaces too high to reach without it. 2) Add (moving) obstacles/entities that give a penalty on contact. 3) Add objects the agent needs to push to reach high surfaces. The task can get increasingly harder within the episode as the agent progresses. Last, what is the purpose of the ⟵ action? It seems to provide no utility for solving the task, and only acts as controller noise.
2. The *curated* pairs for KAGE-Bench are based on the performance of a single PPO-CNN baseline. This makes the benchmark tailored to the weaknesses of that specific model. We don’t see any other baseline, so we don’t know how much performance can vary. The pairs that exhibit a range of generalization behavior for PPO-CNN might yield a flat performance for a different method.
3. While the throughput (SPS) of KAGE-Env is impressive, there is no reporting of actual training time or hardware used for the experiments. This makes it difficult to assess the practical cost of running the benchmark. If I were to evaluate a new method on the full KAGE-Bench (34 pairs × 10 seeds × 25M steps), what should I expect in terms of time and computation requirements?
4. The results in Figure 6 (left) & Figure 7 (left) can be misleading. The authors describe “evaluation on progressively harder visual variants”. This is true for axes such as distractors and filters. Indeed, increasing the number of distractors or the radial light intensity makes the task more difficult. However, this works differently for backgrounds. Note that it is not explained how the cumulative color addition is implemented. I doubt the colors are stacked on top of one another or blended. Either the background transitions during side-scrolling, or the evaluation is split into different segments. In both cases,  performancea drop could arise purely from averaging over a larger set of individually challenging backgrounds. For example, if a policy succeeds on black but fails on any other color, then evaluation on {black} yields 1.0, {black, white} yields 0.5, {black, white, red} yields 0.33, and so on. This would produce a clean “dose-response” curve even though the underlying difficulty of each non-black background remains unchanged.
5. The success rate and progress metrics are redundant, and I’m not convinced they add much beyond reporting raw distance. The progress values are near-identical to distance (Tables 1 & 2), so it’s unclear what additional insight progress is supposed to provide. The success threshold D feels arbitrary and unnecessary. Moreover, it seems to be selected in a way that training performance just barely clears the threshold, which exaggerates the generalization gap when evaluation performance falls slightly short (Tables 1 & 2). Finally, the actual value of D=490 is only specified in the appendix (Corollary A.6).
6. The evaluation is weak. The PPO-CNN baseline provides a reference point, but we don’t know how challenging KAGE-Bench is for methods that are explicitly designed to handle visual distractions and domain shifts. There is a wide array of literature targeting visual generalization and distraction robustness in RL, including data augmentation [1, 2, 3, 4], masking/segmentation [5, 6, 7, 8], and object-centric representations [9, 10, 11]. It is no surprise that vanilla PPO fails at zero-shot generalization. Since no actual tailored methods are evaluated, it is unclear whether KAGE-Bench is challenging or relevant.

### Minor Points

1. lines 347-349 “using a standard PPO-CNN, […], trained from a single RGB frame”. I assume the authors mean PPO is trained solely from pixel observations without frame stacking. The wording could be better there.
2. Figure 4 is on page 3, but referenced for the first time on Page 7. This breaks the narrative flow.
3. The evaluation protocol is incomplete. The total timesteps, eval frequency, and eval episodes should be reported in the main paper, not only at the end of the appendix.

[1] Hansen, Nicklas, et al. "Stabilizing deep q-learning with convnets and vision transformers under data augmentation." 2021.

[2] Yuan, Zhecheng, et al. "Don't touch what matters: Task-aware Lipschitz data augmentation for visual reinforcement learning." 2022.

[3] Choi, Hyesong, et al. "Environment agnostic representation for visual reinforcement learning." 2023.

[4] Lei, Hao, et al. "Scattered data augmentation for generalization in visual reinforcement learning." 2025.

[5] Bertoin, David, et al. "Look where you look! saliency-guided q-networks for generalization in visual reinforcement learning." 2022.

[6] Grooten, Bram, et al. "Madi: Learning to mask distractions for generalization in visual deep reinforcement learning." 2023.

[7] Kim, Kyungmin, et al. "Make the pertinent salient: Task-relevant reconstruction for visual control with distractions." 2024.

[8] Vedant, Ozan Özdenizci, et al. "Learning robust representations for visual reinforcement learning via task-relevant mask sampling." 2025.

[9] Mosbach, Malte, et al. "Sold: Slot object-centric latent dynamics models for relational manipulation learning from pixels." 2024.

[10] Ugadiarov, Leonid, et al. "Object-centric dreamer." 2025.

[11] Zhang, Weipu, et al. "Objects matter: object-centric world models improve reinforcement learning in visually complex environments." 2025.

---

> ### Author Rebuttal · Authors · 2026-03-31
>
> We thank Reviewer DH9i for the detailed feedback and for carefully analyzing both the environment design and the evaluation. We appreciate the positive comments on clarity, visualization, and the scalability of the JAX implementation.
>
>
> > W1. Task simplicity and reflexive policy
>
> **We use simple dynamics to isolate visual generalization from confounds** (exploration, dynamics, reward shaping), ensuring performance drops are attributable to observation changes.
>
> A reflexive policy (Right+Jump) can move forward but is highly suboptimal. It achieves SR 0.59 with Return −5000, while a random policy has SR 0.00. In contrast, a learned policy achieves near-perfect success with much higher return. This shows the** task is not trivial and motivates using success rate to distinguish true completion from partial progress**.
>
> We thank the Reviewer for suggesting added mechanics (e.g., double jump, obstacles). While interesting, they introduce confounds beyond visual generalization, reducing attribution clarity; we will clarify this trade-off.
>
> The left action is kept to maintain classic platformer-style action space. Though rarely useful, it should be ignored by a good policy.
>
> > W2. Benchmark curation based on PPO-CNN
>
> The benchmark is not tailored to PPO-CNN; it serves as a probe to identify challenging visual axes.
>
> To address this, **we added robustness-oriented methods: RAD-PPO (cutout-color, crop-resize), CURL, DrQ, and SVEA**. RAD-PPO (cutout-color) improves over PPO-CNN but still shows large gaps (66–97% SR drop), while CURL, DrQ, SVEA, and RAD (crop-resize) fail to converge (SR ≈ 0.01). Full results: [https://tinyurl.com/anon-icml-kage](https://tinyurl.com/anon-icml-kage).
>
> We attribute this to the platformer’s spatial demands: precise geometry and timing make spatial augmentations disruptive. Thus, the observed difficulties are not PPO-CNN artifacts and persist across methods.
>
> > W3. Training cost and practical runtime
>
> For PPO-CNN, we trained on all 34 configurations with 10 runs each and 25M steps per run, totaling ~500 GPU-hours on 4 NVIDIA A100s. Each run takes ~1.5 hours (including checkpoints and videos) at ~52k steps/sec.
>
> Importantly, evaluating a new method does not require all 34 configs. In practice, a subset of challenging settings (as in our RAD experiments) is sufficient. We will add these details to clarify cost and reproducibility.
>
> > W4. Cumulative background setting
>
> Figures 6 and 7 (left) evaluate performance across distinct background configurations (not mixed within an episode), averaged over them. For example, if trained only on black, “val_b” is similar; “val_bw” uses 50% black and 50% white; “val_bwr” splits 33% black, white, and red. The agent succeeds on black and fails on unseen backgrounds.
>
> Under this protocol, performance drops reflect success on the training background and failure on new ones. This is the intended interpretation: performance as evaluation support expands beyond the training distribution.
>
> > W5. Metrics (distance, progress, success)
>
> Return alone does not reflect progress, as it mixes forward movement with penalties. We therefore introduce distance, which directly measures progress; together, return and distance capture efficiency and task completion.
>
> However, raw distance is hard to interpret. We define dist_to_success = 490 (near-optimal =500), enabling a binary success metric where success rate indicates near-optimal behavior.
>
> We emphasize that return and success rate are sufficient for evaluation, while distance and progress (normalized by dist_to_success) are for interpretability. We will keep only return and success rate in Table 2, move distance and progress to Appendix B (Table 3), and define dist_to_success in the main text.
>
> > W6. Lack of robustness-focused baselines
>
> **We added experiments with RAD-PPO, CURL, DrQ, and SVEA**. These methods either fail to learn with spatial augmentations or still show large generalization gaps, confirming that KAGE-Bench is non-trivial.
>
> > Min 1
>
> Yes, we indeed refer to training without frame stacking, using a single input frame. Thank you for the comment, we will clarify this in the text.
>
> > Min 2
>
> Thank you, we have added a reference to Figure 4 in the Introduction on page 2.
>
> > Min 3
>
> Thank you for the comment. We will move key evaluation details (timesteps, frequency, episodes) from the appendix to the main text.
>
> > Limitations
>
> We will add a dedicated limitations section clarifying that KAGE-Bench targets visual generalization only, uses a 2D platformer for controlled and scalable evaluation, and should be viewed as a diagnostic tool rather than a proxy for real-world deployment.
>
> We thank the Reviewer DH9i again for the detailed and constructive feedback. We hope the clarifications above address the concerns, and we will incorporate the suggested improvements in the revised version.
>
> [1] Laskin et al. (RAD, 2020)
>
> [2] Laskin et al. (CURL, 2020)
>
> [3] Kostrikov et al. (DrQ, 2020)
>
> [4] Hansen et al. (SVEA, 2021)

---

> > ### Author Rebuttal · Reviewer_DH9i · 2026-04-03
> >
> > 1. **Task Simplicity (W1)**. I understand the motivation for isolating visual generalization from other confounds, and I appreciate the authors' clarification. However, then I am lost what makes this interesting for RL. The benchmark is positioned as a contribution to visual RL, yet the simple design deliberately removes parts that make RL challenging and interesting. What remains is essentially a supervised perception problem embedded in an RL loop. The agent needs to detect its position relative to platforms and then select from a tiny action set to perform a near-fixed motor routine.
> >
> >     The mechanics I suggested (double-jump, obstacles) wouldn't necessarily introduce "confounds". They'd make the control problem require more than color/position detection, which would make the visual generalization findings more interesting and harder to dismiss.
> >
> > 2. **PPO-CNN Curation (W2)**. I didn’t mean to say the benchmark is tailored **to** PPO-CNN, I meant to say that solely PPO-CNN determines what is difficult or easy in the benchmark. The stronger methods (CURL, DrQ, and SVEA) appear to not even achieve performance on training, which makes little sense. Likely there was not enough time during the rebuttal to correctly re-implement them in JAX. Last, I don’t buy the reasoning on the “*spatial demands*”. Claiming a 2D platformer to have *precise geometry* is exaggerated. Realistic 3D environments with accurate physics simulation have spatial demands and precise geometry, not KAGE-env. Moreover, this framing is contradictory to the authors’ positioning of KAGE-env as reducing factors of complexity to isolate the representation learning part from images. It is rather ironic that in response to W1, you claim using **simple dynamics**, and now you claim **precise geometry**. How can you have both?
> > 3. **Practical Runtime (W3)**. 52K SPS does not seem like a lot for a JAX-native environment. Brax, Craftax, and JaxMARL land at around 300K-500K during training. Even Mujoco Playground, also using pixel rendering, gets ~400K on cartpole. Are there some throughput issues causing the lower SPS, or did you use a lower number of parallel environments for your experiments? 4 A100s should be capable of more.
> > 4. **Coloured Background (W4)**. Thanks for the clarification. I understand now that *val_bwrg* would mean 25% of evaluation episodes reserved for each color. I think the chosen evaluation setting is good, but the wording should be revised. I think it is not correct to refer to the color setting as *progressively harder visual variants*. For it to be legitimately "progressively harder", you'd need to show that the individual unseen colors themselves vary in difficulty (e.g., white is easier to transfer to than red, which is easier than green). But I doubt, this is the case. Currently, 25% black, 25% white, 25% red, 25% green would yield the same result at 25% black, 75% white.
> > 5. **Metrics (W5)**. It still seems to me that the 490 is selected to exaggerate the generalization gap and make the results seem more dramatic under the success metric. I completely don’t agree that raw distance is hard to interpret. There is nothing easier to interpret in the KAGE-env than forward distance. I therefore don’t see the need for the additional *success* metric. I agree that return can be misleading, because it contains reward shaping penalties and such. I would encourage the authors to stick to a single distance metric. Last, it is not clear to me why return captures *efficiency*.
> > 6. **Baselines (W6)**.Thank you for adding the RAD-PPO baseline. I believe this is a step towards improving the evaluation. As seen in Table 1, RAD, a method from 2020, already yields much higher evaluation return compared to PPO-CNN. If more recent and powerful (SOTA) methods for pixel-based generalization were evaluated, it is plausible that they already perform optimally on KAGE-bench, and the benchmark's utility is limited as it is already saturated.
> >
> >     The authors report that CURL, DrQ, and SVEA did not converge. I believe this is likely due to the challenge to re-implement these methods in JAX. RAD is trivial, as it only needs shifting 4 pixels, but the rest are more complicated to manage in the short rebuttal period, which is understandable.
> >
> >
> > Unfortunately, all of the points I originally raised still stand.

---

> > > ### Author Response · Authors · 2026-04-05
> > >
> > > We appreciate the Reviewer’s detailed feedback, but several of the points below appear to rest on misunderstandings of the benchmark’s design goals and experimental setup. We address each concern precisely.
> > >
> > > **Task Simplicity (W1)**. We believe this criticism relies on a premise inconsistent with the paper’s goal. KAGE-Bench is not designed to maximize task complexity, but **to isolate visual generalization while keeping the latent control problem fixed**. The agent is still trained end-to-end with RL from pixels, with delayed rewards, exploration, and policy optimization, but unrelated sources of variation are removed so failures can be attributed to observation shifts. In this sense, simplicity is not a limitation but the mechanism enabling causal interpretability. Adding mechanics such as double-jump or obstacles would increase complexity but also introduce interacting factors, weakening attribution. We therefore position KAGE-Bench as a **controlled diagnostic benchmark for visual RL**, complementary to more complex but less interpretable environments.
> > >
> > > **PPO-CNN Curation (W2)**. The claim about incorrect implementations is unfounded. We did not re-implement these methods. We used official PyTorch implementations (RAD-PPO, CURL, DrQ, SVEA) and interfaced them via a JAX-to-PyTorch wrapper. RAD-PPO (cutout-color) improves over PPO-CNN as expected, while RAD-PPO (crop-resize), CURL, DrQ, and SVEA fail. This isolates the cause to augmentation choice, not implementation, and is consistent with prior work on augmentation sensitivity. Importantly, we do not define the benchmark based on PPO-CNN performance. PPO-CNN is used as a **diagnostic probe** to reveal which visual shifts break a standard policy. Different methods may fail under different shifts, and this is precisely the purpose of KAGE-Bench: **to expose method-dependent failure modes rather than define a fixed notion of difficulty tied to a single baseline**.
> > >
> > > Second, regarding the claimed contradiction, there is no inconsistency between simple dynamics and spatial precision. The environment is intentionally simple at the latent level to remove confounds, but still imposes strict spatial requirements. Achieving near-optimal return requires precise jump timing, where even small deviations lead to penalties. Augmentations such as crop-resize and translation disrupt pixel-to-geometry correspondence, degrading distance estimation and control. Thus, the task has strong spatial demands in terms of observation consistency, not complex physics. We therefore disagree that this is contradictory; isolating representation learning under controlled dynamics is precisely what exposes these failure modes
> > >
> > > **Practical Runtime (W3)**. The Reviewer’s comparison is invalid. The reported ~52K SPS is full training throughput, **including CNN policy forward passes, backpropagation, optimizer updates, logging, checkpointing, and video recording on 128×128×3 RGB inputs**. In isolation, **the environment reaches ~33M steps/sec** (see Figure 3, as well as L37 and L104 in the main text). Experiments run on 4 A100 GPUs in parallel, but **SPS is reported per single GPU**
> > >
> > > **Coloured Background (W4)**. Thanks, the wording for Figure 6 will be corrected
> > >
> > > **Metrics (W5)**. The Reviewer’s claim that (D=490) is chosen to exaggerate results is incorrect. The optimal policy reaches ~500, so 490 corresponds to near-complete traversal. It is not tuned. Distance alone cannot distinguish partial progress from task completion. In our results, agents often reach high distance but fail to complete due to timing mistakes. Success rate separates these cases (see Table 2). Return captures efficiency because it penalizes unnecessary actions and poor timing
> > >
> > > **Baselines (W6)**. The concern about saturation is not supported by our results. RAD-PPO improves over PPO-CNN but still shows large gaps ($\delta$SR 66–97% on hardest configurations), indicating that even strong augmentation-based methods do not achieve robust generalization. While RAD is a competitive baseline, its improvement does not imply the benchmark is trivial or outdated. Importantly, KAGE-Bench is **not a fixed benchmark but a configurable diagnostic framework**: the 34 configurations are only a subset chosen for clarity, and can be extended as methods improve. Additionally, CURL, DrQ, and SVEA fail not due to implementation, but due to spatial augmentations (crop/shift) breaking geometry, consistent with RAD-PPO (crop-resize), isolating augmentation as the cause. Finally, we **did not re-implement these methods in JAX**; we used official PyTorch implementations via a JAX-to-PyTorch wrapper.
> > >
> > > In summary, most of the concerns raised rely on speculative assumptions not supported by the experimental setup (implementation issues), or expectations that contradict the benchmark’s purpose (maximizing complexity rather than isolating causality). The empirical results consistently support our claims across multiple methods and configurations.

---

### Official Review · Reviewer_UtUh · 2026-03-09

**Soundness:** 3
**Presentation:** 3
**Significance:** 2
**Originality:** 3
**Overall Recommendation:** 5
**Confidence:** 4

**Summary:**

The paper introduces KAGE-Env, a JAX-native 2D platformer where visual axes (background, lighting, filters, agent appearance, etc.) can be controlled independently while the underlying MDP is kept fixed. Building on this, the authors define KAGE-Bench, a benchmark of 34 train-evaluation pairs targeting specific visual axes. Using a PPO-CNN baseline, they show that generalization failures are strongly axis-dependent, with photometric and background shifts being the most damaging. The JAX implementation enables up to 33M steps per second on a single GPU, making large-scale sweeps feasible.

**Compliance With Llm Reviewing Policy:**

Affirmed.

**Final Justification:**

The rebuttal directly addressed my concerns and, I believe, improved the submission quality. Consequently I decided to improve my score.

**Key Questions For Authors:**

**Questions and Recommendations**

- **Positioning against Hansen & Wang (2021)** This prior work should be discussed more carefully. The authors should explicitly explain how KAGE-Bench defends its contribution against this potential combination.

- **Lack of robustness-aware baselines and benchmark validation** The benchmark would be significantly more useful if the authors tested at least one or two methods that are explicitly designed for visual robustness, such as data augmentation approaches (DrAug, RAD) or representation learning methods.

- **Missing visual shift types** There are arguably more relevant shifts for real application. Did the authors consider including them? As Future work?

**Limitations:**

**Limitations:**

- The simplicity of the platformer task is a real limitation: it is unclear whether an algorithm that performs well on KAGE-Bench will transfer its robustness to harder, more realistic settings. One of the limitations of this benchmark is that it is built on a single task: a 2D side-scrolling platformer with simple dynamics.

**Minor Comments:**

- The paper focuses entirely on visual generalization, but the framework has broader potential uses. Environments with independently controllable visual axes could naturally support continual learning research (where shifts arrive sequentially) or curriculum learning (where difficulty is gradually increased along a visual axis). Highlighting these potential applications would strengthen the paper's contribution beyond its current framing.

- It is worth highlighting more explicitly that the JAX RL ecosystem currently lacks pixel-based environments. To my knowledge, only MuJoCo Playground (Zakka et al., 2025) offers a way to obtain image observations in a JAX-native setting, and OCTAX (Radji et al., 2025) provides natively visual tasks but is limited to monochrome arcade environments. KAGE-Env therefore fills a real gap in the community, and making this point more prominently in the paper would strengthen the contribution's positioning and help readers appreciate why a fast, pixel-based JAX environment is needed.

**Strengths And Weaknesses:**

**Strengths:**

- The core idea of factorizing the observation process into independently controllable axes while keeping the latent MDP fixed is clean and well-motivated.

- The formal treatment via induced state policies (Theorem 4.2) gives the benchmark some theoretical grounding.

- The JAX implementation and the reported throughput is a practical contributions that lower the cost of systematic evaluation.

- The observation that return alone can mask generalization failures, with distance and progress staying high while success collapses, is an important and well-demonstrated empirical finding.

**Weaknesses:**

- **Positioning against Hansen & Wang (2021)** Hansen & Wang (2021) already propose a benchmark for visual generalization in continuous control under controlled appearance changes (color randomization, video backgrounds) on the DeepMind Control Suite, which involves more realistic, physically grounded, and diverse tasks than a 2D platformer. More importantly, it is now possible to combine MuJoCo Playground (Zakka et al., 2025), which uses the MJX backend and is fully JAX-native, with the visual shift framework of Hansen & Wang. Such a combination would offer both the scalability argument the authors make and more complex, realistic tasks.

- **Lack of robustness-aware baselines and benchmark validation** All experiments use a vanilla PPO-CNN that is not designed to handle visual shifts. It is expected that such an agent will fail under a distribution shift. More importantly, the paper does not validate that KAGE-Bench produces rankings that are consistent with the broader literature. If method A is known to outperform method B on existing benchmarks, do they rank in the same order on KAGE-Bench? Even a small experiment demonstrating this consistency would substantially strengthen the claim that the benchmark is meaningful and not just an isolated diagnostic tool.

- **Missing visual shift types** The current axes cover color, lighting, textures, and agent appearance. More geometric and perspective-based perturbations, such as image rotation, perspective distortion, or viewpoint changes, are absent.

---

> ### Author Rebuttal · Authors · 2026-03-31
>
> We thank Reviewer UtUh for the detailed and constructive feedback. We especially appreciate the recognition of the clean known-axis design, the theoretical grounding via induced state policies, and the practical value of the high-throughput JAX implementation, as well as the observation that KAGE-Env helps address the lack of pixel-based environments in the JAX RL ecosystem.
>
>
> Below we address the key concerns and questions.
>
> > W1. Positioning against Hansen & Wang (2021)
>
> Hansen & Wang (2021) study visual generalization under controlled appearance changes in continuous control settings. However, their setup still operates in environments where multiple factors interact, and isolating the exact source of performance degradation remains challenging. In addition, large-scale, axis-isolated sweeps are computationally expensive in such environments.
>
> In contrast, **KAGE-Env is intentionally designed to enforce strict factorization of the observation process**: each train–evaluation pair modifies exactly one visual axis while keeping the latent MDP fixed. This enables direct attribution of performance changes to specific visual factors, which is difficult to achieve in more complex domains.
>
> In continuous control (e.g., Hansen & Wang, DMControl), agents mainly localize and perform smooth actions, making them tolerant to spatial distortions. In contrast, KAGE-Env requires precise interaction with a fixed layout (distance estimation, jump timing), making it highly sensitive to spatial consistency.
>
> Its JAX-native design enables large-scale sweeps, and combining MJX-based simulators with visual shift frameworks is a promising future direction.
>
>
> > W2. Lack of robustness-aware baselines and benchmark validation
>
>
> We agree that evaluating robustness-aware methods strengthens the benchmark. In addition to PPO-CNN, **we have conducted new experiments with several established approaches designed for visual robustness, including RAD-PPO [1] (with cutout-color and crop-resize augmentations), CURL [2], DrQ [3], and SVEA [4]**. The full results are provided at the anonymous link: https://tinyurl.com/anon-icml-kage.
>
>
> These experiments lead to two key findings:
>
> (1) Robustness methods help but do not solve the hardest shifts: RAD-PPO improves over PPO-CNN yet still shows large gaps ($\delta$SR 66–97%).
>
> (2) Spatial augmentations can be detrimental in this setting.
> Methods relying on spatial augmentations, such as CURL, DrQ, SVEA, and RAD-PPO with crop-resize, fail to converge even on the base training configuration (SR 0.01%, Return  -2000). We attribute this to the disruption of spatial consistency: random crops and shifts alter the geometric structure of the scene, preventing the agent from reliably estimating distances and interacting with the layout.
>
> This behavior contrasts with continuous control benchmarks (e.g., Hansen & Wang (2021)), where such augmentations are effective because the task primarily requires agent localization and smooth control, rather than precise interaction with a fixed spatial structure.
>
> Together, these results demonstrate that **KAGE-Bench is not trivially solved by standard robustness techniques and that it captures failure modes that persist across multiple classes of methods.**
>
>
>
> > W3. Missing visual shift types
>
> The benchmark focuses on appearance-based factors (color, lighting, textures, distractors, agent appearance) to maintain clean control and factorization. Geometric transformations are important but harder to incorporate without breaking this separation in a 2D platformer. We consider them future work and will note this in the paper.
>
>
> > Limitations
>
> The platformer’s simplicity is intentional to isolate visual factors. By fixing the latent MDP and varying only observations, KAGE-Bench avoids confounds present in complex environments. It serves as a fast diagnostic: failure here suggests likely failure in harder settings.
>
>
> > Minor Comments
>
> > The paper focuses entirely on visual generalization, but the framework has broader potential uses.
>
> Thank you for the suggestion. Continual and curriculum learning are natural extensions we did not explore, and we will discuss these directions in the camera-ready version.
>
>
> > It is worth highlighting more explicitly that the JAX RL ecosystem currently lacks pixel-based environments
>
> Thank you for highlighting this. We will revise Related Work to emphasize that KAGE-Env helps address the lack of pixel-based environments in the JAX RL ecosystem.
>
> We thank the Reviewer UtUh again for the time and effort spent on evaluating our work and for the valuable feedback. We hope the responses above address the questions, and we are happy to provide further clarification if needed.
>
>
> [1] Laskin et al. (RAD, 2020)
>
> [2] Laskin et al. (CURL, 2020)
>
> [3] Kostrikov et al. (DrQ, 2020)
>
> [4] Hansen et al. (SVEA, 2021)

---

> > ### Author Rebuttal · Reviewer_UtUh · 2026-04-04
> >
> > Thanks for the clarification, which solves my concerns. I'm looking forward to seeing the updated version of the manuscript. I will raise my score.

---

> > > ### Author Response · Authors · 2026-04-08
> > >
> > > We’re glad your concerns were resolved and we sincerely appreciate your decision to increase your score. Thank you for your helpful feedback and suggestions.

---

### Official Review · Reviewer_Xmrf · 2026-03-12

**Soundness:** 4
**Presentation:** 3
**Significance:** 4
**Originality:** 4
**Overall Recommendation:** 4
**Confidence:** 4

**Summary:**

This paper introduces KAGE-Bench, a novel benchmark and associated fast JAX-based environment (KAGE-Env) for studying visual generalization in reinforcement learning (RL). The key contributions are:

A Formalized Problem Setup: The paper provides a rigorous mathematical framework for "known-axis visual generalization," formalizing a visual POMDP where the underlying latent dynamics and rewards are fixed, and only the observation kernel (renderer) changes. A core theorem proves that under this setup, a visual shift is equivalent to a shift in the policy's induced behavior in the latent MDP.

KAGE-Env: A lightweight, fully JAX-compatible environment built on the "Keen" game engine. It is designed for massive parallelization (demonstrated with 65,536 parallel environments), making it highly efficient for large-scale experimentation.

KAGE-Bench: A curated benchmark consisting of 34 train-evaluation configuration pairs, grouped into six distinct visual generalization axes (agent appearance, background, distractors, effects, filters, layout). Each pair isolates a single visual parameter change.

Comprehensive Evaluation: The paper presents extensive baseline results using a standard PPO-CNN agent, reporting multiple trajectory-level metrics (distance, progress, success rate) in addition to return. The results quantify the generalization gap across different visual axes, showing, for example, that changes in background and the introduction of distractors cause the largest performance degradation.

**Compliance With Llm Reviewing Policy:**

Affirmed.

**Key Questions For Authors:**

On the Choice of Baseline: The benchmark is currently evaluated with a single PPO-CNN baseline. Do you plan to or could you discuss how you envision the community using KAGE-Bench to compare different classes of methods (e.g., data augmentation (RAD, DrQ), regularization, architectural changes, world models)? Are there any immediate plans to release baseline scores for more advanced algorithms to help bootstrap future research?

On the Simplicity of the Visual Domain: Given the relative visual simplicity of the KAGE environment, how confident are you that generalization challenges identified here (e.g., sensitivity to new backgrounds) are predictive of challenges in more visually complex domains (e.g., 3D navigation with realistic textures)? Do you see KAGE-Bench as a final diagnostic, or more as a scalable and fast "unit test" for visual robustness that should be complemented by other benchmarks?

On the Underlying Causes of Generalization Gaps: Your results clearly show that different visual axes cause different magnitudes of generalization gaps (e.g., "Background" is harder than "Agent"). Do you have any hypotheses or preliminary analyses on why this is the case? For instance, is the CNN learning spurious correlations with background colors, or are the background changes fundamentally more disruptive to the learned features? Providing even a speculative explanation would greatly enrich the interpretation of these results.

On the Choice of Metrics: The paper introduces excellent trajectory-level metrics (distance, progress, success rate). Could you elaborate on how these metrics sometimes diverge from the return-based gap? For example, in Table 1 for the "Background" axis, the return gap is massive (691), but the progress gap (30.5%) seems less severe. Does this suggest the agent is still making progress but dying more often or getting negative rewards, leading to a lower return?

**Limitations:**

Yes

**Strengths And Weaknesses:**

Strengths:
•	The known-axis design is a real strength: train/eval shifts are constructed to change only the observation process while keeping dynamics and rewards fixed, which makes attribution much cleaner than in prior visually entangled benchmarks.
•	The benchmark goes beyond return and also reports distance, progress, and success, which is well motivated and helps reveal failures that reward alone could hide.
•	The environment/benchmark engineering is solid: the paper provides six suites and 34 controlled train-eval pairs, and the fast JAX implementation makes broader sweeps practical.

Weaknesses:
•	The benchmark curation is not yet fully demonstrated to be method-agnostic: the 34 pairs are selected via a pilot sweep using a standard PPO-CNN, so the retained shifts may reflect what this baseline finds difficult rather than a fully method-agnostic notion of visual difficulty. It would be stronger to check whether the suite composition is stable under alternatives such as RAD[1] or DrQ-v2[2].
•	The suite sizes are unbalanced across visual axes, which weakens cross-axis comparability. Some suites are represented by many pairs while others have only a few or even single, the axis-level averages in Table I are not equally statistically supported, and conclusions about relative axis difficulty should be stated more cautiously.
•	Most of the empirical conclusions are drawn from a single baseline, so it is still unclear how much of the reported axis ranking is benchmark-intrinsic versus method-dependent.
[1] Laskin M et al. Reinforcement learning with augmented data. Advances in neural information processing systems, 2020.
[2] Yarats D et al. Mastering visual continuous control: Improved data-augmented reinforcement learning. arXiv preprint, 2021.

---

> ### Author Rebuttal · Authors · 2026-03-31
>
> We thank Reviewer Xmrf for the detailed and thoughtful feedback, and we greatly appreciate the recognition of the benchmark’s controlled known-axis design, the comprehensive metric suite, and the efficiency of the JAX implementation.
>
> Below, we address the key concerns and questions.
>
> > W1, Q1. “ The benchmark curation is not yet fully demonstrated to be method-agnostic…”
>
> KAGE-Bench is not a leaderboard but a **diagnostic tool for isolating visual generalization failures**. We use PPO-CNN as a simple, interpretable probe to reveal which visual axes are genuinely challenging.
>
> To demonstrate that the suite composition and axis difficulties are not merely artifacts of PPO-CNN, **we evaluated a stronger baseline requested by the Reviewer: RAD-PPO [1] utilizing cutout-color and crop-resize augmentations. Furthermore, we conducted experiments with the popular CURL [2] , DrQ [3] , and SVEA [4] algorithms**. The table and figures containing all these results can be found at this anonymous link: https://tinyurl.com/anon-icml-kage.
>
> RAD-PPO (cutout-color) improves over PPO-CNN but still shows large gaps (ΔSR 66–97%), indicating intrinsic difficulty. In contrast, spatial augmentations (CURL, DrQ, SVEA, RAD crop-resize) fail even on the base task, as they break spatial structure—highlighting the distinct challenges of our benchmark.
>
>
>
> > W2. “The suite sizes are unbalanced across visual axes…”
>
> The benchmark is not intended to estimate a statistically fair “difficulty ranking” of axes, but to **provide a set of clean and interpretable stress tests that isolate specific failure modes**.
>
> Different axes allow different numbers of clean variations. Forcing balance would require artificial or redundant configurations, reducing clarity and attribution precision.
>
> Our priority is that each train-eval pair corresponds to a clear and isolated change in the observation process. This ensures that when performance drops, it can be directly attributed to that specific visual factor. From this perspective, diversity and interpretability of configurations are more important than equal counts across axes.
>
> > W3. “ Most of the empirical conclusions are drawn from a single baseline…”
>
> The goal is to study how isolated visual changes affect policy behavior. We show that some factors cause much larger performance drops despite identical task, dynamics, and rewards.
>
>
> PPO-CNN, as a standard architecture without specialized biases or augmentations, provides a **clean view of how visual changes affect performance**. Using specialized methods would confound whether improvements come from easier shifts or method-specific design.
>
> While specialized methods can outperform PPO-CNN, we evaluated stronger approaches (RAD, CURL, DrQ, SVEA) to verify our conclusions. The results confirm that the hardest axes (e.g., background, filters) still cause large generalization gaps for RAD-PPO, while spatial methods fail entirely.
>
> This confirms that our conclusions regarding the underlying difficulty of these visual shifts reflect fundamental challenges in visual generalization, rather than just the limitations of a single baseline.
>
> > Q2. On the Simplicity of the Visual Domain.
>
> The visual simplicity is intentional to ensure causal interpretability by removing confounds present in complex 3D settings. Despite this, the task requires precise spatial reasoning (distance estimation and timing), making it unforgiving. In practice, methods like CURL, DrQ, and SVEA fail due to spatial augmentations breaking layout information. KAGE-Bench thus serves as a diagnostic “unit test” for visual generalization.
>
>
>
>
> > Q3. On the Underlying Causes of Generalization Gaps
>
> A likely explanation is that standard CNN policies rely on global visual statistics that are not stable under shift. Changes like background, filters, or distractors affect large parts of the image and change color distributions, textures, and contrast. This can break the learned features.
>
> Agent appearance changes are more localized and preserve scene structure, making them less disruptive. Some shifts allow forward movement but break task completion, indicating the policy extracts partial signal but fails to identify goal-relevant features.
>
> > Q4. On the Choice of Metrics
>
> **Return alone is insufficient, as it mixes progress and penalties**. We therefore report distance, progress, and success rate to distinguish partial progress from task completion. In practice, agents often achieve non-zero progress but near-zero success, highlighting incomplete behavior.
>
>
> We thank Reviewer Xmrf for the constructive feedback and hope our clarifications and added RAD-PPO experiments address the concerns. We are happy to provide further details if needed.
>
>
> [1] Laskin et al. (RAD, 2020)
>
> [2] Laskin et al. (CURL, 2020)
>
> [3] Kostrikov et al. (DrQ, 2020)
>
> [4] Hansen et al. (SVEA, 2021)

---

> > ### Author Rebuttal · Reviewer_Xmrf · 2026-04-05
> >
> > Thanks for the response, and my concerns have been well addressed.

---

> > > ### Author Response · Authors · 2026-04-06
> > >
> > > We thank the Reviewer for the time and effort devoted to our work, are glad that their concerns have been fully addressed, and hope they will revise their score accordingly.

---

### Official Review · Reviewer_k8uV · 2026-03-13

**Soundness:** 3
**Presentation:** 3
**Significance:** 3
**Originality:** 3
**Overall Recommendation:** 5
**Confidence:** 4

**Summary:**

This paper points out that existing benchmarks fail to attribute failures to specific factors. To address this issue, the authors introduce KAGE-Bench. KAGE-Bench is JAX-native and constructs environments in a fine-grained manner by strictly controlling rendering variables, enabling more efficient identification of the factors that lead to algorithm failures. Experimental results further show that PPO-CNN struggles to perform well across all scenarios, with performance degrading significantly when the environments contain more visual disturbances.

**Compliance With Llm Reviewing Policy:**

Affirmed.

**Key Questions For Authors:**

1. Could the authors test some recent mainstream RL algorithms on the hard configuration in this benchmark to demonstrate whether KAGE-Bench can evaluate a broader range of RL methods?
2. Could the complexity of the benchmark tasks be further increased, for example, by introducing additional reward and penalty mechanisms rather than focusing on distance?

**Limitations:**

Insufficient limitations are presented. Task complexity should be introduced and compared.

**Strengths And Weaknesses:**

### Strengths
**Fine-grained analysis.**
KAGE-Bench enables more detailed evaluation by controlling specific rendering variables (such as background) without changing environment dynamics and other environment factors, allowing researchers to identify the strengths and weaknesses of existing algorithms. It also introduces other metrics through theoretical analysis, which provide a more comprehensive evaluation of algorithm performance.

**Diverse experimental configurations.**
KAGE-Bench supports a wide range of rendering parameters, background adjustments, and characters, covering a wide range of difficulty levels.

**Fast and scalable.**
KAGE-Bench is JAX-native, capable of conducting a large number of environment steps in a short time, and is suitable for high-throughput evaluation.

### Weaknesses

**Limited coverage of RL algorithms.**
The experiments only include PPO-CNN, which makes it difficult to demonstrate whether the benchmark is also effective for analyzing stronger baseline methods.

**Relatively simple and narrow task settings.**
The task scenarios appear to be limited in complexity, which may restrict the benchmark’s ability to reflect the challenges encountered in real-world scenarios.

Overall, the paper addresses a meaningful limitation of prior benchmarks, but it would be helpful if the authors more explicitly discussed the limitations related to task complexity and baseline diversity.

---

> ### Author Rebuttal · Authors · 2026-03-31
>
> We thank Reviewer k8uV for the positive assessment and insightful feedback. We especially appreciate the recognition of the fine-grained control over rendering variables enabling factorized evaluation, and the high-throughput JAX-native design.
>
> We address the specific concerns below.
>
> > W1, Q1. Limited coverage of RL algorithms
>
> We would like to emphasize that the goal of this work is not to benchmark algorithms per se, but to introduce a diagnostic benchmark that isolates failure modes of visual generalization. **We intentionally use PPO-CNN as a controlled probe baseline** for three reasons:
>
> 1. Isolation of failure modes: PPO-CNN is a standard, widely understood architecture whose behavior is well characterized. This makes it suitable for revealing axis-dependent failure patterns, rather than obscuring them behind architectural complexity.
>
>
> 2. Demonstrating benchmark sensitivity: Even this standard baseline exhibits strong axis-dependent collapse, confirming that KAGE-Bench exposes meaningful and non-trivial generalization failures.
>
>
> 3. Identifying hard regimes for further stronger algorithms benchmarking: PPO-CNN allows us to systematically identify the most challenging axes (e.g., background, filters), which then serve as stress-test configurations for evaluating future methods.
>
> To address the Reviewer’s request, **we evaluated stronger baselines including RAD-PPO (crop-resize, cutout-color), as well as CURL, DrQ, and SVEA**. We will include these results and discussion in the revised version (see https://tinyurl.com/anon-icml-kage).
>
> On the 8 hardest configurations identified by PPO-CNN (Table 1: background 1, 8; distractors 5; effects 3; filters 1, 4, 9; layout 1), we observe two key takeaways:
>
> 1. Validating benchmark sensitivity: As expected, RAD-PPO (cutout-color) outperforms PPO-CNN on 7/8 configurations, confirming that KAGE-Bench reflects real algorithmic improvements rather than arbitrary difficulty.
>
>
> 2. Significant challenges remain: Despite these improvements, RAD-PPO (cutout-color) still exhibits large generalization gaps ($\delta$ SR = 66%–97%), falling far short of robust zero-shot generalization.
>
> CURL, DrQ, SVEA, and RAD-PPO (crop-resize) fail to converge even on the base task (SR  0.01%, Return  −2000). We attribute this to spatial augmentations (crop/shift) breaking geometric structure, preventing accurate distance estimation and timing; in practice, crops can even remove the agent from the frame.
>
> While such spatial distortions might be very useful in standard continuous control tasks (e.g., the Distracting Control Suite), **they might be catastrophic in a platformer setting where the agent must interact with a rigid layout to solve the task**. This specific failure mode further underscores the unique and challenging nature of our benchmark compared to existing continuous control alternatives.
>
> Thus, even stronger visual augmentation-based methods either fail to learn the base task due to disrupted spatial structures or fail substantially under controlled visual shifts. This demonstrates that **KAGE-Bench remains challenging and highly informative beyond classic baselines**.
>
>
> > W2, Q2. Task simplicity and limited complexity
>
> **The task simplicity is intentional to ensure causal interpretability.** In complex environments, performance drops are confounded by multiple factors, making it impossible to isolate visual generalization. By fixing the latent MDP and varying only observations, KAGE-Bench enables clean attribution of failures to specific visual axes.
>
> Importantly, this does not limit practical relevance: if an algorithm fails under these controlled settings, it is unlikely to generalize in more complex real-world scenarios. Conversely, KAGE-Bench serves as a fast, high-throughput diagnostic stage before evaluation on more realistic but slower benchmarks.
>
> While complexity is intentionally controlled, the framework itself is not limited:
>
> 1. The environment exposes 93 configurable parameters and is fully defined via YAML (Appendix G);
>
> 2. Reward structure and objectives can be easily modified (Appendix G);
>
> 3. The design naturally supports code extensions to other rewards and more complex objectives, while preserving the known-axis property.
>
> Basic yaml configuration also supports reward function changes (Appendix G).
>
> > Limitations
>
> We will add a dedicated limitations section clarifying that KAGE-Bench targets visual generalization only, uses a simplified 2D platformer for controlled and scalable evaluation, and should be viewed as a diagnostic benchmark rather than a direct proxy for real-world performance.
>
> We thank Reviewer k8uV again for the positive assessment and thoughtful feedback. We hope our responses have addressed the questions and concerns raised, and we welcome any further discussion.
>
>
> [1] Laskin et al. (RAD, 2020)
>
> [2] Laskin et al. (CURL, 2020)
>
> [3] Kostrikov et al. (DrQ, 2020)
>
> [4] Hansen et al. (SVEA, 2021)

---

> > ### Author Rebuttal · Reviewer_k8uV · 2026-04-02
> >
> > Thanks for the clarification, which solves my concerns. I will keep the score as it is.

---

> > > ### Author Response · Authors · 2026-04-08
> > >
> > > Thank you once again for your efforts and your positive evaluation.

---

### Decision · Program_Chairs · 2026-04-30

**Decision:**

Accept (regular)

**Comment:**

KAGE‑Bench offers a fast, JAX‑native benchmark that isolates visual shifts along controllable axes—a clean design.
However, major weaknesses undermine its contribution. The benchmark is curated and validated using only a single PPO‑CNN baseline, making it unclear whether results generalize to other algorithms (e.g., DrQ‑v2, RAD). The environment is a trivial 2D side‑scroller solvable by reflexive policies, lacking realistic dynamics. No robustness‑aware methods are evaluated, so the benchmark’s challenge level is unproven. Practical cost (training time, hardware) is not reported.